# Neural Logic Networks for Interpretable Classification

## Abstract

Traditional neural networks have an impressive classification performance, but what they learn cannot be inspected, verified or extracted. Neural Logic Networks on the other hand have an interpretable structure that enables them to learn a logical mechanism relating the inputs and outputs with AND and OR operations. We generalize these networks with NOT operations and biases that take into account unobserved data and develop a rigorous logical and probabilistic modeling in terms of concept combinations to motivate their use. We also propose a novel factorized IF-THEN rule structure for the model as well as a modified learning algorithm. Our method improves the state-of-the-art in Boolean networks discovery and is able to learn relevant, interpretable rules in tabular classification.

## 1 Introduction

Neural networks have revolutionized Machine Learning (ML) with unprecedented performance in perception tasks, ranging from prediction of complex phenomena to recognition and generation of images, sound, speech and text. However, this impressive performance is accompanied by a lack of explainability of how it is achieved, with neural networks being treated as black-box models due to the opaque nature of their learned parameters. As a result, it has often been claimed that the information that a neural network has learned cannot be inspected, verified or extracted.

As these black box models are increasingly being used to support or automate decision making, transparency has become a critical concern, giving rise to the field of eXplainable Artificial Intelligence (XAI) (Arrieta et al., 2020; Calegari et al., 2020). This is especially important for domains where ethics plays a pivotal role such as medecine, transportation, legal, finance, military (Adadi & Berrada, 2018) and scientific discovery. In those contexts, a prediction or decision is only useful when it is accompanied by an explanation of how it was obtained, as well as by an assurance that is is not the result of unacceptable biases, such as gender or ethnicity for example.

In parallel with this increasing demand for transparency, research on *neuro-symbolic* methods has become more popular. This family of methods aims to combine neural (also known as connectionist or sub-symbolic) with symbolic (logical) techniques to obtain a best-of-both-world scenario, with the complementary strengths of both paradigms. Neural methods are best at perception tasks and use continuous vector representations to learn a distributed representation from raw data, making them fast, strong at handling unstructured data and robust to noise and errors in the data (Yu et al., 2023). On the other hand, symbolic methods are best at cognition tasks and use discrete logical representations to reason deductively about knowledge, making them provably correct, human-intelligible, and with strong generalization ability. Neuro-symbolic methods include many different approaches to unify these two paradigms: neural implementations of logic, logical characterizations of neural systems, and hybrid systems that combine both in more or less equal measures (Besold et al., 2021).

One branch of hybrid neuro-symbolic methods defines new types of neural networks where the neurons represent logical AND and OR combinations, as opposed to the linear combination with non-linear activation of the classical perceptron. Such neural AND/OR networks aim to learn a logical mechanism relating the inputs and outputs that involve only AND, OR and NOT operations, resulting in a transparent and interpretable model. Ironically, the very inception of classical neural networks was justified by their ability to model such AND/OR networks (McCulloch & Pitts, 1943).

**Related work**  Neural networks with neurons that explicitly represent AND/OR operations were first defined by Pedrycz (1993); Hirota & Pedrycz (1994). Their learnable AND/OR nodes were defined in fuzzy logic for general t-norms (fuzzy AND) and t-conorms (fuzzy OR), while we adopt the product t-norm and t-conorm which admit a probabilistic interpretation. Moreover, their definition of the AND node used weights in a counterintuitive manner, having the opposite behavior of what we now understand as weights, *i.e.* whether an input is included in the AND operation. The OR node with general t-norms and t-conorms was also independently defined by Gupta (1993); Gupta & Rao (1994), which included an additional activation function. At the time, the idea of defining new neurons with fuzzy logic to be used in neural networks was an active field, referred to as Hybrid Fuzzy Neural Networks by Buckley & Feuring (1998). With a pre-processing layer to define fuzzy sets $A_{i,k}$ for each input $x_i$, followed by an AND layer and a final OR layer, a Neuro-Fuzzy classifier (Fuller, 1995) could be developed that would learn IF-THEN rules for each class $Y_k$ like

$$\textbf{IF } x_1 \in A_{1,k} \text{ AND } ... \text{ AND } x_d \in A_{d,k}\textbf{, THEN } x \in Y_k.$$

Furthermore, by working with explicit AND/OR nodes instead of perceptrons, problem-specific expert knowledge can be pre-encoded into the network before the learning to be refined and finally extracted from the learned network. This idea was first tried with normal perceptrons in Knowledge-Based Artificial Neural Networks (Towell & Shavlik, 1994). The initial knowledge was successfully encoded into the network, but it could not be re-extracted from the network after learning, because of its distributed representation.

AND/OR neural networks with product fuzzy logic were rediscovered in Payani & Fekri (2019a; 2020) under the name Neural Logic Networks (NLN), which we adopt as well. Their formulation was obtained from an expected desiderata rather than a rigorous logical or probabilistic modeling. The same definitions were then reused by Zhang et al. (2023) and developed further by Wang et al. (2020; 2021; 2024) into the Rule-based Representation Learner (RRL). To combat their notorious vanishing gradient (van Krieken et al., 2022), the RRL uses a new method called gradient grafting to learn the weights, along with approximated definitions of the AND/OR nodes which introduced new hyper-parameters to further improve learning. The RRL also introduced learnable upper and lower bounds to define the pre-processing sets $A_{i,k}$ in the case of continuous inputs $x_i$. Compared to our approach, these AND/OR nodes are limited in two significant ways. Firstly, they cannot consider negated inputs, *i.e.* using their contrary, without doubling the number of weights, whereas we use a single weight to model both cases. Secondly, their modeling assumes that all the relevant data is observed and given, while our formulation takes into account the impact of unobserved relevant data.

Other attempts at AND/OR neural networks were also created by others. Cingillioglu & Russo (2021) constrained the bias of regular perceptrons to obtain either an AND node or an OR node, depending on a hyper-parameter that is tuned during learning. Like us, their approach also considers both inputs and their contrary with a single weight. However, the magnitude of their weights do not explicitly represent their relative importance, unlike our weights which directly represent probabilities. Moreover, their formulation also fails to take into account unobserved data. Sato & Inoue (2023) cleverly uses a ReLU network with constrained biases to learn a 2-layer AND/OR network, but their approach only works for perfect binary data with no noise or errors. Another type of model that produces similar IF-THEN rules is decision trees, and their generalization decision diagrams (Florio et al., 2023). These models are represented by rooted directed graphs in which every node splits the possible values of one or more input $x_i$ in two or more branches, thus dividing up the input space into discrete bins that belong to the same class $Y_k$.

Due to their probabilistic formalization, NLNs also serve as probabilistic models of the target classes/labels when conditioning on the input features. In doing so, they implicitly learn Probabilistic Graphical Models (PGM), which encode the conditional structure of random variables in graph form (see appendix B.1.1 for the PGM behind NLN's probabilistic modeling). However, NLNs require an additional approximation to ensure tractability in practical settings. With additional assumptions regarding the direction of causality, NLNs can also be viewed as a special form of structural causal model (Peters et al., 2017) in which the assignment functions have a clear interpretation as AND/OR combinations of binary random variables, some of which may be unobserved. However these stronger causality assumptions are not required to use NLNs, and are only needed to produce interventional distributions or counterfactuals. NLNs are also related to probabilistic circuits (Choi et al., 2021) which study the tractability of probabilistic queries in sum-product networks via structural constraints. In particular, logistic circuits (Liang & Van den Broeck, 2019) are probabilistic

circuits with strong structural constraints that combine structure learning and logistic regression to learn AND/OR networks for interpretable classification. However, unlike NLNs, their final learned AND/OR networks contain non-quantized weights which are not as easily interpretable.

Finally, other approaches with similar names have also been proposed, in reasoning rather than learning tasks. The first method to be named Neural Logic Network was proposed by Teh (1995) and reintroduced recently by Ding (2018). It defines neural networks with values $(t, f) \in [0, 1]^2$ representing three-valued truth-values (true, false, unknown), with two sets of weights per neuron. Although some arrangements of weights give rise to interpretable AND and OR combinations, a restricted form of learning is required to maintain this interpretability, not making use of the current powerful gradient-based methods. Another approach, called Logical Neural Networks (Riegel et al., 2020; Sen et al., 2022), encodes any first-order logic program template in a neural network. By learning this neural network from a relational database, it aims to discover the best first-order logic rule (on relations) of the form given by the template. In comparison, our approach, by learning from a tabular dataset, aims to discover the best propositional logic rule (on features) of any form.

**Structure and contributions** To our knowledge, for the first time,

- we present a theoretical formalism for the logical and probabilistic modeling behind NLNs and their AND/OR nodes (section 2);

- we introduce biases accounting for unobserved data as well as the possibility of using the contrary of a concept in the weights (section 2);

- we develop an interpretable structure for NLNs with factorized IF-THEN rule modules and appropriate input pre-processing for binary, categorical and continuous features (section 3.1);

- we propose a modified learning algorithm for NLNs with a rule reset scheme at every epoch and a post-learning simplification of the model to increase its interpretability (sections 3.2 and 3.2.2); and

- we test our method on two classification tasks: Boolean network discovery and interpretable tabular classification (section 4).

We will show that our NLN, with its factorized structure and extended modeling, is able to learn sparser and more interpretable rules than its predecessor, the RRL, in tabular classification. For instance, as illustrated in Figure 1, our NLN is able to discover the rules of tic-tac-toe, simply by predicting if $\times$ has won from the end-game board configuration.

## 2 Theory

### 2.1 Probabilistic modeling

We introduce the modeling with a toy example. We are given an object $x$ from which we can derive a number of binary properties $C_i$ about the object $x$, *i.e.* *concepts $C_i$* that are either present or absent in $x$. For instance, we might know whether it is a `ball` $x \in C_1$ where $C_1$ is the set of balls, whether it is `green` $x \in C_2$ where $C_2$ is the set of green objects, if it is `heavy` $x \in C_3$ with $C_3$ the set of heavy objects, and so on. For instance a light green ball $x$ would satisfy $x \in C_1$, $x \in C_2$, and $x \notin C_3$. From these properties, we are interested in predicting whether the object $x$ is a `green_ball` $x \in Y$ where $Y$ is the set of green balls. Moreover, by learning to predict whether an object $x$ is a green ball $Y$, we wish to discover the definition of a green ball $Y = C_1 \cap C_2$, *i.e.* that it is the set of objects that are both green and balls. More generally, we are interested in learning an *AND concept*

$$Y = \bigcap_{i \in \mathcal{N}} C_i,$$

where, in this case, its *necessary* concepts are indexed by $\mathcal{N} = \{1, 2\}$. We learn to estimate which concepts $C_i$ are necessary to $Y$, written $Y \subseteq C_i$ or equivalently $C_i \supseteq Y$, through the statistical learning process of

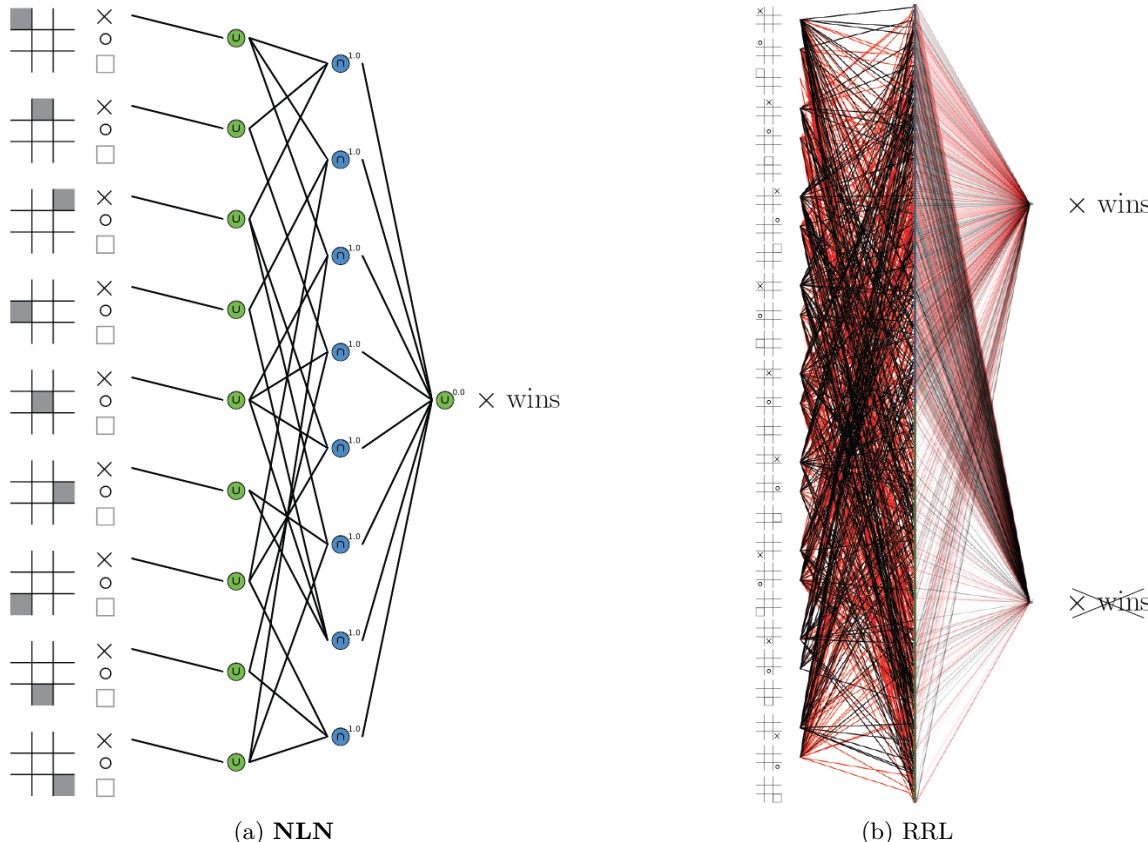

(a) **NLN**  (b) RRL

Figure 1: Interpretability of the learned AND/OR networks on tic-tac-toe, when trying to predict if $\times$ wins from the end board configuration

gradient descent. We do this by modeling the random event $x \in Y$ with respect to its necessary concepts $C_i$.

$$
\begin{aligned}
x \in Y &= \bigcap_{i \in \mathcal{N}} \left( x \in C_i \right) && [\text{definition of } Y] \\
&= \bigcap_i \left( (i \notin \mathcal{N}) \cup \left( (i \in \mathcal{N}) \cap (x \in C_i) \right) \right) && [\text{equivalent rewriting}] \\
&= \bigcap_i \left( (i \notin \mathcal{N}) \cup (x \in C_i) \right) && [\text{distributivity of } \cup \text{ over } \cap] \\
&= \bigcap_i \left( (C_i \not\supseteq Y) \cup (x \in C_i) \right) && [\text{definition of } \mathcal{N}]
\end{aligned}
$$

For instance, an object $x$ is a green ball $x \in Y$ if, for all its input concepts $C_i$, either they are not necessary concepts like heavy $C_3 \not\supseteq Y$, or they are present in $x$ like green $x \in C_2$ and ball $x \in C_1$.

We could otherwise have been interested in a different kind of concept, an *OR concept*. For instance, in a feedforward network where successive layers learn higher-level concepts from lower-level concepts, we could have a first layer of AND concepts that defines concepts such as $C'_1$: `green_ball`, $C'_2$: `yellow_cup`, $C'_3$: `blue_stick`, and so on and we could be interested in learning in the second layer $Y$: `green_ball_or_blue_stick`, *i.e.* $Y = C'_1 \cup C'_3$. More generally, we would be learning an OR concept

$$
Y = \bigcup_{i \in \mathcal{S}} C'_i,
$$

where, in this case, its *sufficient* concepts are indexed by $\mathcal{S} = \{1, 3\}$. We again use ML to estimate which concepts $C_i'$ are sufficient to $Y$, written $C_i' \subseteq Y$, by modeling $x \in Y$ in function of the $C_i'$.

$$
\begin{aligned}
x \in Y &= \bigcup_{i \in \mathcal{S}} \left( x \in C_i' \right) && \text{[definition of } Y] \\
&= \bigcup_i \left( \left( i \in \mathcal{S} \right) \cap \left( x \in C_i' \right) \right) && \text{[equivalent rewriting]} \\
&= \bigcup_i \left( \left( C_i' \subseteq Y \right) \cap \left( x \in C_i' \right) \right) && \text{[definition of } \mathcal{S}]
\end{aligned}
$$

For instance, an object $x$ is a green ball or a blue stick $x \in Y$ if, for any concept in the previous layer $C_i'$, it is both a sufficient concept like green ball $C_1' \subseteq Y$ or blue stick $C_3' \subseteq Y$ and it is present in $x$ like a green ball $x \in C_1'$ or a blue stick $x \in C_3'$.

In general, we are given as *input* a random variable $x = (x_1, x_2, ..., x_m)$ that is defined by $m$ measures, where each *input feature* $x_i$ can be binary, categorical or continuous. We are interested in predicting whether certain *target* concepts $Y_k$ are present in the input $x$, formalized as the random event $x \in Y_k$. These targets can be classes in binary or multi-class classification, or labels in multi-label classification. We use a network of concepts $C_i^l$ arranged in layers $l \in \{0, 1, ..., L\}$ of size $n^l$ with $i \in \{1, ..., n^l\}$, where the network's output is $Y_k = C_k^L$. The input layer $l = 0$ is made up of concepts $C_i^0$ that can be directly extracted from the input $x$. In other words, given an input $x$, we know for each concept $C_i^0$ whether it is present $x \in C_i^0$ or not $x \notin C_i^0$ (see details in Section 3.1.2). The subsequent layers of AND and OR concepts $C_i^l$ will try to learn relevant representations that relate *logically* the input concepts $C_i^0$ to the target concepts $Y_k$ through combinations of AND and OR operations.

In practice, we predict the labels by modeling their conditional probability $\mathbb{P}[x \in Y_k \mid x]$ given the input $x$. We do so by modeling for each concept $C_i^l$ its conditional probability $c_i^l(x) = \mathbb{P}[x \in C_i^l \mid x]$, starting from the input concepts $C_i^0$ for which we already know their probabilities $c_i^0(x) = \mathbb{P}[x \in C_i^0 \mid x]$. These input probabilities $c_i^0(x)$ can take any value in $[0, 1]$, with binary values $\{0, 1\}$ representing certain knowledge about $x$. In a feedforward structure, an AND (resp. OR) concept $C_i^l$ in layer $l$ takes its necessary (resp. sufficient) concepts from the previous layer $l - 1$. Unlike in the previous toy examples, we consider that a concept $C_j^{l-1}$ or its *contrary opposite* $(C_j^{l-1})^c$ can be a necessary or sufficient concept. For instance, a `ball_that_is_not_green` would have as necessary concepts `ball` and `green`$^c$, *i.e.* we would have

$$\texttt{ball} \supseteq \texttt{ball\_that\_is\_not\_green}, \quad \text{and} \quad \texttt{green}^c \supseteq \texttt{ball\_that\_is\_not\_green}.$$

We introduce a matrix of *weights* $A_{i,j}^l \in [-1, 1]$ for AND concepts and $O_{i,j}^l \in [-1, 1]$ for OR concepts to learn these necessary and sufficient relations, such that

$$
A_{i,j}^l = \underbrace{\mathbb{P}\big[C_j^{l-1} \supseteq C_i^l\big]}_{[A_{i,j}^l]_+} - \underbrace{\mathbb{P}\big[(C_j^{l-1})^c \supseteq C_i^l\big]}_{[A_{i,j}^l]_-}, \quad \text{and} \quad O_{i,j}^l = \underbrace{\mathbb{P}\big[C_j^{l-1} \subseteq C_i^l\big]}_{[O_{i,j}^l]_+} - \underbrace{\mathbb{P}\big[(C_j^{l-1})^c \subseteq C_i^l\big]}_{[O_{i,j}^l]_-}.
$$

It is important to note that the set inclusion relations $\subseteq$ are in opposite directions in necessary and sufficient relations. If a concept $C_j^{l-1}$ is necessary to concept $C_i^l$, then whenever we have $x \in C_i^l$, we must also have $x \in C_j^{l-1}$ since it is necessary, hence $C_j^{l-1} \supseteq C_i^l$. On the other hand, if a concept $C_j^{l-1}$ is sufficient to concept $C_i^l$, then whenever we have $x \in C_j^{l-1}$, we must also have $x \in C_i^l$ since $C_j^{l-1}$ is sufficient to $C_i^l$, hence $C_j^{l-1} \subseteq C_i^l$. When $A_{i,j}^l > 0$ or $O_{i,j}^l > 0$, the concept $C_j^{l-1}$ is believed to be necessary or sufficient to $C_i^l$ with probability $A_{i,j}^l$ or $O_{i,j}^l$. When $A_{i,j}^l < 0$ or $O_{i,j}^l < 0$, the absence of the concept $(C_j^{l-1})^c$ is believed to be necessary or sufficient with probability $|A_{i,j}^l|$ or $|O_{i,j}^l|$. This modeling allows a single parameter to learn both possibilities simultaneously, since they are contradictory. However, doing so also assumes that at all times at least one of those probabilities *e.g.* $\mathbb{P}\big[C_j^{l-1} \supseteq C_i^l\big]$, $\mathbb{P}\big[(C_j^{l-1})^c \supseteq C_i^l\big]$ is zero, with the rest of the probability mass distributed between the other and $\mathbb{P}\big[C_j^{l-1} \not\supseteq C_i^l, (C_j^{l-1})^c \not\supseteq C_i^l\big]$. In other words, this modeling introduces a cognitive bias in the model that "jumps to conclusions" regarding the sign of a causal role in the sense that it presumes only one sign is possible at once. It must consider one option fully, for instance $C_j^{l-1} \supseteq C_i^l$ with $A_{i,j}^l > 0$, before reaching $A_{i,j}^l = 0$ and being able to consider the other option $(C_j^{l-1})^c \supseteq C_i^l$ with $A_{i,j}^l < 0$.

Moreover, we consider the possibility of missing or *unobserved* data $u$ being part of the full *relevant* data $\omega = (x, u)$, where $\omega$ is the concatenation of the observed data $x$ and unobserved data $u$. For instance, we could be trying to predict whether an object $\omega = (x, u)$ is a `green_ball` without having any information in $x$ about the color of the object, only that it is a ball. In that case we would have to estimate statistically the probability that a ball is green $\mathbb{P}[u \in \texttt{green} \,|\, x \in \texttt{ball}]$ given the distribution of objects $\omega$ that we have seen. We now consider all the relevant data $\omega$ to model which concepts $C_i^l$ are present and with what probability $c_i^l(x) = \mathbb{P}[\omega \in C_i^l \,|\, x]$. Only for $l = 0$, we have the input concepts $C_i^0$ which depend only on the input $x$, and for which we are always given the probabilities $c_i^0(x) = \mathbb{P}[x \in C_i^0 \,|\, x]$. We use $\mathcal{X}$ and $\mathcal{U}$ to denote the space of possible $x$ and $u$ values respectively, so that we have $\omega \in \mathcal{X} \times \mathcal{U}$ and $C_i^l \subseteq \mathcal{X} \times \mathcal{U}$. With this final extension, an AND concept $C_i^l$ would be defined as

$$C_i^l = \left( \bigcap_{j \in \mathcal{N}_+} C_j^{l-1} \right) \cap \left( \bigcap_{j \in \mathcal{N}_-} (C_j^{l-1})^c \right) \cap \left( \mathcal{X} \times \underbrace{\bigcap_{z \in \tilde{\mathcal{N}}} \tilde{C}_z}_{\tilde{N}_i^l} \right),$$

where $\mathcal{N}_+$ and $\mathcal{N}_-$ are its *observed* necessary concepts in the previous layer, either using directly $C_j^{l-1}$ or using its opposite $(C_j^{l-1})^c$, and the $\tilde{C}_z$ represent unobserved concepts that depend only on the unobserved data $u$ and which are also necessary concepts of $C_i^l$. We define $\tilde{N}_i^l$ as being the intersection of all these necessary but unobserved concepts $\tilde{C}_z$, $\forall z \in \tilde{\mathcal{N}}$, *i.e.* we have $u \in \tilde{N}_i^l$ iff we have $u \in \tilde{C}_z$, $\forall z \in \tilde{\mathcal{N}}$. The random event of whether the AND concept is present $\omega \in C_i^l$ is then given by

$$\omega \in C_i^l = (u \in \tilde{N}_i^l) \cap \bigcap_{C \in \mathcal{C}_\pm^{l-1}} \left( \left( C \not\supseteq C_i^l \right) \cup \left( \omega \in C \right) \right), \tag{D-AND}$$

where we define the concepts of the previous layer and their opposites $\mathcal{C}_\pm^{l-1} = \{C_1^{l-1}, (C_1^{l-1})^c, ..., C_{n^{l-1}}^{l-1}, (C_{n^{l-1}}^{l-1})^c\}$. Equivalently, for an OR concept $C_i^l$, we would have

$$C_i^l = \left( \bigcup_{j \in \mathcal{S}_+} C_j^{l-1} \right) \cup \left( \bigcup_{j \in \mathcal{S}_-} (C_j^{l-1})^c \right) \cup \left( \mathcal{X} \times \underbrace{\bigcup_{z \in \tilde{\mathcal{S}}} \tilde{C}_z}_{\tilde{S}_i^l} \right),$$

where $\tilde{S}_i^l$ is the union of all its sufficient but unobserved concepts $\tilde{C}_z$, $\forall z \in \tilde{\mathcal{S}}$ and

$$\omega \in C_i^l = (u \in \tilde{S}_i^l) \cup \bigcup_{C \in \mathcal{C}_\pm^{l-1}} \left( \left( C \subseteq C_i^l \right) \cap \left( \omega \in C \right) \right). \tag{D-OR}$$

Modeling these necessary/sufficient unobserved concepts introduces *biases* $a_i^l \in [0, 1]$ for AND concepts and $o_i^l \in [0, 1]$ for OR concepts, defined as the conditional probabilities

$$a_i^l = \mathbb{P}\left[ u \in \tilde{N}_i^l \,\middle|\, \bigcap_{C \in \mathcal{C}_\pm^{l-1}} \left( \left( C \not\supseteq C_i^l \right) \cup \left( \omega \in C \right) \right) \right], \qquad o_i^l = \mathbb{P}\left[ u \in \tilde{S}_i^l \,\middle|\, \left( \bigcup_{C \in \mathcal{C}_\pm^{l-1}} \left( \left( C \subseteq C_i^l \right) \cap \left( \omega \in C \right) \right) \right)^c \right].$$

For an AND concept, $a_i^l$ is the probability that all the unobserved necessary concepts are present $u \in \tilde{N}_i^l$ when all the observed necessary concepts are present. This indicates how often this AND concept is indeed activated when the input $x$ suggests that it should. If $a_i^l = 0$, then this AND concept is never activated and it becomes useless in the modeling. For an OR concept, $o_i^l$ is the probability that any unobserved sufficient concept is present $u \in \tilde{S}_i^l$ when no observed sufficient concept is present. This indicates how often this OR concept is activated purely by unobserved concepts. If $o_j^l = 1$, then this OR concept is always trivially activated and it is also useless in the modeling. For both types of concepts, these probabilities are measures of how much relevant information we are missing in the input data $x$ to fully model this concept. Although

these unobserved concepts depend only on the unobserved data $u$, they are modeled with respect to $\omega$ since $u$ and $x$ are likely correlated in practice.

Our modeling contains two distinct types of probabilities, which represent different types of uncertainty. If we knew exactly how the concepts in our network were related logically, *i.e.* if we knew the exact structure of the ground-truth network with its exact weights $A_{i,j}^l$ and $O_{i,j}^l$, then the probabilities of presence of concepts $c_i^l(x)$, $a_i^l$ and $o_i^l$ would all be strictly *aleatoric* probabilities. They would only be statistical quantities that depend on the distribution of specific realizations $\omega = (x, u)$ that the network has seen in training. In contrast, the beliefs in the concepts' roles as necessary or sufficient to other concepts $A_{i,j}^l$ and $O_{i,j}^l$ are *epistemic* probabilities. They represent a priori *beliefs* in the general causal mechanisms that underlie the random phenomenon that generated $\omega$, and are independent of any such particular realization $\omega$. In practice, since $c_i^l(x)$, $a_i^l$ and $o_i^l$ are defined with respect to the believed roles, these probabilities model both aleatoric and epistemic uncertainty.

We obtain a tractable probabilistic modeling of the NLNs by making three assumptions of independence, to which we will return shortly. They allow the probabilities $c_i^l(x)$ to be easily computed in a parallelizable fashion (the full derivation is given in appendix B.1.2).

$$c_i^l(x) = a_i^l \prod_{j=1}^{n^{l-1}} \left(1 - [A_{i,j}^l]_+ \left(1 - c_j^{l-1}(x)\right)\right)\left(1 - [A_{i,j}^l]_- c_j^{l-1}(x)\right) \tag{P-AND}$$

$$c_i^l(x) = 1 - \left(1 - o_i^l\right) \prod_{j=1}^{n^{l-1}} \left(1 - [O_{i,j}^l]_+ c_j^{l-1}(x)\right)\left(1 - [O_{i,j}^l]_- \left(1 - c_j^{l-1}(x)\right)\right) \tag{P-OR}$$

The first two assumptions of independence that are required to obtain (P-AND) and (P-OR) are modeling choices, while the third one is an approximation. The first assumption is *independence between the presence of concepts in some $\omega$ and their general roles in the next layer as necessary or sufficient*

$$\omega \in C \perp\!\!\!\perp C' \supseteq C_i^{l+1}, \qquad \text{for all input concepts } C, C' \in \mathcal{C}_\pm^l \text{ of AND concept } C_i^{l+1},$$
$$\omega \in C \perp\!\!\!\perp C' \subseteq C_i^{l+1}, \qquad \text{for all input concepts } C, C' \in \mathcal{C}_\pm^l \text{ of OR concept } C_i^{l+1}.$$

We compute the probabilities $c_i^l(x) = \mathbb{P}\big[\omega \in C_i^l \,\big|\, x\big]$ using epistemic beliefs $A_{i',j'}^{l'}$, $O_{i',j'}^{l'}$ of the previous layers $l' \leq l$. The presence of a concept is thus certainly not independent of the roles that form its own definition in the previous layers. However, this assumption of independence is between the presence of concepts in a layer $l$ and their roles in the next layer $l + 1$. In other words, we assume that the presence of concepts in a particular realization $\omega$ does not give any information regarding their general roles in higher-level concepts, and vice versa. This independence would not hold if we were conditioning on the presence of the concepts in the next layer, *e.g.*

$$\omega \in C \not\perp\!\!\!\perp C \supseteq C_i^{l+1} \;\Big|\; \omega \in C_i^{l+1} \;.$$

If, for instance, we have for some $\omega$ that $C$ is absent but $C_i^l$ is present, then it is impossible that $C$ could ever be a necessary concept of $C_i^l$. Without conditioning however, it is conceivable that knowing whether we have $\omega \in C$ for some $\omega$ gives by itself no information on its general roles in the next layer, or on the roles of other concepts in the same layer.

The second assumption is *independence between the necessary/sufficient concepts of a concept*

$$C \supseteq C_i^{l+1} \perp\!\!\!\perp C' \supseteq C_i^{l+1}, \quad \text{for all input concepts } C \in \mathcal{C}_\pm^l \text{ and } C' \in \mathcal{C}_\pm^l \backslash \{C, \neg C\} \text{ of AND concept } C_i^{l+1},$$
$$C \subseteq C_i^{l+1} \perp\!\!\!\perp C' \subseteq C_i^{l+1}, \quad \text{for all input concepts } C \in \mathcal{C}_\pm^l \text{ and } C' \in \mathcal{C}_\pm^l \backslash \{C, \neg C\} \text{ of OR concept } C_i^{l+1}.$$

This assumption would be false for an observer who has previous knowledge and understanding about the concepts he is manipulating. For instance, if an AND concept already has `green` as a necessary concept, then it could not also have `red` or another incompatible color as necessary concepts. Moreover, an AND concept that already has `green` and `ball` would also be more likely to have `presence_of_a_tennis_racket`.

Although this assumption does not hold for an observer with previous understanding, it would hold for an observer who has absolutely no idea or previous understanding of what concepts he is manipulating, like an agent in an alien environment or the operator in Searle's "Chinese room" thought experiment (Searle, 1980). However, for an observer with previous knowledge, this assumption would amount to a cognitive bias of "total open-mindedness" that considers every combination of concepts to be equiprobable, irrespective of how related or incongruous they might be. We summarize the probabilistic modeling up to this point, including the first two assumptions of independence, in a PGM presented in appendix B.1.1.

The third assumption which we use as an approximation is *conditional independence between concepts in the same layer, given the input*

$$\omega \in C \perp\!\!\!\perp \omega \in C' \mid x, \qquad \text{for all concepts in the same layer } C, C' \in \mathcal{C}_+^l \text{ such that } C \neq C',$$

where we defined the concepts of a layer $\mathcal{C}_+^l = \{C_1^l, ..., C_{n^l}^l\}$. This assumption is the least realistic being false in most cases. It is trivially satisfied by the input features $x \in C_i^0$ since they only depend on $x$. However, for the other concepts defined in layers $l \geq 1$, this assumption is only true if no two concepts share a common necessary/sufficient concept in the previous layer, a property known as decomposability in the probabilistic circuits literature (Choi et al., 2021). This is very unlikely for non trivial NLNs, and it becomes progressively less likely as $L$ increases. Although this assumption is incorrect in most practical cases, avoiding it results in a combinatorial explosion of computations for the forward pass alone (see appendix B.1.3 for the full derivation without it). As a result, the method would become intractable and unusable in practice without this assumption. As a purely probabilistic model, this issue is catastrophic for NLNs. However, as a ML method, this approximation can be justified. First of all, in the case where all inputs are binary (*i.e.* with binary or categorical features), where all the weights are given from the set $\{-1, 0, 1\}$ and where the biases are full with $a_i^l = 1$, $o^{l'} = 0$, then this assumption is not necessary. In fact, none of the assumptions are necessary in that case because the probabilistic formulation (P-AND) and (P-OR) coincide exactly with the logical definition of the AND and OR nodes (see section 2.2). In general, we always need the weights to be integers for maximum interpretability and, as such, we quantize them during post-processing when learning a NLN (see section 3.2.2). Moreover, for any network that contains continuous features, with enough pre-processing nodes (see section 3.1.2), the signal can become arbitrarily close to binary thanks to appropriately scaled sigmoid curves. In practice, we only relax this binary constraint through the possible non-binary biases, as well as by limiting the number of pre-processing nodes for continuous features, so that values away from 0 and 1 can be obtained close to the learned boundaries (see section 3.1.2). Therefore, we keep this approximation leading to (P-AND) and (P-OR) and, in practice, this modeling is still able to obtain promising predictive performance and interpretable rule discovery, even in non-binary cases.

## 2.2 Fuzzy logic equivalency

A fuzzy logic is a real-valued generalization of classical logic, where instead of a statement like $x \in Y_k$ being either *true* or *false*, it can have a *degree of truth* between 0 (false) and 1 (true). The different fuzzy logics differ by their choices of how to generalize the classical logic operators

$$\wedge: \text{ AND}, \qquad \vee: \text{ OR}, \qquad \neg: \text{ NOT},$$

*etc.* For instance, product fuzzy logic is the fuzzy logic that coincides with probability distributions in which all random events are independent of each other. They thus define their fuzzy logic operators as

$$\overset{\text{P}}{\bigwedge_i} v_i := \prod_i v_i, \qquad \overset{\text{P}}{\bigvee_i} v_i := 1 - \prod_i (1 - v_i), \qquad \overset{\text{P}}{\neg} v := 1 - v,$$

by using the product t-norm ($\overset{\text{P}}{\wedge}$), the probabilistic sum t-conorm ($\overset{\text{P}}{\vee}$), and the strong negation ($\overset{\text{P}}{\neg}$) (van Krieken et al., 2022).

We can rewrite (P-AND) and (P-OR) with the product fuzzy logic operators

$$c_i^l(x) = a_i^l \overset{P}{\wedge} \bigwedge_{j \in \{1,\dots,n^{l-1}\}}^{P} \left( \overset{P}{\neg} [A_{i,j}^l]_+ \overset{P}{\vee} c_j^{l\text{-}1}(x) \right) \overset{P}{\wedge} \left( \overset{P}{\neg} [A_{i,j}^l]_- \overset{P}{\vee} \overset{P}{\neg} c_j^{l\text{-}1}(x) \right), \tag{F-AND}$$

$$c_i^l(x) = o_i^l \overset{P}{\vee} \bigvee_{j \in \{1,\dots,n^{l-1}\}}^{P} \left( [O_{i,j}^l]_+ \overset{P}{\wedge} c_j^{l\text{-}1}(x) \right) \overset{P}{\vee} \left( [O_{i,j}^l]_- \overset{P}{\wedge} \overset{P}{\neg} c_j^{l\text{-}1}(x) \right), \tag{F-OR}$$

By doing so, we obtain the product fuzzy logic generalization of the logical definition of our operators

$$\omega \in C_i^l = (u \in \tilde{N}_i^l) \wedge \bigwedge_{C \in \mathcal{C}_\pm^{l-1}} \left( \neg \left( C \supseteq C_i^l \right) \vee \left( \omega \in C \right) \right), \tag{L-AND}$$

$$\omega \in C_i^l = (u \in \tilde{S}_i^l) \vee \bigvee_{C \in \mathcal{C}_\pm^{l-1}} \left( \left( C \subseteq C_i^l \right) \wedge \left( \omega \in C \right) \right). \tag{L-OR}$$

This logical definition is equivalent to our previous probabilistic definition (D-AND) and (D-OR). Only the symbols have changed from one formalism to another. The intersections $\cap$ of random events have become conjunctions $\wedge$ (AND) of truth values, the unions $\cup$ have become disjunctions $\vee$ (OR) and the complement $\cdot^c$ implied in $\left( C \not\supseteq C_i^l \right) = \left( C \supseteq C_i^l \right)^c$ has become negation $\neg$ (NOT). It is interesting to note that we could rewrite (F-AND) and (F-OR) with a different fuzzy logic with differentiable t-norm and t-conorm such as Łukasiewicz logic and we would still obtain a learnable logic network. Doing so would keep the underlying logical modeling of our method with (L-AND) and (L-OR), but we would lose its probabilistic interpretation.

Previous formalizations of NLNs omitted any probabilistic or logical modeling of the AND and OR concepts and instead directly used fuzzy logic formulations like (F-AND) and (F-OR) (Payani & Fekri, 2019a; 2020; Wang et al., 2020; 2021; 2024; Zhang et al., 2023). They obtained similar formulations to our own by designing them according to a desiderata of the expected behavior of AND/OR nodes. However, their formulations were more restricted, being special cases of our own. They did not consider negated concepts in their formalism, *i.e.* they assumed $A_{i,j}^l, O_{i,j}^l \geq 0$. To consider negated concepts in practice, they would instead duplicate the inputs as hardcoded negated versions of themselves, hence also doubling the number of weights. Moreover, none of the previous formalizations considered the effect of unobserved concepts $\tilde{C}_z$ and data $u$. They assumed no unobserved necessary or sufficient concepts with $a_i^l = 1$ and $o_j^l = 0$. In addition to considering unobserved concepts, our probabilistic modeling also avoids having to assume independence between these unobserved concepts and the rest of the definitions. It also avoids assuming independence between a concept's causal role and its contrary's causal role. These assumptions would have been implied by a strictly product fuzzy logic modeling, such as in previous formalizations.

## 2.3 Logical perspective

AND, OR and NOT are fundamental operators in classical logic and some of their properties are relevant to our approach. Firstly, just like intersection, union and complement, we have that conjunction (AND), disjunction (OR) and negation (NOT) are related by De Morgan's laws and distributivity. Our formalism is compatible with De Morgan's laws, as can be seen from (P-AND)-(P-OR), (F-AND)-(F-OR), (L-AND)-(L-OR) and (D-AND)-(D-OR) (proofs in appendix B.2.1). As a consequence, any AND concept can be converted to an OR concept, and vice versa, by flipping the signs of its incoming and outgoing weights and by taking the complement of its bias. However, since product fuzzy logic operators are not idempotent (van Krieken et al., 2022)

$$a \overset{P}{\wedge} a = a \cdot a \neq a, \qquad\qquad a \overset{P}{\vee} a = 1 - (1 - a) \cdot (1 - a) \neq a,$$

distributivity can not be applied in our formalism, unless all probabilities $c_i^l(x)$, $|A_{i,j}^l|$, $|O_{i,j}^l|$, $a_i^l$, $o_i^l$ are binary.

Classical logic also studies logical formulas, which combine AND, OR and NOT operations. Using De Morgan's laws and distributivity, complex logical formulas can be rewritten in many equivalent formulations.

Two notable forms are the Disjunctive Normal Form (DNF), a disjunction (OR) of conjunctions (AND) with negation, *e.g.*

$$(A_1 \wedge \neg A_2 \wedge A_7) \vee (A_3) \vee (A_2 \wedge A_4 \wedge \neg A_5), \tag{DNF}$$

and the Conjunctive Normal Form (CNF), a conjunction (AND) of disjunctions (OR) with negation, *e.g.*

$$(A_1 \vee A_2 \vee A_3) \wedge (\neg A_2 \vee A_3 \vee A_4) \wedge (\neg A_2 \vee A_3 \vee \neg A_5) \wedge (A_2 \vee A_3 \vee A_7), \tag{CNF}$$

equivalent to the (DNF) above. As this example shows, depending on the logical formula, one form might be much simpler than the other. In our setting, we want to describe when a target concept is present $\omega \in Y_k$ with a logical formula that combines the input features $x \in C_i^0$ and unobserved concepts $u \in \tilde{N}_i^l$, $u \in \tilde{S}_i^l$. In this setting, the DNF has a special interpretation as a (normal) logic program, which is a finite set of rules of the form

$$B_1 \wedge ... \wedge B_m \wedge \neg B_{m+1} \wedge ... \wedge \neg B_n \rightarrow H$$

where $H$ is called the head and the left-hand side is called the body. The usual notation is written from right to left, but we adopt the opposite convention here to coincide with the graphical representation of neural networks. Such a rule says that if the condition in the body is true, then the head $H$ is true. For instance, if we say that $B$ is equal to (DNF), then it can be described by the following logic program

$$A_1 \wedge \neg A_2 \wedge A_7 \rightarrow B,$$
$$A_3 \rightarrow B,$$
$$A_2 \wedge A_4 \wedge \neg A_5 \rightarrow B.$$

If any of these 3 rules is activated by the left-hand side AND combination being true, then $B$ is true, and otherwise it is false. The OR is implied by the fact that any activated rule is sufficient for $B$ to be true. In Section 3.1, we will encode this logic program formulation into the structure of our NLNs in order to be able to interpret what a NLN has learned as a set of rules.

## 2.4 Interpretation

Although a NLN attempts to learn the underlying causal mechanism relating the inputs to the outputs, it cannot uniquely determine the causal structure or the direction of causality. An AND concept or an OR concept can each represent many different cases. Some possible interpretations are given in Table 1 with toy examples pictured in Figure 2.

Table 1: Some possible interpretations of the AND and OR concepts

| AND | | | OR | | |
|---|---|---|---|---|---|
| Necessary concepts | of an | AND concept | Sufficient concepts | of an | OR concept |
| necessary components | of a | situation | possible cases | of an | equivalency class |
| necessary causal ingredients | producing a | consequence | possible causes | of a | consequence |
| necessary consequences | of a | cause | possible consequences | of a | causal ingredient |
| necessary parent concepts | of a | sub-concept | possible sub-concepts | of a | parent concept |

Moreover, since a finite combination of AND (resp. OR) concepts can be represented by a single AND (resp. OR) concept, each type can represent an infinite number of cases. We illustrate some intuitive and counter-intuitive examples of such causal structures in appendix B.3.

# 3 Machine Learning pipeline

## 3.1 Interpretable structure

By interpretability, we mean the ability to provide its meaning in human-understandable terms (Arrieta et al., 2020). In this sense, the AND and OR concepts that we have defined are interpretable so long as their inputs

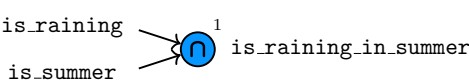

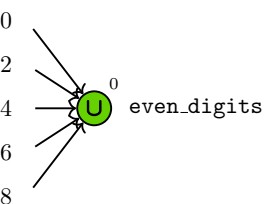

(a) Necessary components of a situation, where 1 indicates that all necessary components are observed

(b) Possible cases of an equivalency class, where 0 indicates that no other case is missing or unobserved

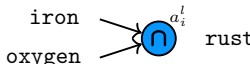

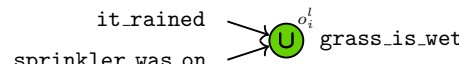

(c) Necessary causal ingredients producing a consequence, where $a_i^l$ represents the probability of other missing ingredients, *e.g.* water or humidity

(d) Possible causes of a consequence, where $o_i^l$ represents the probability of other missing causes, *e.g.* a bucket of water was dropped

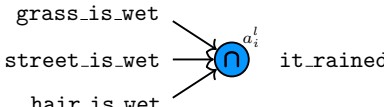

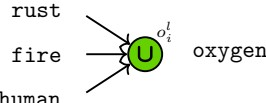

(e) Necessary consequences of a cause, where $a_i^l$ represents the probability that the cause is present when all the consequences are present, *i.e.* that the consequences are not explained by other causes

(f) Possible consequences of a causal ingredient, where $o_i^l$ represents the probability that the causal ingredient is still present when none of these consequences are observed

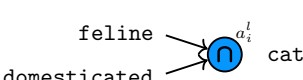

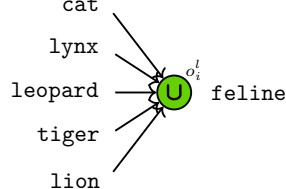

(g) Necessary parent concepts of a sub-concept, where $a_i^l$ represents the probability of the other missing parent concepts of the sub-concept, *e.g.* having partially webbed feet as opposed to the leopard cat, another domesticated feline which has fully webbed feet

(h) Possible sub-concepts of a parent concept, where $o_i^l$ represents the probability of another missing sub-concept, *e.g.* a cougar or a panther, etc.

Figure 2: Toy examples of interpretations for AND and OR concepts

are themselves interpretable. To ensure that the full NLN model is interpretable, we can impose inductive biases in its structure to ensure that each AND/OR concept learns a meaningful concept. We propose the structure pictured in Figure 3. It contains two fully-connected layers arranged in DNF, *i.e.* an AND layer with negation followed by an OR layer without negation, in order to learn a logic program for each target (section 3.1.1). The input features that are not binary are pre-processed with appropriate input modules, one for categorical features, and another for continuous features (section 3.1.2).

### 3.1.1 Fully-connected DNF layers

NLNs, as we have defined them, could be constructed with arbitrary depth. However, deep NLNs pose challenges regarding learning and intepretability. NLNs in general are difficult to learn because of vanishing gradients (Payani & Fekri, 2019a; Wang et al., 2020; 2021; 2024; van Krieken et al., 2022) and depth exacerbates this issue. Deeper networks are also harder to interpret. The first fully-connected layer of

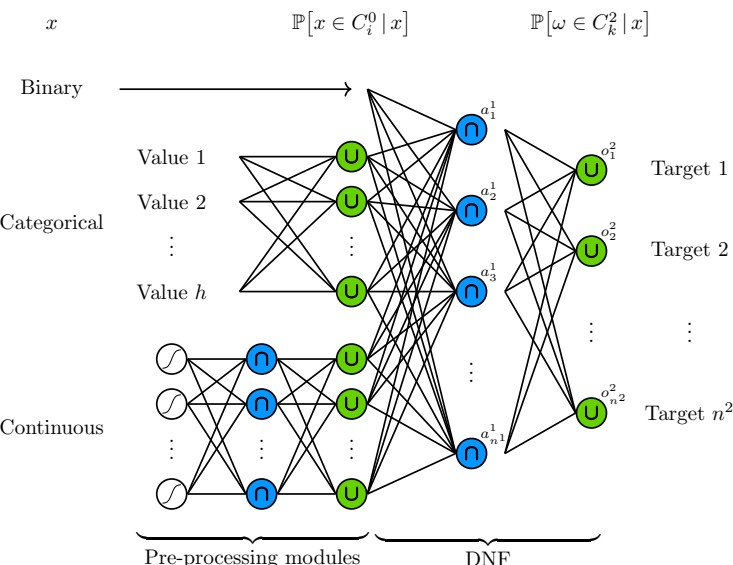

Figure 3: Structure of an interpretable NLN

concepts can be easily interpreted because it is directly defined in terms of the input features, which are usually interpretable. However, the next layers are defined by combining these higher-level concepts and their resulting definitions become increasingly harder to interpret. Their definitions in terms of the input features are not only more indirect, their activation patterns with respect to those features are also much more complex.

To avoid these issues, we restrict ourselves to two fully connected layers arranged in DNF. The hidden layer is made up of AND concepts that allow negation and the output layer is made up of OR concepts that do not allow negation. As mentioned previously, this DNF can learn any logical formula and can be interpreted as a logic program. Each target output concept (OR) is implied by any one of multiple rules (AND), which are each activated by a specific combination of values from the input features.

### 3.1.2 Input pre-processing modules

Our previous modeling assumed that the input concepts $x \in C_i^0$ are binary or represent probabilities of random events $c_i^0(x) = \mathbb{P}\left[x \in C_i^0 \mid x\right]$. Hence, *binary* input features can directly be used as input concepts, but *categorical* (one value out of a finite set of possible values) or *continuous* (in a subset of $\mathbb{R}$) features must be pre-processed. We use a different pre-processing module for each to convert them into interpretable probabilities.

**Categorical features**  Categorical features can be directly converted to binary variables with a one-hot encoding. However, feeding these one-hot encodings to the fully-connected DNF layers would needlessly multiply the number of rules in a model whenever multiple values of a category behave the same way in some circumstance. We instead introduce their own layer of OR concepts without negation and without unobserved sufficient conditions (since we can observe every possible value). These OR concepts learn equivalency classes of categorical values that have the same effect. In addition to reducing the number of duplicate rules for each related value, this encoding is also interpretable and results in a limited form of *predicate invention*.

**Continuous features**  Features that are continuous need to be discretized in order to be manipulated by AND and OR concepts. Wang et al. (2021) introduced the idea of learning upper and lower bounds for each continuous feature, noting that in the following layers these bounds could be combined into intervals (AND) and then arbitrary collections of such intervals (OR). Since our framework can take advantage of negation, only upper bounds $\mathcal{B}_{i,k} \in \mathbb{R}$ are needed, and we additionally learn a sharpness parameter $\alpha_{i,k}$ that

controls how sharp or fuzzy is the transition at the boundary, resulting in a fuzzy discretization. We call these concepts *fuzzy dichotomies*, defined for a continuous feature $x_k$ by

$$\sigma\Big(\alpha_{i,k}(x_k - \mathcal{B}_{i,k})\Big),$$

where $\sigma(\cdot)$ is the sigmoid function. For each continuous feature, we use a number of these fuzzy dichotomies which are fed to their own DNF (without unobserved concepts), in order to learn arbitrary collections of fuzzy intervals. For instance, in a task that uses a continuous feature $x_k$ representing `weight`, one rule might hold only for very light objects or somewhat heavy objects such as $x_k \in [0, 0.1] \cup [10, 15]$ which would be learned as $(x_k < 0.1) \cup \big((x_k > 10) \cap (x_k < 15)\big)$. The resulting collections of fuzzy intervals are then used as input to the fully-connected DNF (with unobserved concepts) that can learn the final rules with all the features. The fuzzy dichotomies are learned conjointly with all the AND/OR concepts in the NLN.

### 3.1.3 Input encodings and rule modules

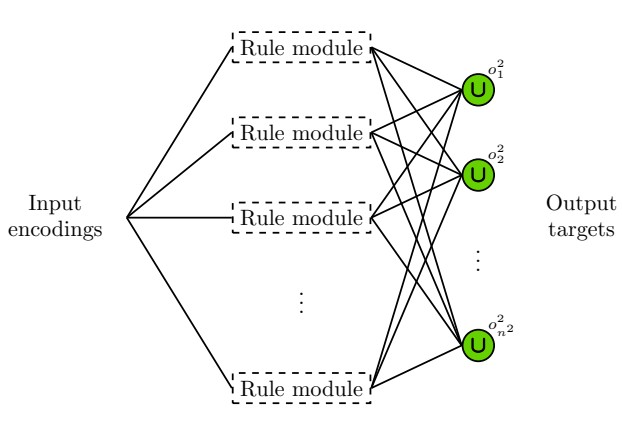

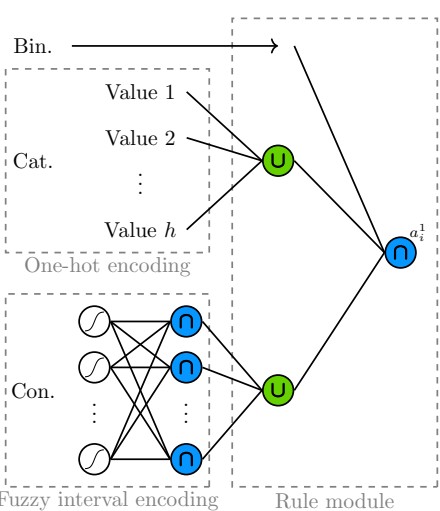

(a) NLN with rule modules and input encoding

(b) Rule module and input encodings

Figure 4: NLN structure used for learning

To help the learning process, we do not learn the NLN directly with the structure in Figure 3. We instead disentangle the learning of each AND rule by introducing separate rule modules and shared input encodings that are used by all rule modules, pictured in Figures 4(a) and 4(b). Each rule module contains a single AND rule that takes its inputs from (1) the binary features, (2) its own OR equivalency classes, one for each categorical feature, and (3) its own OR collections of fuzzy intervals, one for each continuous feature. In turn, binary features are used directly, but categorical features have a shared one-hot encoding, and continuous features are encoded with shared fuzzy dichotomies and AND fuzzy intervals. This factorization of the structure allows each rule to learn more independently of the others and reduces the number of parameters in the model. However, the fuzzy interval encodings of the continuous features are still learned conjointly for the whole NLN. For categorical features, this factorization is equivalent in terms of representability. Since the one-hot encoding is binary, an AND rule that combines multiple OR equivalency classes for the same categorical feature can always be rewritten with a single OR equivalency class when the weights are integers. The same is not true for the fuzzy interval encoding of continuous features since they are never binary. However, in terms of interpretability, a single OR collection of fuzzy intervals is much simpler to interpret in an AND rule, which may already involve many other features.

### 3.2 Learning

Learning a NLN is done in two stages: (1) training, and (2) post-processing, which includes weight quantizing, continuous parameter retraining, and pruning.

### 3.2.1   Training

**Objective and regularization**   We use the ADAM optimizer (Kingma & Ba, 2015) to minimize the $L_2$ loss. Its minimizer is $\mathbb{P}[\omega \in Y_k \,|\, x]$ which is precisely what we want our NLN's outputs $c_k^2(x) = \mathbb{P}[\omega \in C_k^2 \,|\, x]$ to model. To help the learning process we regularize NLNs in two different ways. First, to combat the tendency of unadapted concepts to become trivial, we regularize the AND and OR concepts to have non-empty definitions. For instance, we consider the definition of an AND concept $C_i^l$ to be non-empty if

$$\sum_j |A_{i,j}^l| \geq 1,$$

in other words, we consider a concept $C_i^l$ to be non-empty if it attributes a probability mass of at least 1 across all of its possible input concepts $C_j^{l-1}$. We force non-empty definitions in all AND and OR concepts by penalizing

$$\mathcal{L}_{\text{non-empty}} = \sum_{\text{AND weights } A_{i,\cdot}^l} \left\| \left[ 1 - \sum_j |A_{i,j}^l| \right]_+ \right\|_2^2 + \sum_{\text{OR weights } O_{i,\cdot}^l} \left\| \left[ 1 - \sum_j |O_{i,j}^l| \right]_+ \right\|_2^2,$$

which is only active when a concept $C_i^l$'s definition attributes a probability mass less than 1. In that case, the penalty will increase the weights of all of its input concepts $C_j^{l-1}$ uniformly until a probability mass of at least 1 is attributed. Moreover, in order to encourage sparser, more interpretable solutions, we also penalize the $L_1$ norm of all weights in the network. The full loss function is then given by

$$\mathcal{L}(y, c^L(x)) = \left\| y - c^L(x) \right\|_2^2 + \lambda_{\text{non-empty}} \cdot \mathcal{L}_{\text{non-empty}} + \lambda_{\text{sparsity}} \left( \sum_{\text{AND weights } A_{i,\cdot}^l} \left\| A_{i,\cdot}^l \right\|_1 + \sum_{\text{OR weights } O_{i,\cdot}^l} \left\| O_{i,\cdot}^l \right\|_1 \right),$$

where $\lambda_{\text{non-empty}}, \lambda_{\text{sparsity}} > 0$ are the regularization coefficients of the non-empty penalty and the sparsity penalty respectively. We minimize the expectation of this loss

$$\mathbb{E}_{(X,Y) \sim \mathcal{D}} \left[ \mathcal{L}(Y, c^L(X)) \right]$$

over the training dataset $\mathcal{D}$, subject to the domain constraints of the weights $A_{i,j}^l \in [-1, 1]$, $O_{i',j'}^{l'} \in [0, 1]$, the biases $a_i^l, o_{i'}^{l'} \in [0, 1]$, and the parameters of the fuzzy dichotomies $\mathcal{B}_{i'',k} \in \mathbb{R}$, $\alpha_{i'',k} > 0$ for appropriate indices $(i, j, l)$, $(i', j', l')$ and $(i'', k)$ according to the NLN's structure.

**Initialization**   In our experimentations, the initialization of the NLN was a very important factor in its ability to learn. We have found the best combination to have (1) uniformly random weights, (2) fully observed concepts, and (3) regularly distributed fuzzy interval encodings for continuous features, pictured in Figure 5. The random weights increase our chances to find potential rules that can be further massaged towards relevant rules, with respect to the target concepts. We begin with full binary biases $a_i^1 = 1$ and $o_j^2 = 0$, *i.e.* without unobserved effects. These ensure that the initial gradients are as strong as possible since, in general, their magnitudes are proportional to $a_i^l$ for AND nodes and to $(1 - o_i^l)$ for OR nodes. This is especially important to combat the effect of the vanishing gradients. For binary features and categorical features with one-hot encoding, the random weights can be learned because they receive clean 0-1 signal that is also interpretable. In order to obtain a similarly clean and interpretable signal from continuous features, we initialize the fuzzy interval encoding to regularly distributed intervals with appropriately scaled sharpness. This way, an input $x$ will initially only activate a single fuzzy interval per continuous feature, hence producing a clean, interpretable signal.

**Rule reset**   In practice, we have observed that rules that are not helpful in the model quickly become "dead" concepts. For such an AND rule $C_i^1$, either its bias $a_i^1$ goes to 0 or their weights $O_{\cdot,i}^2$ in the next layer all go to 0. In both cases, they stop receiving signal in the back-propagation and stop learning. We solve this issue by re-initializing dead rule modules. To do so, we randomly re-initialize their weights $A_{i,\cdot}^1 \sim \mathrm{U}(-1, 1)$ and

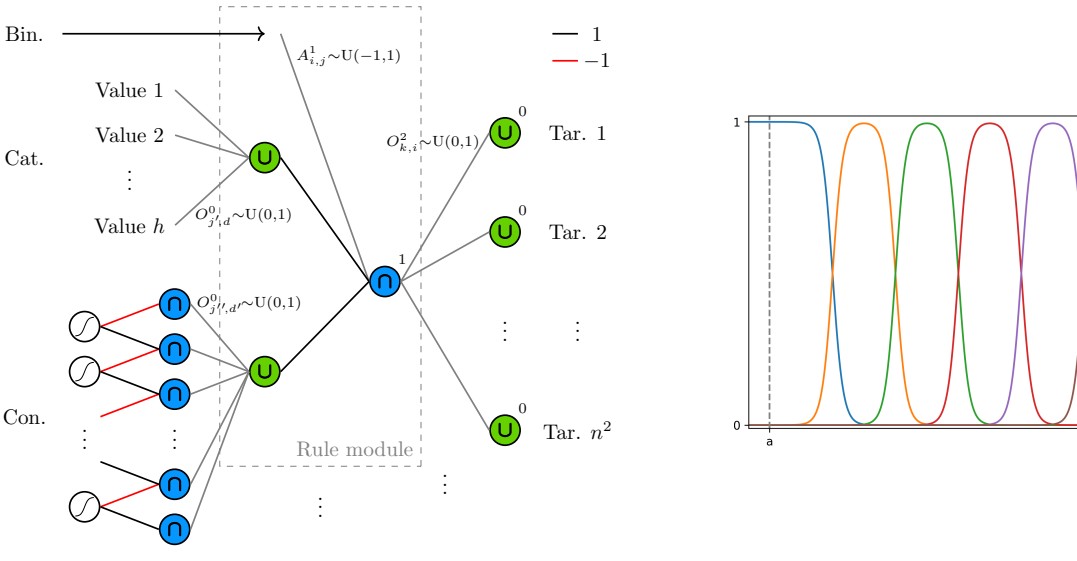

(a) Rule module and input encoding initialization    (b) Fuzzy interval encoding initialization

Figure 5: NLN initialization

their bias to $a_i^1 = 1$. To ensure that the resets do not affect the loss, we also set their outgoing weights in the output layer to $O_{\cdot,i}^2 = 0$. We do this at the end of every epoch by checking for dead rule modules and re-initializing them. By having a big number of rule modules in the network, we can try many random rules at each epoch and keep only those that have potential to be learned further.

### 3.2.2 Post-processing

**Weight quantizing**    At this point, the NLN has been learned but its weights are still probabilities which are hard to interpret, especially in conjunction with one another. For instance, a simple AND rule that would have weights of $(0.80, -0.65, 0.15)$ for respectively `ball`, `green` and `heavy` is difficult to interpret. It represents a concept that is likely a type of ball, but not necessarily; a concept that is probably not a green object; and a concept that might be a heavy object although it is unlikely; all simultaneously. This is not easily interpretable, unlike the same weights after *quantizing* which might be $(1, -1, 0)$ that would represent the concept of a `ball_that_is_not_green`. By quantizing the weights to values of either 0, 1 or $-1$, we obtain instantly understandable concepts that still retain a probabilistic bias in $[0, 1]$, indicating if we are missing other unobserved concepts in its definition and how often they appear. To quantize, previous methods would either threshold the weights at above or below 0.5 (Wang et al., 2020; Zhang et al., 2023) or use a modified learning algorithm to learn the quantized weights directly (Wang et al., 2021; 2024). We have instead developed 4 simple greedy algorithms to quantize the weights one at a time. The weights are quantized according to their effect on the loss of the full training set (including the validation set). Our experimentations suggest that our most effective and reliable approach is Algorithm 1. Our other quantizing algorithms are presented in appendix C.1.1.

**Continuous parameter retraining**    Once the weights are quantized, we retrain briefly the continuous parameters of the model with respect to the new weights. These continuous parameters are the model's parameters which are interpretable for continuous values, *i.e.* the biases $a_{i,j}^l, o_{i,j}^l \in [0, 1]$ as well as the boundaries in $\mathcal{B}_{i,k} \in \mathbb{R}$ and sharpnesses in $\alpha_{i,k} \in \mathbb{R}_+^*$ of the fuzzy dichotomies used to pre-process the continuous features $x_k$.

**Pruning**    Finally, inspired by Payani & Fekri (2019b), we prune unnecessary weights and simplify the NLN accordingly. To do a pruning pass, starting from the output layer, we consider pruning each weight one at a time and, if it improves or does not affect the loss on the full training set (including the validation set), we

---

**For** each layer $l \in \{1, .., L\}$, starting from the last layer $L$,
    **For** each weight $A_{i,j}^l$ (resp. $O_{i,j}^l$), in decreasing likeliness $\left|A_{i,j}^l\right|$ (resp. $\left|O_{i,j}^l\right|$),
        **If** it is non-zero,
            Compare the loss when we fix $A_{i,j}^l \in \left\{0, \text{sign}\left(A_{i,j}^l\right)\right\}$ (resp. $O_{i,j}^l \in \left\{0, \text{sign}\left(O_{i,j}^l\right)\right\}$).
            Commit to the best quantized value.

Do the same for the category and continuous input modules, one at a time.

---

Algorithm 1: Descending selection quantizing algorithm

permanently prune the weight and, otherwise, we restore its previous value. We keep doing pruning passes until the NLN stops changing.

## 4 Experiments

We evaluate our approach on two different tasks: discovery of boolean networks, and tabular classification, *i.e.* classification from structured data. In both cases we use a NLN with 128 rule modules with regularization coefficients of $\lambda_{\text{non-empty}} = 10^{-1}$ for the non-empty definitions regularization and of $\lambda_{\text{sparsity}} = 10^{-3}$ for the sparsity regularization on the $L_1$ norm of the weights. In cases of multi-class classification, the target class with the highest probability is outputted. In binary classification, the threshold with the highest score is used for the binary output (grid search with 0.01 step). We implement our method in Pytorch (Paszke et al., 2019) using the ADAM optimizer.

### 4.1 Boolean networks discovery

Boolean networks were introduced in Kauffman (1969) to model gene regulatory networks in biology. A boolean network models gene interactions where $n$ genes at a time step $t$ are either activated $A_i^t = 1$ or not $A_i^t = 0$. Given the activations $A_i^t$ at step $t$, their activations $A_i^{t+1}$ at the next time step $t + 1$ are deterministically given by a logic program where each gene is activated if one of its rules are satisfied in the previous time step. For instance, we might have for gene 3 a simple logic program with only two rules

$$\neg A_4^t \to A_3^{t+1},$$
$$A_1^t \wedge \neg A_2^t \to A_3^{t+1},$$

*i.e.* gene 3 is activated at time step $t + 1$ if at time step $t$, either gene 4 was not activated or gene 1 was activated and gene 2 was not, or both. For this task, we are provided a dataset with all possible gene state transitions $(A_1^t, ..., A_n^t, A_1^{t+1}, ..., A_n^{t+1})$ and we want to discover the ground-truth logic program by learning to predict the end state $(A_i^{t+1})_i$ from the start state $(A_i^t)_i$. Since we almost never have access to this full distribution, we study the performance of our algorithm for partial datasets with ratios of the full dataset ranging from 10% to 100%. We consider four datasets with known ground-truth logic programs: mammalian cell cycle regulation (Fauré et al., 2006), fission yeast cell cycle regulation (Davidich & Bornholdt, 2008), budding yeast cell cycle regulation (Li et al., 2004) and arabidopsis thaliana flower morphogenesis (Chaos et al., 2006). We evaluate the performance of our algorithm according to its accuracy in two repeats of Five-Fold Cross-Validation (5F-CV).

For this task, we do not split the training set into training and validation sets. Since there is a ground-truth logic program in this task, we have observed in our experiments that NLNs do not tend to overfit and every additional data point is relevant to find all the correct rules. We compare our approach with four other methods that were used for this task. Two of them are neuro-symbolic in nature while the other two are purely symbolic. The first neuro-symbolic method is NN-LFIT (Tourret et al., 2017) which learns a neural network that is then approximated by a logic program. The second neuro-symbolic method is D-LFIT (Gao et al., 2022) which learns a logic program that is embedded in a set of matrices in a novel neural network structure. The purely symbolic methods are the Inductive Logic Programming method LF1T (Inoue et al.,

2014) and the symbolic rule learner JRip (Witten et al., 2017). The results are presented in Table 2, which is partially reproduced from Gao et al. (2022).

Table 2: Comparison of five-fold cross-validation accuracy (%) on partial datasets with different split rates

| Datasets (variables, rules) | Model | Ratio of the full distribution | | | |
|---|---|---|---|---|---|
| | | 8% | 16% | 40% | 80% |
| Mammalian (10, 23) | **NLN** | 94.41 | **98.90** | **100** | **100** |
| | NN-LFIT | **96.60** | 94.35 | 99.89 | 99.91 |
| | D-LFIT | 71.67 | 75.9 | 80.09 | 82.84 |
| | LF1T | 76.01 | 76.48 | 76.73 | 91.56 |
| | JRip | 77.84 | 75.44 | 76.41 | 74.66 |
| Fission (10, 24) | **NLN** | 94.60 | 98.51 | **100** | **100** |
| | NN-LFIT | **98.80** | **99.80** | 99.92 | 99.87 |
| | D-LFIT | 80.45 | 85.33 | 93.13 | 92.89 |
| | LF1T | 76.85 | 77.03 | 77.15 | 100 |
| | JRip | 79.14 | 78.05 | 80.04 | 78.47 |
| Budding (12, 54) | **NLN** | **97.42** | **99.65** | **100** | **100** |
| | NN-LFIT | *ROT* | *ROT* | *ROT* | *ROT* |
| | D-LFIT | 71.96 | 71.39 | 70.50 | 76.52 |
| | LF1T | *ROT* | *ROT* | *ROT* | *ROT* |
| | JRip | 67.97 | 68.55 | 67.91 | 68.35 |
| Arabidopsis (15, 28) | **NLN** | **100** | **100** | **100** | **100** |
| | NN-LFIT | *ROT* | *ROT* | *ROT* | *ROT* |
| | D-LFIT | 84.35 | 86.83 | 88.56 | 89.70 |
| | LF1T | *ROT* | *ROT* | *ROT* | *ROT* |
| | JRip | 68.84 | 69.00 | 68.79 | 68.67 |

In all four datasets, our method achieves more than 98 % accuracy with as little as 16 % of the data. Moreover, it achieves perfect accuracy with only 40 % of the data. In doing so, our method also discovers the ground-truth boolean networks by correctly identifying all of their necessary rules. In some cases, some of the ground-truth rules are subsumed by the disjunction of other ground-truth rules, making them redundant. In appendix D.1.1, we present examples of correctly discovered rules as well as incorrectly discovered rules when there is insufficient data. In contrast, the other methods never achieve perfect accuracy even with 80% of the data. The biggest difficulty for these other methods seem to be the number of rules with the poorest performance being on the budding dataset, which has the most rules. This is not an issue for our approach which, being a learning approach, is instead mostly concerned with the amount of data. It performs most poorly on datasets with lower dimensionality for which the same fraction of the full distribution represents a much smaller amount of data. For the arabidopsis dataset, which has the largest solution space with 15 variables, our model is able to discover the ground-truth logic program with as little as 8 % of the full distribution. This ground-truth rule discovery from small amounts of data is one of the strengths of neuro-symbolic methods as opposed to purely neural methods. On the other hand, the purely symbolic extraction step of NN-LFIT and the ILP LF1T method both Run Out of Time (ROT) (5 hours for this task) on the larger solution spaces of the budding and arabidopsis datasets.

## 4.2 Tabular data classification

We test the more general case of classification from tabular data on 6 classical UCI datasets, often used to test model interpretability. We add a seventh dataset by converting the continuous features of the balance dataset to categorical features, since they only take 5 possible values. These represent two distinct types of datasets: those that can be represented by a ground-truth logic program (chess KRKPA7 (Shapiro, 1983), monk2 (Wnek, 1993) and tic-tac-toe (Aha, 1991)), and those that cannot (adult (Becker & Kohavi, 1996), balance (Siegler, 1976), DARWIN (Cilia et al., 2018), and wine (Aeberhard & Forina, 1992)). However, balance does have an easily interpretable representation since it describes a balance scale which leans to one side or is in balance depending on which side has the highest product $weight \cdot distance$, but this cannot be encoded as a logic program without trivially encoding all 625 possible cases. The characteristics of these

datasets are presented in Table 3. Since these datasets are mostly unbalanced, we use the F1 score to evaluate the prediction performance of the models with five-fold cross-validation.

Table 3: Dataset characteristics and logic network model capacities (number of Pre-Processing nodes (PP), of Fully-Connected nodes (FC) and Total number of Parameters (TP))

| Datasets | Inputs | | | Outputs | Samples | NLN | | | RRL | | |
|---|---|---|---|---|---|---|---|---|---|---|---|
| | Bin. | Cat. | Con. | Classes | | PP | FC | TP | PP | FC | TP |
| adult | 1 | 7 | 6 | 2 | 32 561 | 2 054 | 128, 1 | 46 913 | 120 | 4096, 4096, 2 | 17 686 648 |
| balance | 0 | 0 | 4 | 3 | 625 | 772 | 128, 3 | 22 403 | 80 | 1024, 3 | 83 536 |
| balance (cat.) | 0 | 4 | 0 | 3 | 625 | 512 | 128, 3 | 3 587 | 0 | 1024, 3 | 22 016 |
| chess | 35 | 1 | 0 | 2 | 3 196 | 128 | 128, 1 | 5 249 | 0 | 1024, 2 | 39 936 |
| DARWIN | 0 | 0 | 450 | 2 | 174 | 86 850 | 128, 1 | 2 462 657 | 9 000 | 1024, 2 | 9 226 024 |
| monk2 | 0 | 6 | 0 | 2 | 432 | 768 | 128, 1 | 3 201 | 0 | 1024, 2 | 18 432 |
| tic-tac-toe | 0 | 9 | 0 | 2 | 958 | 1 152 | 128, 1 | 4 865 | 0 | 1024, 2 | 28 672 |
| wine | 0 | 0 | 13 | 3 | 178 | 2 509 | 128, 3 | 71 651 | 260 | 1024, 3 | 268 036 |

For this task, we do not expect in general that there are actual logic programs that can predict perfectly these datasets. As such, we opt to split the training set into training (80 %) and validation (20 %) sets to select the best model in early stopping, in order to avoid overfitting. In the post-processing phase, we use the full training set including the validation set, which is especially important for these datasets because of their limited size. We compare our approach with three similar models. The first is RRL (Wang et al., 2024) which uses the same modeling for the AND/OR nodes, with the exception of the missing bias and the need to double the weights to consider negated concepts (see section 2.2 for their special case). It uses an approximated version of this modeling however by introducing three hyper-parameters $(\alpha, \beta, \gamma)$ to reduce the vanishing gradients problem inherent to this approach. Its structure is also different with (1) no input pre-processing except for learnable upper and lower bounds for the continuous features, (2) between 1 and 4 logical layers which are each made up of half AND nodes and half OR nodes, and (3) its output layer is linear. It thus introduces many more hyper-parameters to tune than our approach which has none in practice, since it uses by default 32 fuzzy dichotomies with 33 fuzzy interval encodings per continuous feature. We follow their instructions to learn the models and tune their hyper-parameters with the final selected structures presented side-by-side with our own in Table 3. The final two models are the Optimal Decision Tree (ODT) and its generalization, the Optimal Decision Diagram (ODD) (Florio et al., 2023). These approaches use a Mixed Integer Programming formulation to find the optimal decision tree (resp. diagram) given a dataset and a graph topology. For both models, we follow their instructions for which topologies to test, and select the best performing one. We also add a final model that is not interpretable to show the level of prediction performance that is attainable for each dataset, XGBoost (XGB) (Chen & Guestrin, 2016; Grinsztajn et al., 2024). The results are presented in Table 4.

Table 4: Comparison of five-fold cross-validation f1-score (%)

| Datasets | NLN | RRL | ODD | ODT | XGB |
|---|---|---|---|---|---|
| adult | 66.03 | **80.20** | 63.65 | 63.52 | 71.62 |
| balance | 58.78 | **77.72** | 73.31 | 73.63 | 69.11 |
| balance (cat.) | 59.69 | 82.20 | **94.70** | 82.37 | 66.33 |
| chess | **99.58** | 99.43 | 96.78 | 95.31 | 99.55 |
| DARWIN | 77.22 | **86.01** | 70.87 | 78.95 | 88.84 |
| monk2 | 86.81 | **98.30** | 73.02 | 75.95 | 88.92 |
| tic-tac-toe | **100** | **100** | 96.69 | 97.07 | 100 |
| wine | 94.44 | **98.23** | 74.48 | 78.97 | 98.88 |

In the datasets with underlying ground-truth logic programs (chess, monk2 and tic-tac-toe), our NLN performs very well as expected. It achieves perfect prediction on tic-tac-toe, is nearly as good as the uninterpretable XGBoost on monk2, and it even narrowly beats the much larger RRL network on chess with almost 8 times fewer parameters. On the other datasets however (adult, balance, DARWIN, wine), with the exception of

wine on which we obtained a good performance, our method was not able to find enough relevant rules to predict the datasets accurately. On the other hand, the similar RRL with more nodes and better learning properties was able to achieve similar performance to XGBoost in all datasets, even improving upon its performance in 4 out of the 7 datasets. This suggests that our initialization and rule reset scheme for learning NLNs is not as optimal for finding relevant rules as the RRL's additional rule nodes and improved learning properties through additional hyper-parameters and linear output layer.

However, the main advantage of these methods is their interpretability and, by using much more nodes, the RRL runs the risk of losing this capacity. Figure 1 presents networks with perfect predictive accuracy on tic-tac-toe found by NLN and RRL (for the RRL, the smallest one in the five-fold cross-validation was selected). The network found by NLN is minimal and describes in a straight-forward manner the 8 possible rules that would make $\times$ win, *i.e.* when the $\times$s form any of the 3 rows, 3 columns and 2 diagonals. In contrast, the smallest network found by RRL requires 318 rule nodes, half of which are AND nodes and the other half are OR nodes. In fact, for the two datasets where the NLN performed as well or better than RRL, it did so with 10 rules or less while the RRL needed over 300 rules to achieve comparable results (see appendix D.2.1 for the full comparison of average number of rules and rule size on all datasets). This could be a consequence of the RRL's linear output layer. By imposing a rigid interpretable structure in NLNs, the learning is more difficult, but the final learned model is directly interpretable. In contrast, the RRL's output linear layer makes the interpretation of its learned rules much less straightforward. Since it produces hundreds of very small rules, many of them involving a single feature value, it seems that the actual decision rules learned by the RRL are contained in the distributed representation of its output linear layer that is much harder to interpret. Hence, even in cases where the NLN fails to learn a model with sufficient predictive performance, the relevant rules that it does find are at least interpretable (see appendix D.2.2 and D.2.3 for examples on the adult and DARWIN datasets). However, since our search algorithm is stochastic, many different models with equivalent predictive performance can be obtained during learning with differing sets of rules. For instance, in preliminary tests, a model with perfect accuracy on tic-tac-toe was found that discovered the 3 rows and 3 columns of $\times$, but instead of the 2 diagonals of $\times$, it discovered 2 equivalent rules that each say that, if a diagonal has no $\circ$ anywhere on it, then $\times$ wins. Such equivalent sets of rules can provide additional insight as in this case, although this variability can also be detrimental in other cases where different users may obtain different insights from the same data.

## 5 Conclusion

NLNs are a powerful learning and modeling tool for situations that can be described by logic programs, *i.e.* when the output classes/labels $Y_k$ can be described by a set of IF-THEN rules on the input $x$ of the form

$$\textbf{IF } x_1 \in A_{1,k} \text{ AND } ... \text{ AND } x_d \in A_{d,k}\textbf{, THEN } x \in Y_k.$$

By learning to predict the output classes/labels from the input, a NLN can discover this underlying causal structure. Such problems can arise often in practical settings. For instance, a NLN could easily be implemented in a Logical Analysis of Data (Lejeune et al., 2019) framework for industrial operations and maintenance.

The probabilistic modeling behind NLN's derivation of (P-AND) and (P-OR) supposes three independence assumptions. The first one is between aleatoric and epistemic quantities, *i.e.* specific realizations versus general roles as being necessary or sufficient to other concepts. The second assumption is equivalent to a cognitive bias of "total open-mindedness" that considers every combination of concepts to be equiprobable, irrespective of how related or incongruous they might be. The third assumption is an approximation that considers every concept in the same layer to be conditionally independent given the input.

We proposed a factorized rule module structure with pre-processing modules that ensure that the learned rules are easily interpretable to a domain expert. We also proposed a modified learning algorithm with a rule reset scheme to tackle the NLN's notorious vanishing gradient problem. However, in practice, this strategy does not seem to be sufficient to learn good predictive NLNs in general, especially in cases where there is no underlying ground-truth logic program. More work needs to be done on this issue to unlock the full modeling capabilities of NLNs. One possible direction is the RRL's modified gradient descent algorithm and approximated AND/OR nodes that introduce hyper-parameters to improve their learning properties.

Interpretable tabular classification is a starting point for NLNs. In future research, we will explore how they can be adapted to more complex tasks by using different neural network structures. For instance, convolutional NLNs could leverage the AND/OR nodes to tackle interpretable image classification. With convolutional AND kernels and pooling OR layers, these networks could produce interpretable representations of higher-level concepts from 2D arrangements of lower-level concepts. Another example is recurrent NLNs for multi-step reasoning. The tabular NLN presented here can only do single-step reasoning, but by chaining this reasoning through multiple steps, a recurrent NLN could produce multi-step reasoning and solve problems like Sudoku. Again, like in tabular NLN, by learning to predict the finished Sudoku puzzle, the NLN could also discover the rules of Sudoku. Finally, graph NLNs, by working on input graphs with entities and relations, would introduce features of first-order logic by generalizing AND/OR nodes to define universal quantification $\forall$ and existential quantification $\exists$. Moreover, in doing so, graph NLNs could learn to not only predict missing edges or attributes, but also discover underlying relational rules, also known as rule mining in knowledge graphs.

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

# A    Symbols and notation

| | | |
|---:|:---:|:---|
| **input data** | $x$ | an input to the network $x = (x_1, ..., x_m)$ |
| **unobserved data** | $u$ | the values of relevant but unobserved variables |
| **relevant data** | $\omega = (x, u)$ | a complete realization of the modeled random phenomenon |
| | | |
| **concept** | $C_i^l$ | concept $i$ of layer $l$ |
| **input concept** | $C_i^0$ | concept $i$ of input layer $0$, known for input $x$ |
| **output concept** | $C_k^L$ | concept $k$ of last layer $L$, models target concept $Y_k$ |
| **layer** $l +$ | $\mathcal{C}_+^l$ | the set of all the concepts $C_i^l$ of layer $l$ |
| **layer** $l \pm$ | $\mathcal{C}_\pm^l$ | the set of all the concepts $C_i^l$ of layer $l$ and their opposites $(C_i^l)^c$ |
| **presence (rand. event)** | $\omega \in C_i^l$ | random event that the concept $C_i^l$ is present in the realization $\omega$ |
| **presence (cond. prob.)** | $c_i^l(x)$ | probability in $[0, 1]$ that the concept $C_i^l$ is present in $\omega$ given $x$, i.e. $c_i^l(x) = \mathbb{P}\big[\omega \in C_i^l \,|\, x\big]$ |
| | | |
| **necessary concept** | $C \supseteq C_i^l$ | a necessary concept $C$ of AND concept $C_i^l$ |
| **weights AND** | $A_{i,j}^l$ | if $> 0$, probability that $C_j^{l\text{-}1} \supseteq C_i^l$; if $< 0$, $-$probability that $(C_j^{l\text{-}1})^c \supseteq C_i^l$ |
| **unobserved necessary concepts** | $u \in \tilde{N}_i^l$ | random event that the unobserved necessary concepts of AND concept $C_i^l$ are present in $\omega$ |
| **bias AND** | $a_i^l$ | probability that the unobserved necessary concepts $\tilde{N}_i^l$ are present when the observed necessary concepts are present |
| | | |
| **sufficient concept** | $C \subseteq C_i^l$ | a sufficient concept $C$ of OR concept $C_i^l$ |
| **weights OR** | $O_{i,j}^l$ | if $> 0$, probability that $C_j^{l\text{-}1} \subseteq C_i^l$; if $< 0$, $-$probability that $(C_j^{l\text{-}1})^c \subseteq C_i^l$ |
| **unobserved sufficient concepts** | $u \in \tilde{S}_i^l$ | random event that the unobserved sufficient concepts of OR concept $C_i^l$ are present in $\omega$ |
| **bias OR** | $o_i^l$ | probability that the unobserved sufficient concepts $\tilde{S}_i^l$ are present when the observed sufficient concepts are not |
| | | |
| **independence** | $A \perp\!\!\!\perp B$ | random events $A$ and $B$ are independent |
| **set difference** | $A \setminus B$ | set of elements in $A$ that are not in $B$, i.e. $A \setminus B = \{a \,|\, \forall a \in A \text{ s.t. } a \notin B\}$ |
| **indicator function** | $\mathbb{1}(\Phi)$ | equals 1 if its argument $\Phi$ is true and 0 otherwise |
| **positive part** | $[\lambda]_+$ | the positive part of $\lambda \in \mathbb{R}$, i.e. $[\lambda]_+ = \max\{0, \lambda\}$ |
| **negative part** | $[\lambda]_-$ | the negative part of $\lambda \in \mathbb{R}$, i.e. $[\lambda]_- = \max\{0, -\lambda\}$ |

# B  Theory

## B.1  Probabilistic modeling

### B.1.1  Graphical summary of the first two assumptions of independence

The first two assumptions of independence that we make in our probabilistic modeling of NLNs are summarized in the PGM below, where we only specify the structure for an AND concept $C_i^l$ since an OR concept would have the same structure.

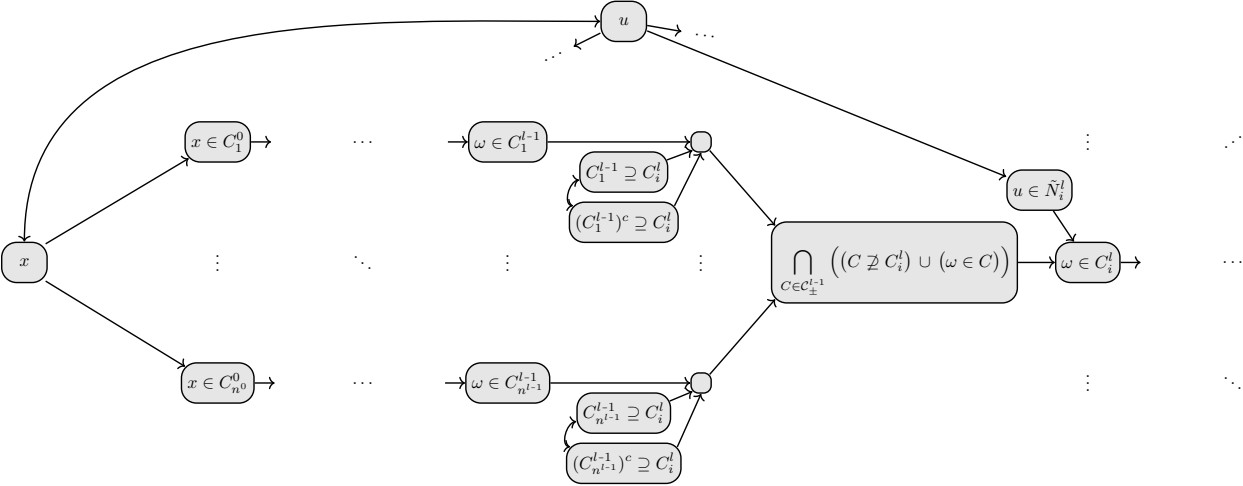

Figure B.1: Probabilistic Graphical Model of NLN's probabilistic modeling (rectangles: random variables, arrows: dependencies, such that two sets of random variables are conditionally independent with respect to a third set of random variables if they are d-separated by it)

- The observed and unobserved data $x$ and $u$ can be correlated;

- The zero-th layer containing the input features $x \in C_i^0$ depends only on the input $x$;

- The unobserved necessary/sufficient concepts $u \in \tilde{N}_i^l$ and $u \in \tilde{S}_i^l$ depend only on the unobserved $u$;

- [**1$^{\text{st}}$ assumption**] The role of a concept $C_j^{l-1}$ as necessary/sufficient to a concept in the next layer $C_j^{l-1} \supseteq C_i^l$ or $C_j^{l-1} \subseteq C_i^l$ is independent of every presence of concept $\omega \in C_{i'}^{l'}$ in the layers $l' \le l - 1$;

- [**2$^{\text{nd}}$ assumption**] The role of a concept $C_j^{l-1}$ as necessary/sufficient to a concept in the next layer $C_j^{l-1} \supseteq C_i^l$ or $C_j^{l-1} \subseteq C_i^l$ is independent of every other roles in the network except for $(C_j^{l-1})^c \supseteq C_i^l$ or $(C_j^{l-1})^c \subseteq C_i^l$, depending on $C_i^l$'s node type.

Critically, we can see that the

- [**3$^{\text{rd}}$ assumption**] The presence of concepts in the same layer $\omega \in C_i^l$ and $\omega \in C_j^l$ (with $i \ne j$) are conditionally independent given the input $x$.

is incorrect whenever two concepts share a common necessary/sufficient concept in the previous layers. In that case, this common "ancestor" creates a fork that d-connects the two concepts, even when conditioning on $x$.

### B.1.2 Derivation with the 3$^{\mathrm{rd}}$ assumption

**AND (conjunction)**   We will derive the following formula for the probability of an AND node $C_i^l$ given $x$

$$\mathbb{P}\big[\omega \in C_i^l \,\big|\, x\big] = \mathbb{P}\big[u \in \tilde{N}_i^l \,\big|\, \omega \in D\big] \cdot \prod_{j=1}^{n^{l-1}} \Big(1 - \mathbb{P}\big[C_j^{l-1} \supseteq C_i^l\big] \cdot \big(1 - \mathbb{P}\big[\omega \in C_j^{l-1} \,\big|\, x\big]\big)\Big)\Big(1 - \mathbb{P}\big[(C_j^{l-1})^c \supseteq C_i^l\big] \cdot \mathbb{P}\big[\omega \in C_j^{l-1} \,\big|\, x\big]\Big),$$

where we define the observed part of definition

$$\omega \in D := \bigcap_{C \in \mathcal{C}_\pm^{l-1}} \Big(\big(C \not\supseteq C_i^l\big) \cup \big(\omega \in C\big)\Big).$$

To make this derivation easier to read, we will abuse notation in the following ways

$$Y := C_i^l,$$
$$C_j := C_j^{l-1}.$$

We will also omit the conditioning on $x$ for the same reason. We remind the reader that the roles of necessary concepts $C_j \supseteq Y$ are independent of the input $x$.

We want to compute

$$\mathbb{P}\left[\omega \in Y\right] = \mathbb{P}\left[(u \in \tilde{N}_i^l) \cap \bigcap_{C \in \mathcal{C}_\pm^{l-1}} \Big(\big(C \not\supseteq Y\big) \cup \big(\omega \in C\big)\Big)\right].$$

Let $D$ be the observed part of the definition

$$\omega \in D := \bigcap_{C \in \mathcal{C}_\pm^{l-1}} \Big(\big(C \not\supseteq Y\big) \cup \big(\omega \in C\big)\Big).$$

Then $\mathbb{P}\left[\omega \in Y\right]$ can be written

$$\mathbb{P}\left[\omega \in Y\right] = \mathbb{P}\left[\omega \in D\right] \cdot \mathbb{P}\left[u \in \tilde{N}_i^l \,\big|\, \omega \in D\right],$$

and we only need to compute $\mathbb{P}\left[\omega \in D\right]$. We will derive it by induction by considering the contribution of every concept $C_j$ in the previous layer, one at a time. We begin with the law of total probability with respect to the role of $C_1$ or its negation as a necessary concept, of which there are three possible cases

$$C_1 \supseteq Y, \ (C_1)^c \not\supseteq Y, \qquad (C_1)^c \supseteq Y, \ C_1 \not\supseteq Y, \qquad C_1 \not\supseteq Y, \ (C_1)^c \not\supseteq Y.$$

Since we assume that at most one of $C_1$ and $(C_1)^c$ can be a necessary concept, we will note these three cases as

$$C_1 \supseteq Y, \qquad\qquad (C_1)^c \supseteq Y, \qquad\qquad C_1 \not\supseteq Y, (C_1)^c \not\supseteq Y.$$

We obtain

$$\mathbb{P}\left[\omega \in D\right] = \mathbb{P}\left[C_1 \supseteq Y\right] \cdot \mathbb{P}\left[\omega \in D \,|\, C_1 \supseteq Y\right]$$
$$+ \mathbb{P}\left[(C_1)^c \supseteq Y\right] \cdot \mathbb{P}\left[\omega \in D \,|\, (C_1)^c \supseteq Y\right]$$
$$+ \mathbb{P}\left[C_1 \not\supseteq Y, (C_1)^c \not\supseteq Y\right] \cdot \mathbb{P}\left[\omega \in D \,|\, C_1 \not\supseteq Y, (C_1)^c \not\supseteq Y\right].$$

Expanding $\mathbb{P}\left[\omega \in D \,|\, C_1 \supseteq Y\right]$, we get

$$\mathbb{P}\left[\omega \in D \,|\, C_1 \supseteq Y\right] = \mathbb{P}\left[\omega \in C_1 \,|\, C_1 \supseteq Y\right] \cdot \mathbb{P}\left[\omega \in D \,|\, \omega \in C_1, C_1 \supseteq Y\right]$$
$$+ \big(1 - \mathbb{P}\left[\omega \in C_1 \,|\, C_1 \supseteq Y\right]\big) \cdot \mathbb{P}\left[\omega \in D \,|\, \omega \in (C_1)^c, C_1 \supseteq Y\right]$$
$$= \mathbb{P}\left[\omega \in C_1\right] \cdot \mathbb{P}\left[\omega \in D \,|\, \omega \in C_1, C_1 \supseteq Y\right]$$

where we assume [**1ˢᵗ assumption**] independence between the presence of concept $C_1$ and its role

$$\omega \in C_1 \perp\!\!\!\perp C_1 \sqsupseteq Y.$$

Expanding $\mathbb{P}\left[\omega \in D \mid (C_1)^c \sqsupseteq Y\right]$, we get

$$
\begin{aligned}
\mathbb{P}\left[\omega \in D \mid (C_1)^c \sqsupseteq Y\right] &= \mathbb{P}\left[\omega \in (C_1)^c \mid (C_1)^c \sqsupseteq Y\right] \cdot \mathbb{P}\left[\omega \in D \mid \omega \in (C_1)^c, (C_1)^c \sqsupseteq Y\right] \\
&\quad + \left(1 - \mathbb{P}\left[\omega \in (C_1)^c \mid (C_1)^c \sqsupseteq Y\right]\right) \cdot \mathbb{P}\left[\omega \in D \mid \omega \in C_1, (C_1)^c \sqsupseteq Y\right]^{\nearrow 0} \\
&= \mathbb{P}\left[\omega \in (C_1)^c\right] \cdot \mathbb{P}\left[\omega \in D \mid \omega \in (C_1)^c, (C_1)^c \sqsupseteq Y\right]
\end{aligned}
$$

where we assume [**1ˢᵗ assumption**] independence between the presence of concept $C_1$ and its negated role

$$\omega \in (C_1)^c \perp\!\!\!\perp (C_1)^c \sqsupseteq Y.$$

Combining the two previous cases, so far we have

$$
\begin{aligned}
\mathbb{P}\left[\omega \in D\right] &= \mathbb{P}\left[C_1 \sqsupseteq Y\right] \cdot \mathbb{P}\left[\omega \in C_1\right] \cdot \mathbb{P}\left[\omega \in D \mid \omega \in C_1, C_1 \sqsupseteq Y\right] \\
&\quad + \mathbb{P}\left[(C_1)^c \sqsupseteq Y\right] \cdot \left(1 - \mathbb{P}\left[\omega \in C_1\right]\right) \cdot \mathbb{P}\left[\omega \in D \mid \omega \in (C_1)^c, (C_1)^c \sqsupseteq Y\right] \\
&\quad + \mathbb{P}\left[C_1 \not\sqsupseteq Y, (C_1)^c \not\sqsupseteq Y\right] \cdot \mathbb{P}\left[\omega \in D \mid C_1 \not\sqsupseteq Y, (C_1)^c \not\sqsupseteq Y\right].
\end{aligned}
$$

By using the logical definition of $D$, we get

$$
\begin{aligned}
\mathbb{P}\left[\omega \in D\right] &= \mathbb{P}\left[C_1 \sqsupseteq Y\right] \cdot \mathbb{P}\left[\omega \in C_1\right] \cdot \mathbb{P}\left[\bigcap_{C \in \mathcal{C}_\pm^{l-1}} \left((C \not\sqsupseteq Y) \cup (\omega \in C)\right) \,\middle|\, \omega \in C_1, C_1 \sqsupseteq Y\right] \\
&\quad + \mathbb{P}\left[(C_1)^c \sqsupseteq Y\right] \cdot \left(1 - \mathbb{P}\left[\omega \in C_1\right]\right) \cdot \mathbb{P}\left[\bigcap_{C \in \mathcal{C}_\pm^{l-1}} \left((C \not\sqsupseteq Y) \cup (\omega \in C)\right) \,\middle|\, \omega \in (C_1)^c, (C_1)^c \sqsupseteq Y\right] \\
&\quad + \mathbb{P}\left[C_1 \not\sqsupseteq Y, (C_1)^c \not\sqsupseteq Y\right] \cdot \mathbb{P}\left[\bigcap_{C \in \mathcal{C}_\pm^{l-1}} \left((C \not\sqsupseteq Y) \cup (\omega \in C)\right) \,\middle|\, C_1 \not\sqsupseteq Y, (C_1)^c \not\sqsupseteq Y\right] \\
&= \mathbb{P}\left[C_1 \sqsupseteq Y\right] \cdot \mathbb{P}\left[\omega \in C_1\right] \cdot \mathbb{P}\left[\bigcap_{C \in (\mathcal{C}_\pm^{l-1} \setminus \{C_1, \neg C_1\})} \left((C \not\sqsupseteq Y) \cup (\omega \in C)\right) \,\middle|\, \omega \in C_1, C_1 \sqsupseteq Y\right] \\
&\quad + \mathbb{P}\left[(C_1)^c \sqsupseteq Y\right] \cdot \left(1 - \mathbb{P}\left[\omega \in C_1\right]\right) \cdot \mathbb{P}\left[\bigcap_{C \in (\mathcal{C}_\pm^{l-1} \setminus \{C_1, \neg C_1\})} \left((C \not\sqsupseteq Y) \cup (\omega \in C)\right) \,\middle|\, \omega \in (C_1)^c, (C_1)^c \sqsupseteq Y\right] \\
&\quad + \mathbb{P}\left[C_1 \not\sqsupseteq Y, (C_1)^c \not\sqsupseteq Y\right] \cdot \mathbb{P}\left[\bigcap_{C \in (\mathcal{C}_\pm^{l-1} \setminus \{C_1, \neg C_1\})} \left((C \not\sqsupseteq Y) \cup (\omega \in C)\right) \,\middle|\, C_1 \not\sqsupseteq Y, (C_1)^c \not\sqsupseteq Y\right]
\end{aligned}
$$

If we assume the following independences

$$C_1 \supseteq Y \; \perp\!\!\!\perp \bigcap_{C \in (\mathcal{C}^{l-1}_{\pm} \setminus \{C_1, \neg C_1\})} \Big( (C \not\supseteq Y) \cup (\omega \in C) \Big) \; \Big| \; x,$$

$$(C_1)^c \supseteq Y \; \perp\!\!\!\perp \bigcap_{C \in (\mathcal{C}^{l-1}_{\pm} \setminus \{C_1, \neg C_1\})} \Big( (C \not\supseteq Y) \cup (\omega \in C) \Big) \; \Big| \; x,$$

$$\omega \in C_1 \; \perp\!\!\!\perp \bigcap_{C \in (\mathcal{C}^{l-1}_{\pm} \setminus \{C_1, \neg C_1\})} \Big( (C \not\supseteq Y) \cup (\omega \in C) \Big) \; \Big| \; x,$$

*i.e.* if we additionally assume

- **[1st assumption]** The role of a concept $C_j$ as necessary to $Y$ ($C_j \supseteq Y$) is independent of every presence of concept $\omega \in C_{j'}$;

- **[2nd assumption]** The role of a concept $C_j$ as necessary to $Y$ ($C_j \supseteq Y$) is independent of every other concept $C_{j'}$'s role as necessary to $Y$ except for $(C_j)^c \supseteq Y$;

- **[3rd assumption]** The presence of concepts in the same layer $\omega \in C_j$ and $\omega \in C_{j'}$ (with $j \neq j'$) are conditionally independent given the input $x$;

we get

$$\mathbb{P}\left[\omega \in D\right] = \Big( \mathbb{P}\left[C_1 \supseteq Y\right] \cdot \mathbb{P}\left[\omega \in C_1\right] + \mathbb{P}\left[(C_1)^c \supseteq Y\right] \cdot \big(1 - \mathbb{P}\left[\omega \in C_1\right]\big) + \mathbb{P}\left[C_1 \not\supseteq Y, (C_1)^c \not\supseteq Y\right] \Big)$$

$$\cdot \mathbb{P}\left[ \bigcap_{C \in (\mathcal{C}^{l-1}_{\pm} \setminus \{C_1, \neg C_1\})} \Big( (C \not\supseteq Y) \cup (\omega \in C) \Big) \right]$$

$$= \prod_{j=1}^{n^{l-1}} \underbrace{\Big( \mathbb{P}\left[C_j \supseteq Y\right] \cdot \mathbb{P}\left[\omega \in C_j\right] + \mathbb{P}\left[(C_j)^c \supseteq Y\right] \cdot \big(1 - \mathbb{P}\left[\omega \in C_j\right]\big) + \mathbb{P}\left[C_j \not\supseteq Y, (C_j)^c \not\supseteq Y\right] \Big)}_{P_j}.$$

We make one final assumption about the probability structure of $C_j \supseteq Y$ and $(C_j)^c \supseteq Y$. We assume that if the network believes one is possible (probability $> 0$), then it believes the other is impossible (probability $= 0$) and vice versa. This is how we are able to model both $\mathbb{P}\left[C_j \supseteq Y\right]$ and $\mathbb{P}\left[(C_j)^c \supseteq Y\right]$ with a single variable

$$A^l_{i,j} = \underbrace{\mathbb{P}\left[C_j \supseteq Y\right]}_{[A^l_{i,j}]_+} - \underbrace{\mathbb{P}\left[(C_j)^c \supseteq Y\right]}_{[A^l_{i,j}]_-}.$$

If we look at the term $P_j$ in the product on $j$, we can see that

$$P_j = \begin{cases} \mathbb{P}\left[C_j \supseteq Y\right] \cdot \mathbb{P}\left[\omega \in C_j\right] + \mathbb{P}\left[C_j \not\supseteq Y, (C_j)^c \not\supseteq Y\right], & \text{if } \mathbb{P}\left[C_j \supseteq Y\right] > 0, \ \ \mathbb{P}\left[(C_j)^c \supseteq Y\right] = 0, \\ \mathbb{P}\left[(C_j)^c \supseteq Y\right] \cdot \left(1 - \mathbb{P}\left[\omega \in C_j\right]\right) + \mathbb{P}\left[C_j \not\supseteq Y, (C_j)^c \not\supseteq Y\right], & \text{if } \mathbb{P}\left[C_j \supseteq Y\right] = 0, \ \ \mathbb{P}\left[(C_j)^c \supseteq Y\right] > 0, \\ \mathbb{P}\left[C_j \not\supseteq Y, (C_j)^c \not\supseteq Y\right], & \text{if } \mathbb{P}\left[C_j \supseteq Y\right] = \mathbb{P}\left[(C_j)^c \supseteq Y\right] = 0. \end{cases}$$

$$= \begin{cases} \mathbb{P}\left[C_j \supseteq Y\right] \cdot \mathbb{P}\left[\omega \in C_j\right] + \left(1 - \mathbb{P}\left[C_j \supseteq Y\right]\right), & \text{if } \mathbb{P}\left[C_j \supseteq Y\right] > 0, \ \ \mathbb{P}\left[(C_j)^c \supseteq Y\right] = 0, \\ \mathbb{P}\left[(C_j)^c \supseteq Y\right] \cdot \left(1 - \mathbb{P}\left[\omega \in C_j\right]\right) + \left(1 - \mathbb{P}\left[(C_j)^c \supseteq Y\right]\right), & \text{if } \mathbb{P}\left[C_j \supseteq Y\right] = 0, \ \ \mathbb{P}\left[(C_j)^c \supseteq Y\right] > 0, \\ 1, & \text{if } \mathbb{P}\left[C_j \supseteq Y\right] = \mathbb{P}\left[(C_j)^c \supseteq Y\right] = 0. \end{cases}$$

$$= \begin{cases} 1 - \mathbb{P}\left[C_j \supseteq Y\right] \cdot \left(1 - \mathbb{P}\left[\omega \in C_j\right]\right), & \text{if } \mathbb{P}\left[C_j \supseteq Y\right] > 0, \ \ \mathbb{P}\left[(C_j)^c \supseteq Y\right] = 0, \\ 1 - \mathbb{P}\left[(C_j)^c \supseteq Y\right] \cdot \mathbb{P}\left[\omega \in C_j\right], & \text{if } \mathbb{P}\left[C_j \supseteq Y\right] = 0, \ \ \mathbb{P}\left[(C_j)^c \supseteq Y\right] > 0, \\ 1, & \text{if } \mathbb{P}\left[C_j \supseteq Y\right] = \mathbb{P}\left[(C_j)^c \supseteq Y\right] = 0. \end{cases}$$

$$= \left(1 - \mathbb{P}\left[C_j \supseteq Y\right] \cdot \left(1 - \mathbb{P}\left[\omega \in C_j\right]\right)\right) \cdot \left(1 - \mathbb{P}\left[(C_j)^c \supseteq Y\right] \cdot \mathbb{P}\left[\omega \in C_j\right]\right)$$

$$= 1 - \mathbb{P}\left[C_j \supseteq Y\right] \cdot \left(1 - \mathbb{P}\left[\omega \in C_j\right]\right) - \mathbb{P}\left[(C_j)^c \supseteq Y\right] \cdot \mathbb{P}\left[\omega \in C_j\right] + \underbrace{\mathbb{P}\left[C_j \supseteq Y\right] \cdot \mathbb{P}\left[(C_j)^c \supseteq Y\right] \cdot \left(1 - \mathbb{P}\left[\omega \in C_j\right]\right) \cdot \mathbb{P}\left[\omega \in C_j\right]}_{0}$$

$$= 1 - \mathbb{P}\left[C_j \supseteq Y\right] \cdot \left(1 - \mathbb{P}\left[\omega \in C_j\right]\right) - \mathbb{P}\left[(C_j)^c \supseteq Y\right] \cdot \mathbb{P}\left[\omega \in C_j\right]$$

since we only ever have at most one of $\mathbb{P}\left[C_j \supseteq Y\right]$ and $\mathbb{P}\left[(C_j)^c \supseteq Y\right]$ that is non-zero. We conclude

$$\mathbb{P}\left[\omega \in Y\right] = \mathbb{P}\left[u \in \tilde{N}_i^l \,\middle|\, \omega \in D\right] \prod_{j=1}^{n^{l-1}} \left(1 - \mathbb{P}\left[C_j \supseteq Y\right] \cdot \left(1 - \mathbb{P}\left[\omega \in C_j\right]\right)\right)\left(1 - \mathbb{P}\left[(C_j)^c \supseteq Y\right] \cdot \mathbb{P}\left[\omega \in C_j\right]\right)$$

$$= \mathbb{P}\left[u \in \tilde{N}_i^l \,\middle|\, \omega \in D\right] \prod_{j=1}^{n^{l-1}} \left(1 - \mathbb{P}\left[C_j \supseteq Y\right] \cdot \left(1 - \mathbb{P}\left[\omega \in C_j\right]\right) - \mathbb{P}\left[(C_j)^c \supseteq Y\right] \cdot \mathbb{P}\left[\omega \in C_j\right]\right).$$

**OR (disjunction)** We will derive the following formula for the probability of an AND node $C_j^l$ given $x$

$$\mathbb{P}[\omega \in C_i^l \,|\, x] = 1 - \left(1 - \mathbb{P}[u \in \tilde{S}_i^l \,|\, (\omega \in D)^c]\right)\prod_{j=1}^{n^{l-1}}\left(1 - \mathbb{P}[C_j^{l-1} \subseteq C_i^l] \cdot \mathbb{P}[\omega \in C_j^{l-1} \,|\, x]\right)\left(1 - \mathbb{P}[(C_j^{l-1})^c \subseteq C_i^l] \cdot \left(1 - \mathbb{P}[\omega \in C_j^{l-1} \,|\, x]\right)\right),$$

where we define the observed part of definition

$$\omega \in D := \bigcup_{C \in \mathcal{C}_\pm^{l-1}} \left((C \subseteq Y) \cap (\omega \in C)\right).$$

To make this derivation easier to read, we will again abuse notation in the following ways

$$Y := C_i^l,$$
$$C_j := C_j^{l-1}.$$

We will also omit the conditioning on $x$ for the same reason. We remind the reader that the roles of sufficient concepts $C_j \subseteq Y$ are independent of the input $x$.

We want to compute

$$\mathbb{P}\left[\omega \in Y\right] = \mathbb{P}\left[(u \in \tilde{S}_i^l) \cup \bigcup_{C \in \mathcal{C}_\pm^{l-1}} \left((C \subseteq Y) \cap (\omega \in C)\right)\right].$$

Let $\omega \in D$ be the observed part of the definition

$$\omega \in D := \bigcup_{C \in \mathcal{C}_{\pm}^{l-1}} \left( (C \subseteq Y) \cap (\omega \in C) \right).$$

Then $\mathbb{P}[\omega \in Y]$ can be rewritten

$$\mathbb{P}[\omega \in Y] = \mathbb{P}\left[ \left( u \in \tilde{S}_i^l \right) \cup (\omega \in D) \right].$$

Considering the complement of $\omega \in Y$ (its absence $(\omega \in Y)^c$), we have

$$
\begin{aligned}
1 - \mathbb{P}[\omega \in Y] &= \mathbb{P}[(\omega \in Y)^c] \\
&= \mathbb{P}\left[ (u \in \tilde{S}_i^l)^c \cap (\omega \in D)^c \right] \\
&= \mathbb{P}[(\omega \in D)^c] \cdot \mathbb{P}\left[ (u \in \tilde{S}_i^l)^c \,\middle|\, (\omega \in D)^c \right] \\
&= \left( 1 - \mathbb{P}[\omega \in D] \right) \cdot \left( 1 - \mathbb{P}\left[ u \in \tilde{S}_i^l \,\middle|\, (\omega \in D)^c \right] \right) \\
\mathbb{P}[\omega \in Y] &= 1 - \left( 1 - \mathbb{P}[\omega \in D] \right) \cdot \left( 1 - \mathbb{P}\left[ u \in \tilde{S}_i^l \,\middle|\, (\omega \in D)^c \right] \right),
\end{aligned}
$$

and we only need to compute $\mathbb{P}[D]$. Again, we will derive it by induction by considering the contribution of every concept $C_j$ in the previous layer, one at a time. We begin with the law of total probability with respect to the role of $C_1$ or its negation as a sufficient concept, of which there are three possible cases

$$C_1 \subseteq Y, \ (C_1)^c \not\subseteq Y, \qquad (C_1)^c \subseteq Y, \ C_1 \not\subseteq Y, \qquad C_1 \not\subseteq Y, \ (C_1)^c \not\subseteq Y.$$

Since we assume that at most one of $C_1$ and $(C_1)^c$ can be a sufficient concept, we will note these three cases as

$$C_1 \subseteq Y, \qquad\qquad (C_1)^c \subseteq Y, \qquad\qquad C_1 \not\subseteq Y, (C_1)^c \not\subseteq Y.$$

We obtain

$$
\begin{aligned}
\mathbb{P}[\omega \in D] &= \mathbb{P}[C_1 \subseteq Y] \cdot \mathbb{P}[\omega \in D \,|\, C_1 \subseteq Y] \\
&\quad + \mathbb{P}[(C_1)^c \subseteq Y] \cdot \mathbb{P}[\omega \in D \,|\, (C_1)^c \subseteq Y] \\
&\quad + \mathbb{P}[C_1 \not\subseteq Y, (C_1)^c \not\subseteq Y] \cdot \mathbb{P}[\omega \in D \,|\, C_1 \not\subseteq Y, (C_1)^c \not\subseteq Y].
\end{aligned}
$$

Expanding $\mathbb{P}[\omega \in D \,|\, C_1 \subseteq Y]$, we get

$$
\begin{aligned}
\mathbb{P}[\omega \in D \,|\, C_1 \subseteq Y] &= \mathbb{P}[\omega \in C_1 \,|\, C_1 \subseteq Y] \cdot \overbrace{\mathbb{P}[\omega \in D \,|\, \omega \in C_1, C_1 \subseteq Y]}^{1} \\
&\quad + \left( 1 - \mathbb{P}[\omega \in C_1 \,|\, C_1 \subseteq Y] \right) \cdot \mathbb{P}[\omega \in D \,|\, \omega \in (C_1)^c, C_1 \subseteq Y] \\
&= \mathbb{P}[\omega \in C_1] + \left( 1 - \mathbb{P}[\omega \in C_1] \right) \cdot \mathbb{P}[\omega \in D \,|\, \omega \in (C_1)^c, C_1 \subseteq Y]
\end{aligned}
$$

where we assume [**1$^{\text{st}}$ assumption**] independence between the presence of concept $C_1$ and its role

$$\omega \in C_1 \perp\!\!\!\perp C_1 \subseteq Y.$$

Expanding $\mathbb{P}[\omega \in D \,|\, (C_1)^c \subseteq Y]$, we get

$$
\begin{aligned}
\mathbb{P}[\omega \in D \,|\, (C_1)^c \subseteq Y] &= \mathbb{P}[\omega \in (C_1)^c \,|\, (C_1)^c \subseteq Y] \cdot \overbrace{\mathbb{P}[\omega \in D \,|\, \omega \in (C_1)^c, (C_1)^c \subseteq Y]}^{1} \\
&\quad + \left( 1 - \mathbb{P}[\omega \in (C_1)^c \,|\, (C_1)^c \subseteq Y] \right) \cdot \mathbb{P}[\omega \in D \,|\, \omega \in C_1, (C_1)^c \subseteq Y] \\
&= \mathbb{P}[\omega \in (C_1)^c] + \left( 1 - \mathbb{P}[\omega \in (C_1)^c] \right) \cdot \mathbb{P}[\omega \in D \,|\, \omega \in C_1, (C_1)^c \subseteq Y]
\end{aligned}
$$

where we assume [**$1^{\text{st}}$ assumption**] independence between the presence of concept $C_1$ and its role

$$\omega \in (C_1)^c \perp\!\!\!\perp (C_1)^c \subseteq Y.$$

Combining the two previous cases, so far we have

$$\mathbb{P}\left[\omega \in D\right] = \mathbb{P}\left[C_1 \subseteq Y\right] \cdot \left(\mathbb{P}\left[\omega \in C_1\right] + \left(1 - \mathbb{P}\left[\omega \in C_1\right]\right) \cdot \mathbb{P}\left[\omega \in D \,\middle|\, \omega \in (C_1)^c, C_1 \subseteq Y\right]\right)$$
$$+ \mathbb{P}\left[(C_1)^c \subseteq Y\right] \cdot \left(\mathbb{P}\left[\omega \in (C_1)^c\right] + \left(1 - \mathbb{P}\left[\omega \in (C_1)^c\right]\right) \cdot \mathbb{P}\left[\omega \in D \,\middle|\, \omega \in C_1, (C_1)^c \subseteq Y\right]\right)$$
$$+ \mathbb{P}\left[C_1 \not\subseteq Y, (C_1)^c \not\subseteq Y\right] \cdot \mathbb{P}\left[\omega \in D \,\middle|\, (C_1)^c \not\subseteq Y, C_1 \not\subseteq Y\right].$$

By using the logical definition of $D$ and by denoting $\mathcal{C}^{l-1}_{\pm,-1} := \mathcal{C}^{l-1}_{\pm} \setminus \{C_1, \neg C_1\}$, we get

$$\mathbb{P}\left[\omega \in D\right] = \mathbb{P}\left[C_1 \subseteq Y\right] \cdot \left(\mathbb{P}\left[\omega \in C_1\right] + \left(1 - \mathbb{P}\left[\omega \in C_1\right]\right) \cdot \mathbb{P}\left[\bigcup_{C \in \mathcal{C}^{l-1}_{\pm}} \left((C \subseteq Y) \cap (\omega \in C)\right) \,\middle|\, \omega \in (C_1)^c, C_1 \subseteq Y\right]\right)$$

$$+ \mathbb{P}\left[(C_1)^c \subseteq Y\right] \cdot \left(\mathbb{P}\left[\omega \in (C_1)^c\right] + \left(1 - \mathbb{P}\left[\omega \in (C_1)^c\right]\right) \cdot \mathbb{P}\left[\bigcup_{C \in \mathcal{C}^{l-1}_{\pm}} \left((C \subseteq Y) \cap (\omega \in C)\right) \,\middle|\, \omega \in C_1, (C_1)^c \subseteq Y\right]\right)$$

$$+ \mathbb{P}\left[C_1 \not\subseteq Y, (C_1)^c \not\subseteq Y\right] \cdot \mathbb{P}\left[\bigcup_{C \in \mathcal{C}^{l-1}_{\pm}} \left((C \subseteq Y) \cap (\omega \in C)\right) \,\middle|\, (C_1)^c \not\subseteq Y, C_1 \not\subseteq Y\right]$$

$$= \mathbb{P}\left[C_1 \subseteq Y\right] \cdot \left(\mathbb{P}\left[\omega \in C_1\right] + \left(1 - \mathbb{P}\left[\omega \in C_1\right]\right) \cdot \mathbb{P}\left[\bigcup_{C \in \mathcal{C}^{l-1}_{\pm,-1}} \left((C \subseteq Y) \cap (\omega \in C)\right) \,\middle|\, \omega \in (C_1)^c, C_1 \subseteq Y\right]\right)$$

$$+ \mathbb{P}\left[(C_1)^c \subseteq Y\right] \cdot \left(\mathbb{P}\left[\omega \in (C_1)^c\right] + \left(1 - \mathbb{P}\left[\omega \in (C_1)^c\right]\right) \cdot \mathbb{P}\left[\bigcup_{C \in \mathcal{C}^{l-1}_{\pm,-1}} \left((C \subseteq Y) \cap (\omega \in C)\right) \,\middle|\, \omega \in C_1, (C_1)^c \subseteq Y\right]\right)$$

$$+ \mathbb{P}\left[C_1 \not\subseteq Y, (C_1)^c \not\subseteq Y\right] \cdot \mathbb{P}\left[\bigcup_{C \in \mathcal{C}^{l-1}_{\pm,-1}} \left((C \subseteq Y) \cap (\omega \in C)\right) \,\middle|\, (C_1)^c \not\subseteq Y, C_1 \not\subseteq Y\right].$$

If we assume the following independences

$$C_1 \subseteq Y \;\perp\!\!\!\perp\; \bigcup_{C \in (\mathcal{C}^{l-1}_{\pm} \setminus \{C_1, \neg C_1\})} \left((C \subseteq Y) \cap (\omega \in C)\right) \;\middle|\; x,$$

$$(C_1)^c \subseteq Y \;\perp\!\!\!\perp\; \bigcup_{C \in (\mathcal{C}^{l-1}_{\pm} \setminus \{C_1, \neg C_1\})} \left((C \subseteq Y) \cap (\omega \in C)\right) \;\middle|\; x,$$

$$\omega \in C_1 \;\perp\!\!\!\perp\; \bigcup_{C \in (\mathcal{C}^{l-1}_{\pm} \setminus \{C_1, \neg C_1\})} \left((C \subseteq Y) \cap (\omega \in C)\right) \;\middle|\; x,$$

*i.e.* if we additionally assume

- [**$1^{\text{st}}$ assumption**] The role of a concept $C_j$ as sufficient to $Y$ ($C_j \subseteq Y$) is independent of every presence of concept $\omega \in C_{j'}$;

- [**2nd assumption**] The role of a concept $C_j$ as sufficient to $Y$ $(C_j \subseteq Y)$ is independent of every other concept $C_{j'}$'s role as sufficient to $Y$ except for $(C_j)^c \subseteq Y$;

- [**3rd assumption**] The presence of concepts in the same layer $\omega \in C_j$ and $\omega \in C_{j'}$ (with $j \neq j'$) are conditionally independent given the input $x$;

we get

$$\mathbb{P}\left[\omega \in D\right] = \mathbb{P}\left[C_1 \subseteq Y\right] \cdot \mathbb{P}\left[\omega \in C_1\right] + \mathbb{P}\left[(C_1)^c \subseteq Y\right]\left(1 - \mathbb{P}\left[\omega \in C_1\right]\right)$$
$$+ \underbrace{\left(\mathbb{P}\left[C_1 \subseteq Y\right]\left(1 - \mathbb{P}\left[\omega \in C_1\right]\right) + \mathbb{P}\left[(C_1)^c \subseteq Y\right] \cdot \mathbb{P}\left[\omega \in C_1\right] + \mathbb{P}\left[C_1 \not\subseteq Y, (C_1)^c \not\subseteq Y\right]\right)}_{R_1}$$
$$\cdot \mathbb{P}\left[\bigcup_{C \in (\mathcal{C}_{\pm}^{l-1} \setminus \{C_1, \neg C_1\})} \left((C \subseteq Y) \cap (\omega \in C)\right)\right].$$

If we look at $R_1$, we can see that

$$R_1 = \begin{cases} \mathbb{P}\left[C_1 \subseteq Y\right] \cdot \left(1 - \mathbb{P}\left[\omega \in C_1\right]\right) + \mathbb{P}\left[C_1 \not\subseteq Y, (C_1)^c \not\subseteq Y\right], & \text{if } \mathbb{P}\left[C_1 \subseteq Y\right] > 0, \ \mathbb{P}\left[(C_1)^c \subseteq Y\right] = 0, \\ \mathbb{P}\left[(C_1)^c \subseteq Y\right] \cdot \mathbb{P}\left[\omega \in C_1\right] + \mathbb{P}\left[C_1 \not\subseteq Y, (C_1)^c \not\subseteq Y\right], & \text{if } \mathbb{P}\left[C_1 \subseteq Y\right] = 0, \ \mathbb{P}\left[(C_1)^c \subseteq Y\right] > 0, \\ \mathbb{P}\left[C_1 \not\subseteq Y, (C_1)^c \not\subseteq Y\right], & \text{if } \mathbb{P}\left[C_1 \subseteq Y\right] = \mathbb{P}\left[(C_1)^c \subseteq Y\right] = 0. \end{cases}$$

$$= \begin{cases} \mathbb{P}\left[C_1 \subseteq Y\right] \cdot \left(1 - \mathbb{P}\left[\omega \in C_1\right]\right) + \left(1 - \mathbb{P}\left[C_1 \subseteq Y\right]\right), & \text{if } \mathbb{P}\left[C_1 \subseteq Y\right] > 0, \ \mathbb{P}\left[(C_1)^c \subseteq Y\right] = 0, \\ \mathbb{P}\left[(C_1)^c \subseteq Y\right] \cdot \mathbb{P}\left[\omega \in C_1\right] + \left(1 - \mathbb{P}\left[(C_1)^c \subseteq Y\right]\right), & \text{if } \mathbb{P}\left[C_1 \subseteq Y\right] = 0, \ \mathbb{P}\left[(C_1)^c \subseteq Y\right] > 0, \\ 1, & \text{if } \mathbb{P}\left[C_1 \subseteq Y\right] = \mathbb{P}\left[(C_1)^c \subseteq Y\right] = 0. \end{cases}$$

$$= \begin{cases} 1 - \mathbb{P}\left[C_1 \subseteq Y\right] \cdot \mathbb{P}[\omega \in C_1], & \text{if } \mathbb{P}\left[C_1 \subseteq Y\right] > 0, \ \mathbb{P}\left[(C_1)^c \subseteq Y\right] = 0, \\ 1 - \mathbb{P}[(C_1)^c \subseteq Y] \cdot \left(1 - \mathbb{P}\left[\omega \in C_1\right]\right), & \text{if } \mathbb{P}\left[C_1 \subseteq Y\right] = 0, \ \mathbb{P}\left[(C_1)^c \subseteq Y\right] > 0, \\ 1, & \text{if } \mathbb{P}\left[C_1 \subseteq Y\right] = \mathbb{P}\left[(C_1)^c \subseteq Y\right] = 0. \end{cases}$$

$$= \left(1 - \mathbb{P}\left[C_1 \subseteq Y\right] \cdot \mathbb{P}[\omega \in C_1]\right) \cdot \left(1 - \mathbb{P}[(C_1)^c \subseteq Y] \cdot \left(1 - \mathbb{P}\left[\omega \in C_1\right]\right)\right)$$

$$= 1 - \mathbb{P}\left[C_1 \subseteq Y\right] \cdot \mathbb{P}[\omega \in C_1] - \mathbb{P}[(C_1)^c \subseteq Y] \cdot \left(1 - \mathbb{P}\left[\omega \in C_1\right]\right) + \overbrace{\mathbb{P}\left[C_1 \subseteq Y\right] \cdot \mathbb{P}[(C_1)^c \subseteq Y] \cdot \mathbb{P}[\omega \in C_1] \cdot \left(1 - \mathbb{P}\left[\omega \in C_1\right]\right)}^{0}$$

$$= 1 - \mathbb{P}\left[C_1 \subseteq Y\right] \cdot \mathbb{P}[\omega \in C_1] - \mathbb{P}[(C_1)^c \subseteq Y] \cdot \left(1 - \mathbb{P}\left[\omega \in C_1\right]\right)$$

We conclude that

$$\mathbb{P}\left[\omega \in Y\right] = 1 - \left(1 - \mathbb{P}\left[u \in \tilde{S}_i^l \mid (\omega \in D)^c\right]\right) \prod_{j=1}^{n^{l-1}} \left(1 - \mathbb{P}\left[C_j \subseteq Y\right] \cdot \mathbb{P}[\omega \in C_j]\right)\left(1 - \mathbb{P}[(C_j)^c \subseteq Y] \cdot \left(1 - \mathbb{P}\left[\omega \in C_j\right]\right)\right)$$

$$= 1 - \left(1 - \mathbb{P}\left[u \in \tilde{S}_i^l \mid (\omega \in D)^c\right]\right) \prod_{j=1}^{n^{l-1}} \left(1 - \mathbb{P}\left[C_j \subseteq Y\right] \cdot \mathbb{P}[\omega \in C_j] - \mathbb{P}[(C_j)^c \subseteq Y] \cdot \left(1 - \mathbb{P}\left[\omega \in C_j\right]\right)\right).$$

To get this final form, we proceed by induction to prove

$$\mathbb{P}\left[\omega \in D\right] = 1 - \prod_{j=1}^{n^{l-1}} \left(1 - \mathbb{P}\left[C_j \subseteq Y\right] \cdot \mathbb{P}[\omega \in C_j] - \mathbb{P}[(C_j)^c \subseteq Y] \cdot \left(1 - \mathbb{P}\left[\omega \in C_j\right]\right)\right).$$

For $n^{l-1} = 1$, we have

$$
\begin{aligned}
\mathbb{P}\left[\omega \in D\right] &= \mathbb{P}\left[C_1 \subseteq Y\right] \cdot \mathbb{P}\left[\omega \in C_1\right] + \mathbb{P}\left[(C_1)^c \subseteq Y\right]\left(1 - \mathbb{P}\left[\omega \in C_1\right]\right) \\
&\quad + \left(1 - \mathbb{P}\left[C_1 \subseteq Y\right] \cdot \mathbb{P}[\omega \in C_1] - \mathbb{P}[(C_1)^c \subseteq Y]\left(1 - \mathbb{P}\left[\omega \in C_1\right]\right)\right) \cdot 0 \\
&= 1 - \left(1 - \mathbb{P}\left[C_1 \subseteq Y\right] \cdot \mathbb{P}[\omega \in C_1] - \mathbb{P}[(C_1)^c \subseteq Y]\left(1 - \mathbb{P}\left[\omega \in C_1\right]\right)\right) \\
&= 1 - \prod_{j=1}^{n^{l-1}} \left(1 - \mathbb{P}\left[C_j \subseteq Y\right] \cdot \mathbb{P}[\omega \in C_j] - \mathbb{P}[C_j \not\subseteq Y]\left(1 - \mathbb{P}\left[\omega \in C_j\right]\right)\right).
\end{aligned}
$$

If this expression is true for $n^l = k$, for $n^l = k+1$, we have

$$
\begin{aligned}
\mathbb{P}\left[\omega \in D\right] &= \mathbb{P}\left[C_1 \subseteq Y\right] \cdot \mathbb{P}\left[\omega \in C_1\right] + \mathbb{P}\left[(C_1)^c \subseteq Y\right]\left(1 - \mathbb{P}\left[\omega \in C_1\right]\right) \\
&\quad + \left(1 - \mathbb{P}\left[C_1 \subseteq Y\right] \cdot \mathbb{P}[\omega \in C_j] - \mathbb{P}[(C_j)^c \subseteq Y]\left(1 - \mathbb{P}\left[\omega \in C_1\right]\right)\right) \\
&\quad \cdot \mathbb{P}\left[\bigcup_{C \in (\mathcal{C}_\pm^{l-1} \setminus \{C_1 \neg C_1\})} \left((C \subseteq Y) \cap (\omega \in C)\right)\right] \\
&= \mathbb{P}\left[C_1 \subseteq Y\right] \cdot \mathbb{P}\left[\omega \in C_1\right] + \mathbb{P}\left[(C_1)^c \subseteq Y\right]\left(1 - \mathbb{P}\left[\omega \in C_1\right]\right) \\
&\quad + \left(1 - \mathbb{P}\left[C_1 \subseteq Y\right] \cdot \mathbb{P}[\omega \in C_j] - \mathbb{P}[(C_j)^c \subseteq Y]\left(1 - \mathbb{P}\left[\omega \in C_1\right]\right)\right) \\
&\quad \cdot \left(1 - \prod_{j=2}^{n^{l-1}} \left(1 - \mathbb{P}\left[C_j \subseteq Y\right] \cdot \mathbb{P}[\omega \in C_j] - \mathbb{P}[(C_j)^c \subseteq Y] \cdot \left(1 - \mathbb{P}\left[\omega \in C_j\right]\right)\right)\right) \\
&= \mathbb{P}\left[C_1 \subseteq Y\right] \cdot \mathbb{P}\left[\omega \in C_1\right] + \mathbb{P}\left[(C_1)^c \subseteq Y\right]\left(1 - \mathbb{P}\left[\omega \in C_1\right]\right) \\
&\quad + \left(1 - \mathbb{P}\left[C_1 \subseteq Y\right] \cdot \mathbb{P}[\omega \in C_j] - \mathbb{P}[(C_j)^c \subseteq Y]\left(1 - \mathbb{P}\left[\omega \in C_1\right]\right)\right) \\
&\quad - \prod_{j=1}^{n^{l-1}} \left(1 - \mathbb{P}\left[C_j \subseteq Y\right] \cdot \mathbb{P}[\omega \in C_j] - \mathbb{P}[(C_j)^c \subseteq Y] \cdot \left(1 - \mathbb{P}\left[\omega \in C_j\right]\right)\right) \\
&= 1 - \prod_{j=1}^{n^{l-1}} \left(1 - \mathbb{P}\left[C_j \subseteq Y\right] \cdot \mathbb{P}[\omega \in C_j] - \mathbb{P}[(C_j)^c \subseteq Y] \cdot \left(1 - \mathbb{P}\left[\omega \in C_j\right]\right)\right).
\end{aligned}
$$

### B.1.3 Derivation without the 3$^{\text{rd}}$ assumption

The third assumption of conditional independence between concepts in the same layer is necessary for both AND and OR nodes to obtain an easily computable solution at the last step when we need, for AND,

$$
\begin{aligned}
\mathbb{P}\left[\bigcap_{C \in (\mathcal{C}_\pm^{l-1} \setminus \{C_1, \neg C_1\})} \left((C \not\supseteq Y) \cup (\omega \in C)\right)\right] &= \mathbb{P}\left[\bigcap_{C \in (\mathcal{C}_\pm^{l-1} \setminus \{C_1, \neg C_1\})} \left((C \not\supseteq Y) \cup (\omega \in C)\right) \,\middle|\, \omega \in C_1, C_1 \supseteq Y\right] \\
&= \mathbb{P}\left[\bigcap_{C \in (\mathcal{C}_\pm^{l-1} \setminus \{C_1, \neg C_1\})} \left((C \not\supseteq Y) \cup (\omega \in C)\right) \,\middle|\, \omega \in (C_1)^c, (C_1)^c \supseteq Y\right] \\
&= \mathbb{P}\left[\bigcap_{C \in (\mathcal{C}_\pm^{l-1} \setminus \{C_1, \neg C_1\})} \left((C \not\supseteq Y) \cup (\omega \in C)\right) \,\middle|\, (C_1)^c \not\supseteq Y, C_1 \not\supseteq Y\right]
\end{aligned}
$$

and, for OR,

$$\mathbb{P}\left[\bigcup_{C\in(\mathcal{C}_{\pm}^{l-1}\setminus\{C_1,\neg C_1\})}\Big((C\subseteq Y)\cap(\omega\in C)\Big)\right]=\mathbb{P}\left[\bigcup_{C\in(\mathcal{C}_{\pm}^{l-1}\setminus\{C_1,\neg C_1\})}\Big((C\subseteq Y)\cap(\omega\in C)\Big)\,\middle|\,\omega\in(C_1)^c,C_1\subseteq Y\right]$$

$$=\mathbb{P}\left[\bigcup_{C\in(\mathcal{C}_{\pm}^{l-1}\setminus\{C_1,\neg C_1\})}\Big((C\subseteq Y)\cap(\omega\in C)\Big)\,\middle|\,\omega\in C_1,(C_1)^c\subseteq Y\right]$$

$$=\mathbb{P}\left[\bigcup_{C\in(\mathcal{C}_{\pm}^{l-1}\setminus\{C_1,\neg C_1\})}\Big((C\subseteq Y)\cap(\omega\in C)\Big)\,\middle|\,(C_1)^c\not\subseteq Y,C_1\not\subseteq Y\right].$$

Given the other assumptions of independence between aleatoric and epistemic probabilities as well as between all epistemic probabilities in the same node, what remains is, for AND,

$$\mathbb{P}\left[\bigcap_{C\in(\mathcal{C}_{\pm}^{l-1}\setminus\{C_1\neg C_1\})}\Big((C\not\supseteq Y)\cup(\omega\in C)\Big)\right]=\mathbb{P}\left[\bigcap_{C\in(\mathcal{C}_{\pm}^{l-1}\setminus\{C_1\neg C_1\})}\Big((C\not\supseteq Y)\cup(\omega\in C)\Big)\,\middle|\,\omega\in C_1\right]$$

and, for OR,

$$\mathbb{P}\left[\bigcup_{C\in(\mathcal{C}_{\pm}^{l-1}\setminus\{C_1\neg C_1\})}\Big((C\subseteq Y)\cap(\omega\in C)\Big)\right]=\mathbb{P}\left[\bigcup_{C\in(\mathcal{C}_{\pm}^{l-1}\setminus\{C_1\neg C_1\})}\Big((C\subseteq Y)\cap(\omega\in C)\Big)\,\middle|\,\omega\in C_1\right].$$

To conclude, we need the missing assumption of conditional independence between all concepts in the same layer

$$\omega\in C_i^l\perp\!\!\!\perp\omega\in C_j^l\ \Big|\ x.$$

This assumption seems improbable to say the least since, for $l>0$, the concepts $C_i^l$ and $C_j^l$ depend in general on the same upstream concepts, *i.e.* on the concepts from the previous layers. However, if we condition on the concepts of the preceding layer $l-1$, the concepts in layer $l$ become independent by d-separation

$$\omega\in C_i^l\perp\!\!\!\perp\omega\in C_j^l\ \Big|\ \mathrm{C}^{l-1}(\omega)\,,$$

where we introduce the notation $\mathrm{C}^{l-1}(\omega)=\mathbb{1}\big(\omega\in C^{l-1}\big)\in\{0,1\}^{n^l}$ to represent the vector of indicator binary random variables $\mathrm{C}_i^{l-1}(\omega)=\mathbb{1}\big(\omega\in C_i^{l-1}\big)$ such that the random event $\omega\in C_i^{l-1}$ is equal to $\mathrm{C}_i^{l-1}(\omega)=1$ and its complement $\omega\notin C_i^{l-1}$ is equal to $\mathrm{C}_i^{l-1}(\omega)=0$. The conditional probabilities $\mathbb{P}\left[\omega\in C_j^l\,\middle|\,\mathrm{C}^{l-1}(\omega)=\mathrm{c}^{l-1}\right]$ given the previous layer for both AND and OR nodes can be easily computed with

$$\mathbb{P}\left[\omega\in C_j^l\,\middle|\,\mathrm{C}^{l-1}=\mathrm{c}^{l-1}\right]=a_j^l\prod_{i=1}^{n^{l-1}}\Big(1-[A_{i,j}^l]_+\big(1-c_i^{l-1}\big)\Big)\Big(1-[-A_{i,j}^l]_+\,c_i^{l-1}\Big),$$

for AND and

$$\mathbb{P}\left[\omega\in C_j^l\,\middle|\,\mathrm{C}^{l-1}(\omega)=\mathrm{c}^{l-1}\right]=1-\big(1-o_j^l\big)\prod_{i=1}^{n^{l-1}}\Big(1-[O_{i,j}^l]_+\,c_i^{l-1}\Big)\Big(1-[-O_{i,j}^l]_+\big(1-c_i^{l-1}\big)\Big),$$

for OR. These are the same formulas as when there is full independence and we derive them in the same way. Moreover, because of their conditional independence, their joint conditional probability is given by, $\forall(\mathrm{c}^{l-1},\mathrm{c}^l)\in\{0,1\}^{n^{l-1}\times n^l}$,

$$\mathbb{P}\left[\mathrm{C}^l(\omega)=\mathrm{c}^l\,\middle|\,\mathrm{C}^{l-1}(\omega)=\mathrm{c}^{l-1}\right]=\prod_{i=1}^{n^l}\Big(\mathrm{c}_i^l\cdot\mathbb{P}\left[\omega\in C_i^l\,\middle|\,\mathrm{C}^{l-1}=\mathrm{c}^{l-1}\right]+\big(1-\mathrm{c}_i^l\big)\big(1-\mathbb{P}\left[\omega\in C_i^l\,\middle|\,\mathrm{C}^{l-1}=\mathrm{c}^{l-1}\right]\big)\Big).$$

The input layer 0 has no preceding layer, but it does satisfy

$$x \in C_i^0 \perp\!\!\!\perp x \in C_j^0 \mid x.$$

Its joint probability is thus given by

$$\mathbb{P}[C^0(x) = c^0] = \prod_{i=1}^{n^0} \left( c_i^0 \cdot \mathbb{P}[x \in C_i^0] + \left(1 - c_i^0\right)\left(1 - \mathbb{P}[x \in C_i^0]\right) \right), \qquad \forall c^0 \in \{0,1\}^{n^0}$$

where the probabilities $\mathbb{P}[x \in C_i^0]$ are given.

Armed with these independences, we can take into account the dependences between concepts that are defined partially on the same concepts. We compute the probabilities layer by layer starting with the first logical layer. We can compute their joint probability with

$$\mathbb{P}[C^l(\omega) = c^l] = \sum_{c^{l-1} \in \{0,1\}^{n^{l-1}}} \mathbb{P}[C^{l-1}(\omega) = c^{l-1}] \cdot \mathbb{P}\left[C^l(\omega) = c^l \mid C^{l-1}(\omega) = c^{l-1}\right]$$

and their marginal probabilities with

$$\mathbb{P}[\omega \in C_j^l] = \sum_{c^{l-1} \in \{0,1\}^{n^{l-1}}} \mathbb{P}[C^{l-1}(\omega) = c^{l-1}] \cdot \mathbb{P}\left[\omega \in C_j^l \mid C^{l-1}(\omega) = c^{l-1}\right].$$

The issue with this approach is that we need full joint probability distributions $\mathbb{P}\left[C^l(\omega) = c^l \mid C^{l-1} = c^{l-1}\right]$, for every pair of joint values $(c^{l-1}, c^l) \in \{0,1\}^{n^{l-1} \times n^l}$, for every layer. For each layer, a tensor of $2^{n^{l-1}+n^l}$ entries taking values in $[0,1]$ would be needed for inference and would have to be re-computed after every learning step. This combinatorial explosion results in a exponential number of computations and memory that is only viable in applications with a very small number of nodes per layer.

Moreover, based on preliminary test results, it seems that this extended formulation does not improve the modeling ability of the framework meaningfully. We tested both formulations on randomly generated data that follows the assumptions of our model. We sampled random logical networks of 10 conjunction layers with

- inputs modeled by independent Bernouillis with parameters uniformly sampled in $(0,1)$,

- weights between layers of either $A_{i,j}^l = 0$ with probability $1/2$ or $A_{i,j}^l \in \{1,-1\}$ with probability $1/4$ each,

- and unobserved concepts also modeled by independent Bernouillis of parameter $a_j^l$ uniformly sampled in $(0,1)$.

We additionally added weights of either $+1$ or $-1$ when a concept node had only incoming weights of 0 (*i.e.* an empty definition) or when a node had only outgoing weights of 0 (*i.e.* was unused in the next layer). We assumed a fixed width for each network that had the same number of inputs, outputs and concepts in all layers. We considered widths between 2 and 6 and, for each width, we sampled 30 such generating models, which produced datasets of 1000 points for each. We then measured the $L_2$ loss of both of our frameworks on these datasets. We computed this loss between our frameworks and the data points for all concepts at all depths, from layer $l = 1$ to $l = L = 10$. The results are given in Figure B.2 where the independent inputs are labeled "min" (for minimal independence hypothesis) and independent concepts in all layers are labeled "max" (for maximal independence hypothesis). For $l = 1$, the two frameworks are equivalent and their results are the same. Only for bigger depths do we start to see a small difference in performance, although it seems negligible in this preliminary testing.

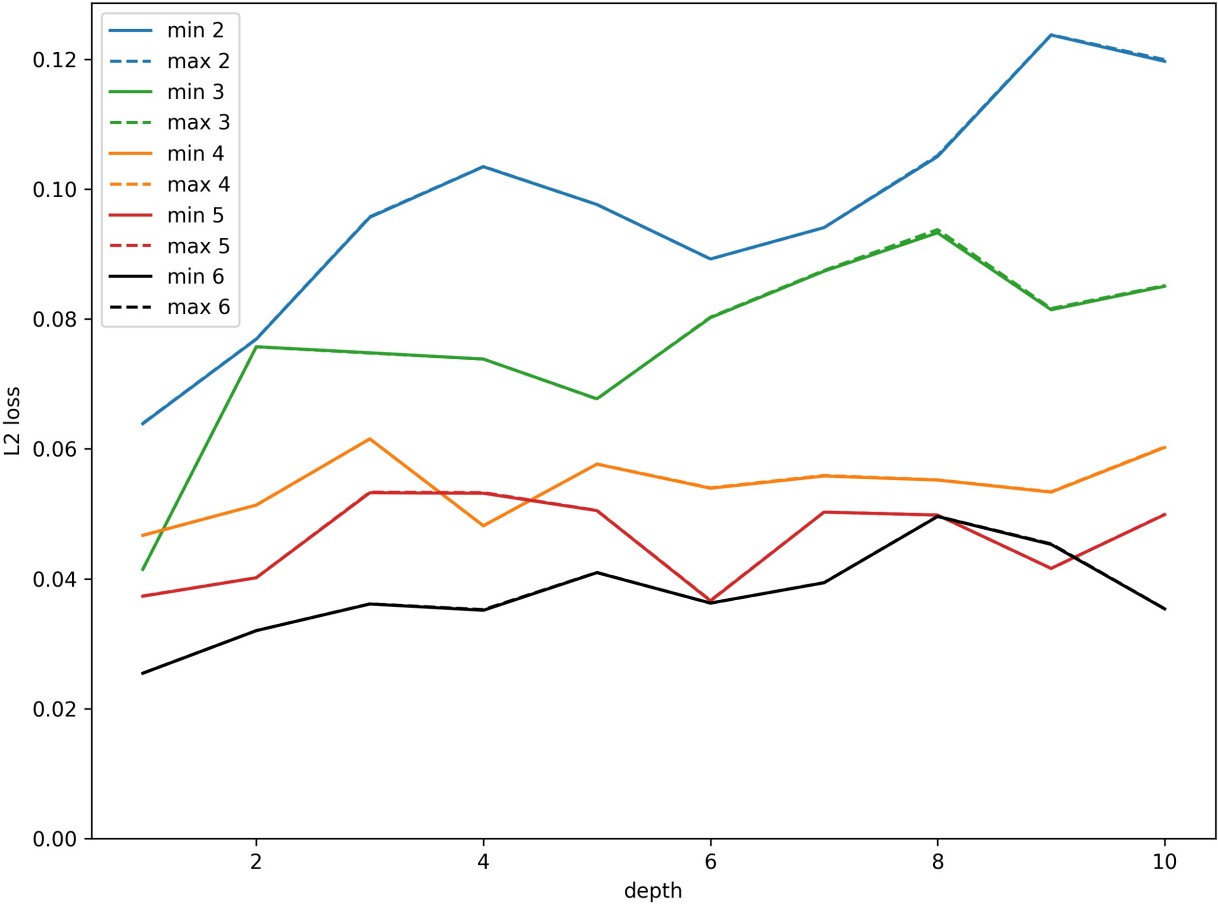

Figure B.2: Preliminary comparison of independence hypotheses

## B.2 Logical modeling

### B.2.1 De Morgan's laws for the AND/OR concepts

**Between (P-AND) and (P-OR)** Since (F-AND) and (F-OR) are equivalent rewritings and since product fuzzy logic's t-norm $\overset{\text{P}}{\wedge}$ and t-conorm $\overset{\text{P}}{\vee}$ follow de Morgan's laws with strong negation $\overset{\text{P}}{\neg}$ (van Krieken et al., 2022), it is easier to show the De Morgan's laws between (F-AND) and (F-OR).

Let $C_i^l$ be an AND concept with

$$c_i^l(x) = a_i^l \overset{\text{P}}{\wedge} \overset{\text{P}}{\underset{j \in \{1,\dots,n^{l-1}\}}{\bigwedge}} \left( \overset{\text{P}}{\neg} [A_{i,j}^l]_+ \overset{\text{P}}{\vee} c_j^{l-1}(x) \right) \overset{\text{P}}{\wedge} \left( \overset{\text{P}}{\neg} [A_{i,j}^l]_- \overset{\text{P}}{\vee} \overset{\text{P}}{\neg} c_j^{l-1}(x) \right).$$

We want to show that its opposite is an OR concept $C_{i'}^l$ with

$$c_{i'}^l(x) = o_{i'}^l \overset{\text{P}}{\vee} \overset{\text{P}}{\underset{j \in \{1,\dots,n^{l-1}\}}{\bigvee}} \left( [O_{i',j}^l]_+ \overset{\text{P}}{\wedge} c_j^{l-1}(x) \right) \overset{\text{P}}{\vee} \left( [O_{i',j}^l]_- \overset{\text{P}}{\wedge} \overset{\text{P}}{\neg} c_j^{l-1}(x) \right).$$

We begin by taking the opposite of the AND concept $C_i^l$.

$$\overset{\text{P}}{\neg} c_i^l(x) = \overset{\text{P}}{\neg} \left( a_i^l \overset{\text{P}}{\wedge} \overset{\text{P}}{\underset{j \in \{1,\dots,n^{l-1}\}}{\bigwedge}} \left( \overset{\text{P}}{\neg} [A_{i,j}^l]_+ \overset{\text{P}}{\vee} c_j^{l-1}(x) \right) \overset{\text{P}}{\wedge} \left( \overset{\text{P}}{\neg} [A_{i,j}^l]_- \overset{\text{P}}{\vee} \overset{\text{P}}{\neg} c_j^{l-1}(x) \right) \right)$$

$$= \overset{\text{P}}{\neg} a_i^l \overset{\text{P}}{\vee} \overset{\text{P}}{\underset{j \in \{1,\dots,n^{l-1}\}}{\bigvee}} \overset{\text{P}}{\neg} \left( \overset{\text{P}}{\neg} [A_{i,j}^l]_+ \overset{\text{P}}{\vee} c_j^{l-1}(x) \right) \overset{\text{P}}{\vee} \overset{\text{P}}{\neg} \left( \overset{\text{P}}{\neg} [A_{i,j}^l]_- \overset{\text{P}}{\vee} \overset{\text{P}}{\neg} c_j^{l-1}(x) \right)$$

$$= \overset{\text{P}}{\neg} a_i^l \overset{\text{P}}{\vee} \overset{\text{P}}{\underset{j \in \{1,\dots,n^{l-1}\}}{\bigvee}} \left( [A_{i,j}^l]_+ \overset{\text{P}}{\wedge} \overset{\text{P}}{\neg} c_j^{l-1}(x) \right) \overset{\text{P}}{\vee} \left( [A_{i,j}^l]_- \overset{\text{P}}{\wedge} c_j^{l-1}(x) \right)$$

By identification, we can see that

$$o_{i'}^l = \overset{\text{P}}{\neg} a_i^l = 1 - a_i^l, \qquad \text{and} \qquad O_{i',j}^l = -A_{i,j}^l,$$

*i.e.* and AND concept can be converted to an OR concept, and vice versa, by taking the complement of its bias and flipping the signs of its incoming and outgoing weights.

**Between (D-AND) and (D-OR)** Since there is a one-to-one translation from (D-AND) and (D-OR) to (L-AND) and (L-OR) and since both intersection ($\cap$) and union ($\cup$) as well as conjunction ($\wedge$) and disjunction ($\vee$) follow De Morgan's laws, it is enough to show that De Morgan's laws hold between (D-AND) and (D-OR) to have the same between (L-AND) and (L-OR).

Let $C_i^l$ be an AND concept with

$$\omega \in C_i^l = (u \in \tilde{N}_i^l) \cap \bigcap_{C \in \mathcal{C}_\pm^{l-1}} \left( \left( C \supseteq C_i^l \right)^c \cup \left( \omega \in C \right) \right),$$

We want to show that its opposite is an OR concept $C_{i'}^l$ with

$$\omega \in C_{i'}^l = (u \in \tilde{S}_{i'}^l) \cup \bigcup_{C \in \mathcal{C}_\pm^{l-1}} \left( \left( C \subseteq C_{i'}^l \right) \cap \left( \omega \in C \right) \right),$$

We begin by taking the opposite of the AND concept $C_i^l$.

$$
\begin{aligned}
\omega \in (C_i^l)^c = (\omega \in C_i^l)^c &= \left( (u \in \tilde{N}_i^l) \cap \bigcap_{C \in \mathcal{C}_{\pm}^{l-1}} \left( (C \supseteq C_i^l)^c \cup (\omega \in C) \right) \right)^c \\
&= (u \in \tilde{N}_i^l)^c \cup \bigcup_{C \in \mathcal{C}_{\pm}^{l-1}} \left( (C \supseteq C_i^l)^c \cup (\omega \in C) \right)^c \\
&= (u \in \tilde{N}_i^l)^c \cup \bigcup_{C \in \mathcal{C}_{\pm}^{l-1}} \left( (C \supseteq C_i^l) \cap (\omega \in C)^c \right) \\
&= (u \in (\tilde{N}_i^l)^c) \cup \bigcup_{C \in \mathcal{C}_{\pm}^{l-1}} \left( (C \supseteq C_i^l) \cap (\omega \in (C)^c) \right) \\
&= (u \in (\tilde{N}_i^l)^c) \cup \bigcup_{C' \in \mathcal{C}_{\pm}^{l-1}} \left( ((C')^c \supseteq C_i^l) \cap (\omega \in C') \right) \\
&= (u \in (\tilde{N}_i^l)^c) \cup \bigcup_{C' \in \mathcal{C}_{\pm}^{l-1}} \left( (C' \subseteq (C_i^l)^c) \cap (\omega \in C') \right) \\
&= (u \in (\tilde{N}_i^l)^c) \cup \bigcup_{C' \in \mathcal{C}_{\pm}^{l-1}} \left( (C' \subseteq C_{i'}^l) \cap (\omega \in C') \right)
\end{aligned}
$$

By identification, we can see that

$$
\tilde{S}_{i'}^l = (\tilde{N}_i^l)^c,
$$

and that a concept $C$ in the previous layer is a sufficient concept $C \subseteq C_{i'}^l$ of OR concept $C_{i'}^l$ whenever its opposite $(C)^c$ was a necessary concept $(C)^c \supseteq C_i^l$ of AND concept $C_i^l$.

### B.3 Interpretation

Since any finite combination of AND (resp. OR) concepts can be represented by a single AND (resp. OR) concept, each type of concept can represent an infinite number of cases. We give some intuitive and counter-intuitive examples below. Furthermore, in any such case, the missing necessary (resp. sufficient) concepts that are needed to determine the presence of the target concept $\omega \in C_i^l$ can all be absorbed into the unobserved concepts $u \in \tilde{N}_i^l$ (resp. $u \in \tilde{S}_i^l$), through the probability $a_i^l$ (resp. $o_i^l$). Moreover, this multiplicity of cases is exacerbated by the fact that a logical formula can be rewritten in many equivalent ways through De Morgan's laws and distributivity. This makes the interpretation of a NLN very difficult without additional expert knowledge. However, the equivalence of AND/OR concepts through De Morgan's laws can also be used as an advantage in the learning of deep NLNs. A layer of concepts using negation that is followed by another layer allowing negation can be learned with an arbitrary type and then be interpreted a posteriori, once the learning is done, by experts.

In the following figures of causal structures, the circles are concepts, the arrows are cause-to-consequence relations (implications) and the bracket signifies a conjunction (AND) of concepts. The whites circles are the necessary concepts, the black circle is the target concept and the gray circles are concepts that are not needed to determine the target's presence if given the white circles.

#### B.3.1 Examples of causal structures that can be represented by an AND node

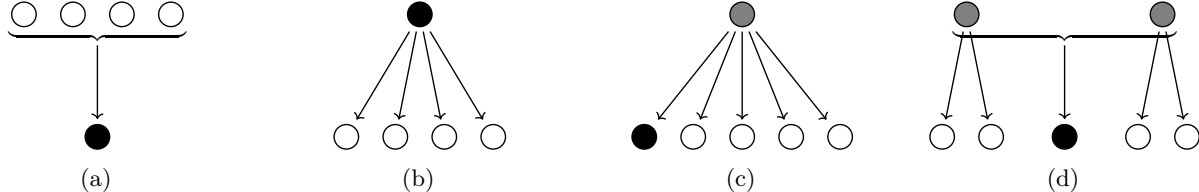

Figure B.3: Examples of causal structures that can be represented by an AND node

An AND node can represent a consequence of necessary causal ingredients (Figure B.3a). The same AND node can represent the opposite direction of causality, where the AND concept is the cause and the necessary concepts are its consequences (Figure B.3b). In this case, the unobserved concepts probability $a_i^l$ quantifies how often this common cause is what caused these consequences, when they are all present simultaneously. A similar case is when the AND node represents another one of these consequences from the same common cause (Figure B.3c). An AND node can even represent a consequence of causal ingredients which, themselves produce their own individual necessary consequences (Figure B.3d).

#### B.3.2 Examples of causal structures that can be represented by an OR node



Figure B.4: Examples of causal structures that can be represented by an OR node

An OR node can represent a consequence of some sufficient concepts (Figure B.4a). The same OR node can represent the opposite direction of causality, where the OR concept is a causal ingredient and the sufficient concepts are its possible consequences (Figure B.4b). In this case, the unobserved sufficient concepts probability $o_i^l$ quantifies how often this common causal ingredient is still present even when none of these consequences are present. A similar case is when the OR node represents a direct consequence of this common

causal ingredient (Figure B.4c). An OR node can even represent a consequence of causes which, together with other causal ingredients produce their own consequences (Figure B.4d).

## C  Machine Learning pipeline

### C.1  Post-processing

#### C.1.1  Other weight quantizing algorithms

---

**For** each layer $l \in \{1, .., L\}$, starting from the last layer $L$ **,**
    Quantize all weights in the layer to their sign $A^l_{\cdot,\cdot} = \text{sign}(A^l_{\cdot,\cdot})$ (resp. $O^l_{i,j} = \text{sign}\left(O^l_{i,j}\right)$) and keep
    original values in $\tilde{A}^l_{\cdot,\cdot}$ (resp. $\tilde{O}^l_{\cdot,\cdot}$).
    Compute the new best loss $e^*$.
    **For** each weight $A^l_{i,j}$ (resp. $O^l_{i,j}$), in increasing likeliness $\left|\tilde{A}^l_{i,j}\right|$ (resp. $\left|\tilde{O}^l_{i,j}\right|$) **,**
        **If** it is non-zero **,**
            Compare the loss when we prune the weight $A^l_{i,j} = 0$ (resp. $O^l_{i,j} = 0$) to the best loss $e^*$.
            Commit to the pruning iff the loss is decreased and update the best loss $e^*$ in that case.

Do the same for the category and continuous input modules, one at a time.

---

Algorithm C.1: Subtractive quantizing algorithm

---

**For** each layer $l \in \{1, .., L\}$, starting from the last layer $L$ **,**
    Prune all weights in the layer $A^l_{\cdot,\cdot} = 0$ (resp. $O^l_{i,j} = 0$) and keep original values in $\tilde{A}^l_{\cdot,\cdot}$ (resp. $\tilde{O}^l_{\cdot,\cdot}$).
    Compute the new best loss $e^*$.
    **For** each weight $A^l_{i,j}$ (resp. $O^l_{i,j}$), in decreasing likeliness $\left|\tilde{A}^l_{i,j}\right|$ (resp. $\left|\tilde{O}^l_{i,j}\right|$) **,**
        **If** it is non-zero **,**
            Compare the loss when we fix $A^l_{i,j} = \text{sign}(\tilde{A}^l_{i,j})$ (resp. $O^l_{i,j} = \text{sign}(\tilde{O}^l_{i,j})$) to the best loss $e^*$.
            Commit to the new value iff the loss is decreased and update the best loss $e^*$ in that case.

Do the same for the category and continuous input modules, one at a time.

---

Algorithm C.2: Additive quantizing algorithm

---

**For** each layer $l \in \{1, .., L\}$, starting from the last layer $L$ **,**
    **For** each weight $A^l_{i,j}$ (resp. $O^l_{i,j}$), in increasing likeliness $\left|A^l_{i,j}\right|$ (resp. $\left|O^l_{i,j}\right|$) **,**
        **If** it is non-zero **,**
            Compare the loss when we fix $A^l_{i,j} \in \left\{0, \text{sign}\left(A^l_{i,j}\right)\right\}$ (resp. $O^l_{i,j} \in \left\{0, \text{sign}\left(O^l_{i,j}\right)\right\}$).
            Commit to the best quantized value.

Do the same for the category and continuous input modules, one at a time.

---

Algorithm C.3: Ascending selection quantizing algorithm

# D    Experiments

## D.1    Boolean networks

### D.1.1    Examples of discovered logic programs

In this first example, on the mammalian dataset, the ground-truth logic program is discovered fully, except for one redundant rule that is subsumed by the disjunction of two other ground-truth rules.

| Ground-truth logic program | Discovered logic program |
|---|---|
| $A_1^t \rightarrow A_1^{t+1},$ | $A_1^t \rightarrow A_1^{t+1},$ |
| $\neg A_3^t \wedge A_4^t \rightarrow A_2^{t+1},$ | $\neg A_3^t \wedge A_4^t \rightarrow A_2^{t+1},$ |
| $\neg A_1^t \wedge A_6^t \wedge \neg A_{10}^t \rightarrow A_3^{t+1},$ | $\neg A_1^t \wedge A_6^t \wedge \neg A_{10}^t \rightarrow A_3^{t+1},$ |
| $\neg A_1^t \wedge \neg A_2^t \wedge \neg A_5^t \wedge \neg A_{10}^t \rightarrow A_3^{t+1},$ | $\neg A_1^t \wedge \neg A_2^t \wedge \neg A_5^t \wedge \neg A_{10}^t \rightarrow A_3^{t+1},$ |
| $\neg A_3^t \wedge A_6^t \wedge \neg A_{10}^t \rightarrow A_4^{t+1},$ | $\neg A_3^t \wedge A_6^t \wedge \neg A_{10}^t \rightarrow A_4^{t+1},$ |
| $\neg A_3^t \wedge \neg A_5^t \wedge \neg A_{10}^t \rightarrow A_4^{t+1},$ | $\neg A_3^t \wedge \neg A_5^t \wedge \neg A_{10}^t \rightarrow A_4^{t+1},$ |
| $\neg A_3^t \wedge A_5^t \wedge \neg A_7^t \wedge \neg A_8^t \rightarrow A_5^{t+1},$ | $\neg A_3^t \wedge A_5^t \wedge \neg A_7^t \wedge \neg A_8^t \rightarrow A_5^{t+1},$ |
| $\neg A_3^t \wedge A_4^t \wedge \neg A_7^t \wedge \neg A_9^t \rightarrow A_5^{t+1},$ | $\neg A_3^t \wedge A_4^t \wedge \neg A_7^t \wedge \neg A_9^t \rightarrow A_5^{t+1},$ |
| $\neg A_3^t \wedge A_5^t \wedge \neg A_7^t \wedge \neg A_9^t \rightarrow A_5^{t+1},$ | $\neg A_3^t \wedge A_5^t \wedge \neg A_7^t \wedge \neg A_9^t \rightarrow A_5^{t+1},$ |
| $\neg A_3^t \wedge A_4^t \wedge \neg A_7^t \wedge \neg A_8^t \rightarrow A_5^{t+1},$ | $\neg A_3^t \wedge A_4^t \wedge \neg A_7^t \wedge \neg A_8^t \rightarrow A_5^{t+1},$ |
| $\neg A_1^t \wedge \neg A_2^t \wedge \neg A_5^t \wedge \neg A_{10}^t \rightarrow A_6^{t+1},$ | $\neg A_1^t \wedge \neg A_2^t \wedge \neg A_5^t \wedge \neg A_{10}^t \rightarrow A_6^{t+1},$ |
| $\neg A_1^t \wedge \neg A_5^t \wedge A_6^t \wedge \neg A_{10}^t \rightarrow A_6^{t+1},$ | $\neg A_1^t \wedge \neg A_5^t \wedge A_6^t \wedge \neg A_{10}^t \rightarrow A_6^{t+1},$ |
| $\neg A_1^t \wedge \neg A_2^t \wedge A_6^t \wedge \neg A_{10}^t \rightarrow A_6^{t+1},$ | $\neg A_1^t \wedge \neg A_2^t \wedge A_6^t \wedge \neg A_{10}^t \rightarrow A_6^{t+1},$ |
| $A_{10}^t \rightarrow A_7^{t+1},$ | $A_{10}^t \rightarrow A_7^{t+1},$ |
| $\neg A_9^t \rightarrow A_8^{t+1},$ | $\neg A_9^t \rightarrow A_8^{t+1},$ |
| $A_7^t \wedge A_8^t \rightarrow A_8^{t+1},$ | $A_7^t \wedge A_8^t \rightarrow A_8^{t+1},$ |
| $A_8^t \wedge A_{10}^t \rightarrow A_8^{t+1},$ | $A_8^t \wedge A_{10}^t \rightarrow A_8^{t+1},$ |
| $A_5^t \wedge A_8^t \rightarrow A_8^{t+1},$ | $A_5^t \wedge A_8^t \rightarrow A_8^{t+1},$ |
| $A_7^t \rightarrow A_9^{t+1},$ | $A_7^t \rightarrow A_9^{t+1},$ |
| $\neg A_5^t \wedge \neg A_{10}^t \rightarrow A_9^{t+1},$ | $\neg A_5^t \wedge \neg A_{10}^t \rightarrow A_9^{t+1},$ |
| $A_6^t \wedge A_{10}^t \rightarrow A_9^{t+1},$ | $A_6^t \wedge A_{10}^t \rightarrow A_9^{t+1},$ |
| $\neg A_5^t \wedge A_6^t \rightarrow A_9^{t+1},$ | |
| $\neg A_7^t \wedge \neg A_9^t \rightarrow A_{10}^{t+1},$ | $\neg A_7^t \wedge \neg A_9^t \rightarrow A_{10}^{t+1},$ |

where $\neg A_5^t \wedge A_6^t \rightarrow A_9^{t+1}$ is subsumed by the disjunction of $\neg A_5^t \wedge \neg A_{10}^t \rightarrow A_9^{t+1}$ and $A_6^t \wedge A_{10}^t \rightarrow A_9^{t+1}$, *i.e.* whenever the first rule should be activated, either the second or the third rule is activated, thus making the first rule redundant.

In this second example, on the arabidopsis dataset, the ground-truth logic program is discovered fully, except for one rule, which was discovered as two rules that imply it by their resolution.

| Ground-truth logic program | Discovered logic program |
|---|---|

$$A_2^t \wedge A_7^t \to A_1^{t+1},$$
$$A_1^t \wedge A_5^t \wedge A_{14}^t \wedge A_{15}^t \to A_1^{t+1},$$
$$A_1^t \wedge A_{10}^t \wedge A_{14}^t \wedge A_{15}^t \to A_1^{t+1},$$
$$A_2^t \to A_2^{t+1},$$
$$\neg A_5^t \wedge \neg A_{13}^t \to A_3^{t+1},$$
$$\neg A_6^t \to A_4^{t+1},$$
$$A_4^t \wedge \neg A_{10}^t \to A_5^{t+1},$$
$$\neg A_{10}^t \wedge \neg A_{13}^t \to A_5^{t+1},$$
$$A_7^t \wedge \neg A_{10}^t \to A_5^{t+1},$$
$$\neg A_7^t \to A_6^{t+1},$$
$$\neg A_6^t \to A_7^{t+1},$$
$$\neg A_{13}^t \to A_7^{t+1},$$
$$\neg A_{13}^t \to A_8^{t+1},$$
$$A_9^t \wedge \neg A_{15}^t \to A_9^{t+1},$$
$$A_9^t \wedge \neg A_{10}^t \to A_9^{t+1},$$
$$\neg A_5^t \wedge A_7^t \to A_{10}^{t+1},$$
$$A_7^t \wedge \neg A_{11}^t \to A_{10}^{t+1},$$
$$A_7^t \wedge A_{10}^t \wedge A_{15}^t \to A_{10}^{t+1},$$
$$\neg A_8^t \wedge \neg A_{13}^t \to A_{10}^{t+1},$$
$$A_7^t \wedge \neg A_8^t \to A_{10}^{t+1},$$
$$A_7^t \wedge A_9^t \to A_{10}^{t+1},$$
$$A_7^t \wedge \neg A_{12}^t \to A_{10}^{t+1},$$
$$\neg A_5^t \wedge A_6^t \wedge \neg A_7^t \to A_{13}^{t+1},$$
$$A_1^t \wedge A_5^t \wedge A_{14}^t \wedge A_{15}^t \to A_{14}^{t+1},$$
$$A_1^t \wedge A_{10}^t \wedge A_{14}^t \wedge A_{15}^t \to A_{14}^{t+1},$$
$$A_1^t \wedge A_7^t \to A_{14}^{t+1},$$
$$A_7^t \wedge A_{10}^t \to A_{14}^{t+1},$$
$$A_7^t \to A_{15}^{t+1},$$

Discovered:
$$A_2^t \wedge A_7^t \to A_1^{t+1},$$
$$A_1^t \wedge A_5^t \wedge A_{14}^t \wedge A_{15}^t \to A_1^{t+1},$$
$$A_1^t \wedge A_{10}^t \wedge A_{14}^t \wedge A_{15}^t \to A_1^{t+1},$$
$$A_2^t \to A_2^{t+1},$$
$$\neg A_5^t \wedge \neg A_{13}^t \to A_3^{t+1},$$
$$\neg A_6^t \to A_4^{t+1},$$
$$A_4^t \wedge \neg A_{10}^t \to A_5^{t+1},$$
$$\neg A_{10}^t \wedge \neg A_{13}^t \to A_5^{t+1},$$
$$A_7^t \wedge \neg A_{10}^t \to A_5^{t+1},$$
$$\neg A_7^t \to A_6^{t+1},$$
$$\neg A_6^t \to A_7^{t+1},$$
$$\neg A_{13}^t \to A_7^{t+1},$$
$$\neg A_{13}^t \to A_8^{t+1},$$
$$A_9^t \wedge \neg A_{15}^t \to A_9^{t+1},$$
$$A_9^t \wedge \neg A_{10}^t \to A_9^{t+1},$$
$$\neg A_5^t \wedge A_7^t \to A_{10}^{t+1},$$
$$A_7^t \wedge \neg A_{11}^t \to A_{10}^{t+1},$$
$$A_7^t \wedge A_{10}^t \wedge A_{15}^t \to A_{10}^{t+1},$$
$$\neg A_8^t \wedge \neg A_{13}^t \to A_{10}^{t+1},$$
$$A_7^t \wedge \neg A_8^t \to A_{10}^{t+1},$$
$$A_7^t \wedge A_9^t \to A_{10}^{t+1},$$
$$A_7^t \wedge \neg A_{12}^t \to A_{10}^{t+1},$$
$$\neg A_5^t \wedge A_6^t \wedge \neg A_7^t \to A_{13}^{t+1},$$
$$A_1^t \wedge A_5^t \wedge A_{14}^t \wedge A_{15}^t \to A_{14}^{t+1},$$
$$A_1^t \wedge A_{10}^t \wedge A_{14}^t \wedge A_{15}^t \to A_{14}^{t+1},$$
$$A_1^t \wedge A_7^t \to A_{14}^{t+1},$$
$$A_7^t \wedge A_{10}^t \to A_{14}^{t+1},$$
$$A_7^t \wedge \neg A_{10}^t \to A_{15}^{t+1},$$
$$A_7^t \wedge A_{10}^t \to A_{15}^{t+1},$$

where $A_7^t \to A_{15}^{t+1}$ is implied by the resolution of $A_7^t \wedge \neg A_{10}^t \to A_{15}^{t+1}$ and $A_7^t \wedge A_{10}^t \to A_{15}^{t+1}$, *i.e.* the first rule is a direct consequence of the second and third rules.

In this third example, on the budding dataset, we again have redundant ground-truth rules which are subsumed by the disjunction of two other discovered rules. These have no impact on the predictive accuracy of the model. In this case however, we also have two missing ground-truth rules that are instead incorrectly discovered as a single more general rule.

| Ground-truth logic program | Discovered logic program |
|---|---|
| $A_1^t \to A_2^{t+1}$, | $A_1^t \to A_2^{t+1}$, |
| $A_2^t \wedge A_3^t \to A_3^{t+1}$, | $A_2^t \wedge A_3^t \to A_3^{t+1}$, |
| $A_3^t \wedge \neg A_9^t \to A_3^{t+1}$, | $A_3^t \wedge \neg A_9^t \to A_3^{t+1}$, |
| $A_2^t \wedge \neg A_9^t \to A_3^{t+1}$, | $A_2^t \wedge \neg A_9^t \to A_3^{t+1}$, |
| $A_2^t \wedge \neg A_9^t \to A_4^{t+1}$, | $A_2^t \wedge \neg A_9^t \to A_4^{t+1}$, |
| $A_4^t \wedge \neg A_9^t \to A_4^{t+1}$, | $A_4^t \wedge \neg A_9^t \to A_4^{t+1}$, |
| $A_2^t \wedge A_4^t \to A_4^{t+1}$, | $A_2^t \wedge A_4^t \to A_4^{t+1}$, |
| $A_3^t \to A_5^{t+1}$, | $A_3^t \to A_5^{t+1}$, |
| $\neg A_5^t \wedge A_6^t \wedge A_{11}^t \wedge A_{12}^t \to A_6^{t+1}$, | $\neg A_5^t \wedge A_6^t \wedge A_{11}^t \wedge A_{12}^t \to A_6^{t+1}$, |
| $\neg A_5^t \wedge A_6^t \wedge \neg A_7^t \wedge A_{11}^t \to A_6^{t+1}$, | $\neg A_5^t \wedge A_6^t \wedge \neg A_7^t \wedge A_{11}^t \to A_6^{t+1}$, |
| $\neg A_5^t \wedge A_6^t \wedge \neg A_9^t \wedge A_{11}^t \to A_6^{t+1}$, | $\neg A_5^t \wedge A_6^t \wedge \neg A_9^t \wedge A_{11}^t \to A_6^{t+1}$, |
| $A_6^t \wedge \neg A_7^t \wedge \neg A_9^t \wedge A_{11}^t \to A_6^{t+1}$, | $A_6^t \wedge \neg A_7^t \wedge \neg A_9^t \wedge A_{11}^t \to A_6^{t+1}$, |
| $A_6^t \wedge \neg A_9^t \wedge A_{11}^t \wedge A_{12}^t \to A_6^{t+1}$, | $A_6^t \wedge \neg A_9^t \wedge A_{11}^t \wedge A_{12}^t \to A_6^{t+1}$, |
| $\neg A_7^t \wedge \neg A_9^t \wedge A_{11}^t \wedge A_{12}^t \to A_6^{t+1}$, | $\neg A_7^t \wedge \neg A_9^t \wedge A_{11}^t \wedge A_{12}^t \to A_6^{t+1}$, |
| $\neg A_5^t \wedge A_6^t \wedge \neg A_7^t \wedge A_{12}^t \to A_6^{t+1}$, | $\neg A_5^t \wedge A_6^t \wedge \neg A_7^t \wedge A_{12}^t \to A_6^{t+1}$, |
| $A_6^t \wedge \neg A_7^t \wedge A_{11}^t \wedge A_{12}^t \to A_6^{t+1}$, | $A_6^t \wedge \neg A_7^t \wedge A_{11}^t \wedge A_{12}^t \to A_6^{t+1}$, |
| $A_6^t \wedge \neg A_7^t \wedge \neg A_9^t \wedge A_{12}^t \to A_6^{t+1}$, | $A_6^t \wedge \neg A_7^t \wedge \neg A_9^t \wedge A_{12}^t \to A_6^{t+1}$, |
| $\neg A_5^t \wedge \neg A_7^t \wedge \neg A_9^t \wedge A_{11}^t \to A_6^{t+1}$, | $\neg A_5^t \wedge \neg A_7^t \wedge \neg A_9^t \wedge A_{11}^t \to A_6^{t+1}$, |
| $\neg A_5^t \wedge A_6^t \wedge \neg A_7^t \wedge \neg A_9^t \to A_6^{t+1}$, | $\neg A_5^t \wedge A_6^t \wedge \neg A_7^t \wedge \neg A_9^t \to A_6^{t+1}$, |
| $\neg A_5^t \wedge \neg A_7^t \wedge A_{11}^t \wedge A_{12}^t \to A_6^{t+1}$, | $\neg A_5^t \wedge \neg A_7^t \wedge A_{11}^t \wedge A_{12}^t \to A_6^{t+1}$, |
| $\neg A_5^t \wedge \neg A_9^t \wedge \neg A_{11}^t \wedge A_{12}^t \to A_6^{t+1}$, | $\neg A_5^t \wedge \neg A_9^t \wedge \neg A_{11}^t \wedge A_{12}^t \to A_6^{t+1}$, |
| $\neg A_5^t \wedge \neg A_7^t \wedge \neg A_9^t \wedge A_{12}^t \to A_6^{t+1}$, | |
| $\neg A_5^t \wedge A_6^t \wedge \neg A_9^t \wedge A_{12}^t \to A_6^{t+1}$, | |
| $A_4^t \wedge A_7^t \wedge \neg A_{11}^t \to A_7^{t+1}$, | $A_4^t \wedge A_7^t \wedge \neg A_{11}^t \to A_7^{t+1}$, |
| $\neg A_6^t \wedge A_7^t \wedge \neg A_{11}^t \to A_7^{t+1}$, | $\neg A_6^t \wedge A_7^t \wedge \neg A_{11}^t \to A_7^{t+1}$, |
| $A_4^t \wedge \neg A_6^t \wedge A_7^t \to A_7^{t+1}$, | $A_4^t \wedge \neg A_6^t \wedge A_7^t \to A_7^{t+1}$, |
| $A_4^t \wedge \neg A_6^t \wedge \neg A_{11}^t \to A_7^{t+1}$, | $A_4^t \wedge \neg A_6^t \wedge \neg A_{11}^t \to A_7^{t+1}$, |
| $\neg A_5^t \wedge \neg A_7^t \wedge A_8^t \wedge A_{11}^t \to A_8^{t+1}$, | $\neg A_5^t \wedge \neg A_7^t \wedge A_8^t \wedge A_{11}^t \to A_8^{t+1}$, |
| $\neg A_5^t \wedge A_8^t \wedge \neg A_9^t \wedge A_{11}^t \to A_8^{t+1}$, | $\neg A_5^t \wedge A_8^t \wedge \neg A_9^t \wedge A_{11}^t \to A_8^{t+1}$, |
| $\neg A_7^t \wedge A_8^t \wedge \neg A_9^t \wedge A_{11}^t \to A_8^{t+1}$, | $\neg A_7^t \wedge A_8^t \wedge \neg A_9^t \wedge A_{11}^t \to A_8^{t+1}$, |
| $\neg A_5^t \wedge \neg A_7^t \wedge A_8^t \wedge \neg A_9^t \to A_8^{t+1}$, | $\neg A_5^t \wedge \neg A_7^t \wedge A_8^t \wedge \neg A_9^t \to A_8^{t+1}$, |
| $\neg A_5^t \wedge \neg A_7^t \wedge \neg A_9^t \wedge A_{11}^t \to A_8^{t+1}$, | $\neg A_5^t \wedge \neg A_7^t \wedge \neg A_9^t \wedge A_{11}^t \to A_8^{t+1}$, |
| $\neg A_6^t \wedge A_7^t \wedge \neg A_8^t \wedge \neg A_{11}^t \to A_9^{t+1}$, | $\neg A_6^t \wedge A_7^t \wedge \neg A_8^t \wedge \neg A_{11}^t \to A_9^{t+1}$, |
| $A_7^t \wedge A_9^t \wedge A_{10}^t \wedge \neg A_{11}^t \to A_9^{t+1}$, | $A_7^t \wedge A_9^t \wedge A_{10}^t \wedge \neg A_{11}^t \to A_9^{t+1}$, |
| $\neg A_8^t \wedge A_9^t \wedge A_{10}^t \wedge \neg A_{11}^t \to A_9^{t+1}$, | $\neg A_8^t \wedge A_9^t \wedge A_{10}^t \wedge \neg A_{11}^t \to A_9^{t+1}$, |
| $\neg A_6^t \wedge A_7^t \wedge A_{10}^t \wedge \neg A_{11}^t \to A_9^{t+1}$, | $\neg A_6^t \wedge A_7^t \wedge A_{10}^t \wedge \neg A_{11}^t \to A_9^{t+1}$, |
| $A_7^t \wedge \neg A_8^t \wedge A_9^t \wedge A_{10}^t \to A_9^{t+1}$, | $A_7^t \wedge \neg A_8^t \wedge A_9^t \wedge A_{10}^t \to A_9^{t+1}$, |
| $A_7^t \wedge \neg A_8^t \wedge A_9^t \wedge \neg A_{11}^t \to A_9^{t+1}$, | $A_7^t \wedge \neg A_8^t \wedge A_9^t \wedge \neg A_{11}^t \to A_9^{t+1}$, |
| $\neg A_6^t \wedge A_7^t \wedge \neg A_8^t \wedge A_9^t \to A_9^{t+1}$, | $\neg A_6^t \wedge A_7^t \wedge \neg A_8^t \wedge A_9^t \to A_9^{t+1}$, |
| $\neg A_6^t \wedge \neg A_8^t \wedge A_{10}^t \wedge \neg A_{11}^t \to A_9^{t+1}$, | $\neg A_6^t \wedge \neg A_8^t \wedge A_{10}^t \wedge \neg A_{11}^t \to A_9^{t+1}$, |
| $\neg A_6^t \wedge A_7^t \wedge \neg A_8^t \wedge A_{10}^t \to A_9^{t+1}$, | $\neg A_6^t \wedge A_7^t \wedge \neg A_8^t \wedge A_{10}^t \to A_9^{t+1}$, |
| $A_7^t \wedge \neg A_8^t \wedge A_{10}^t \wedge \neg A_{11}^t \to A_9^{t+1}$, | $A_7^t \wedge \neg A_8^t \wedge A_{10}^t \wedge \neg A_{11}^t \to A_9^{t+1}$, |
| $\neg A_6^t \wedge A_9^t \wedge A_{10}^t \wedge \neg A_{11}^t \to A_9^{t+1}$, | $\neg A_6^t \wedge A_9^t \wedge A_{10}^t \wedge \neg A_{11}^t \to A_9^{t+1}$, |
| $\neg A_6^t \wedge \neg A_8^t \wedge A_9^t \wedge A_{10}^t \to A_9^{t+1}$, | $\neg A_6^t \wedge \neg A_8^t \wedge A_9^t \wedge A_{10}^t \to A_9^{t+1}$, |
| $\neg A_6^t \wedge \neg A_8^t \wedge A_9^t \wedge \neg A_{11}^t \to A_9^{t+1}$, | $\neg A_6^t \wedge \neg A_8^t \wedge A_9^t \wedge \neg A_{11}^t \to A_9^{t+1}$, |
| $\neg A_6^t \wedge A_7^t \wedge A_9^t \wedge A_{10}^t \to A_9^{t+1}$, | $A_6^t \wedge A_7^t \wedge A_9^t \to A_9^{t+1}$, |
| $\neg A_6^t \wedge A_7^t \wedge A_9^t \wedge \neg A_{11}^t \to A_9^{t+1}$, | |
| $A_7^t \to A_{10}^{t+1}$, | $A_7^t \to A_{10}^{t+1}$, |
| $A_9^t \to A_{10}^{t+1}$, | $A_9^t \to A_{10}^{t+1}$, |
| $A_9^t \to A_{11}^{t+1}$, | $A_9^t \to A_{11}^{t+1}$, |
| $A_{10}^t \to A_{11}^{t+1}$, | $A_{10}^t \to A_{11}^{t+1}$, |
| $\neg A_9^t \wedge A_{11}^t \to A_{12}^{t+1}$, | $\neg A_9^t \wedge A_{11}^t \to A_{12}^{t+1}$, |
| $A_{10}^t \wedge A_{11}^t \to A_{12}^{t+1}$, | $A_{10}^t \wedge A_{11}^t \to A_{12}^{t+1}$, |
| $\neg A_9^t \wedge A_{10}^t \to A_{12}^{t+1}$, | $\neg A_9^t \wedge A_{10}^t \to A_{12}^{t+1}$, |

where

- $\neg A_5^t \wedge \neg A_7^t \wedge \neg A_9^t \wedge A_{12}^t \rightarrow A_6^{t+1}$ is correctly subsumed by $\neg A_5^t \wedge \neg A_7^t \wedge A_{11}^t \wedge A_{12}^t \rightarrow A_6^{t+1}$ and $\neg A_5^t \wedge \neg A_9^t \wedge \neg A_{11}^t \wedge A_{12}^t \rightarrow A_6^{t+1}$,

- $\neg A_5^t \wedge A_6^t \wedge \neg A_9^t \wedge A_{12}^t \rightarrow A_6^{t+1}$ is correctly subsumed by $\neg A_5^t \wedge A_6^t \wedge \neg A_9^t \wedge A_{11}^t \rightarrow A_6^{t+1}$ and $\neg A_5^t \wedge \neg A_9^t \wedge \neg A_{11}^t \wedge A_{12}^t \rightarrow A_6^{t+1}$,

- but $\neg A_6^t \wedge A_7^t \wedge A_9^t \wedge A_{10}^t \rightarrow A_9^{t+1}$ and $\neg A_6^t \wedge A_7^t \wedge A_9^t \wedge \neg A_{11}^t \rightarrow A_9^{t+1}$ are incorrectly discovered as the more general $\neg A_6^t \wedge A_7^t \wedge A_9^t \rightarrow A_9^{t+1}$, *i.e.* the third rule covers all the cases when the first two rules are activated, but it is also activated in other cases where it should not.

### D.2 Tabular classification

### D.2.1 Comparison of interpretability in number of rules and average size of rules

To compare the interpretability of the learned models by NLN and by RRL, we use two traditional measures of interpretability for logic programs: average number of rules per target and average rule size. Since both networks can learn a rule in a factorized fashion through the previous layers, we consider the size of a rule to be the number of nodes in the input layer that are used

- for binary features, if the feature itself is used;

- for categorical features, the number of its values that are used (in the one-hot encoding);

- for continuous features, the number of boundary nodes that are used (*i.e.* the number of fuzzy dichotomies in our case).

The results are given in Table D.1.

Table D.1: Comparison of interpretability in number of rules and average size of rules

| Datasets | **NLN** | | | RRL | | |
|---|---|---|---|---|---|---|
| | f1 (%) | nbr. | size | f1 (%) | nbr. | size |
| adult | 66.03 | **77.4** | 22.95 | **80.20** | 3599.2 | 5.34 |
| balance | 58.78 | **33.7** | 11.70 | **77.72** | 341.3 | 2.27 |
| balance (cat.) | 59.69 | **18.5** | 4.99 | **82.20** | 97.5 | 1.23 |
| chess | **99.58** | **9.8** | 5.45 | 99.43 | 692.8 | 1.49 |
| DARWIN | 77.22 | **12.8** | 10.49 | **86.01** | 650.4 | 1.97 |
| monk2 | 86.81 | **17.8** | 5.71 | **98.30** | 591.6 | 2.49 |
| tic-tac-toe | **100** | **8.0** | 3.00 | **100** | 462.0 | 2.07 |
| wine | 94.44 | **12.3** | 5.27 | **98.23** | 20.7 | 1.05 |

In general, the RRL uses many times more rules than the NLN and these rules are of even smaller size on average. The average rule sizes in the NLN are already small enough to be easily interpretable, but those in RRL are suspiciously simple in size with many rules involving only a single feature value. Since their number is so high, it suggests that the actual rules learned by the RRL are contained in the distributed representation of the output linear layer. Hence, it is difficult to interpret what is the value of the interpretable "rules" that it learns in the preceding logical layer(s). In the two datasets where the NLN performs as well as or better than the RRL, chess and tic-tac-toe, the NLN only needed 10 rules or less in all 5 runs while the RRL needed over 300.

### D.2.2 Example of rules found for the adult dataset

The adult dataset aims to predict whether an individual earns a salary of more than $ 50 K/year (in the US in 1994) based on census data. The NLN with the best predictive performance on the whole dataset that was found in the five-fold cross-validation is pictured in Figure D.1(a). It contains 70 rules, some of which are pictured in Figures D.1(b-c) and D.2. Although this NLN only has a f1 score of 72.48 % on the full dataset, the relevant rules that it found are for the most part easily interpretable. In some cases, as in D.2(a) and D.2(e), a rule may be "overfitted" over a continuous feature like the individual's age or number of work hours per week, resulting in a less interpretable rule.

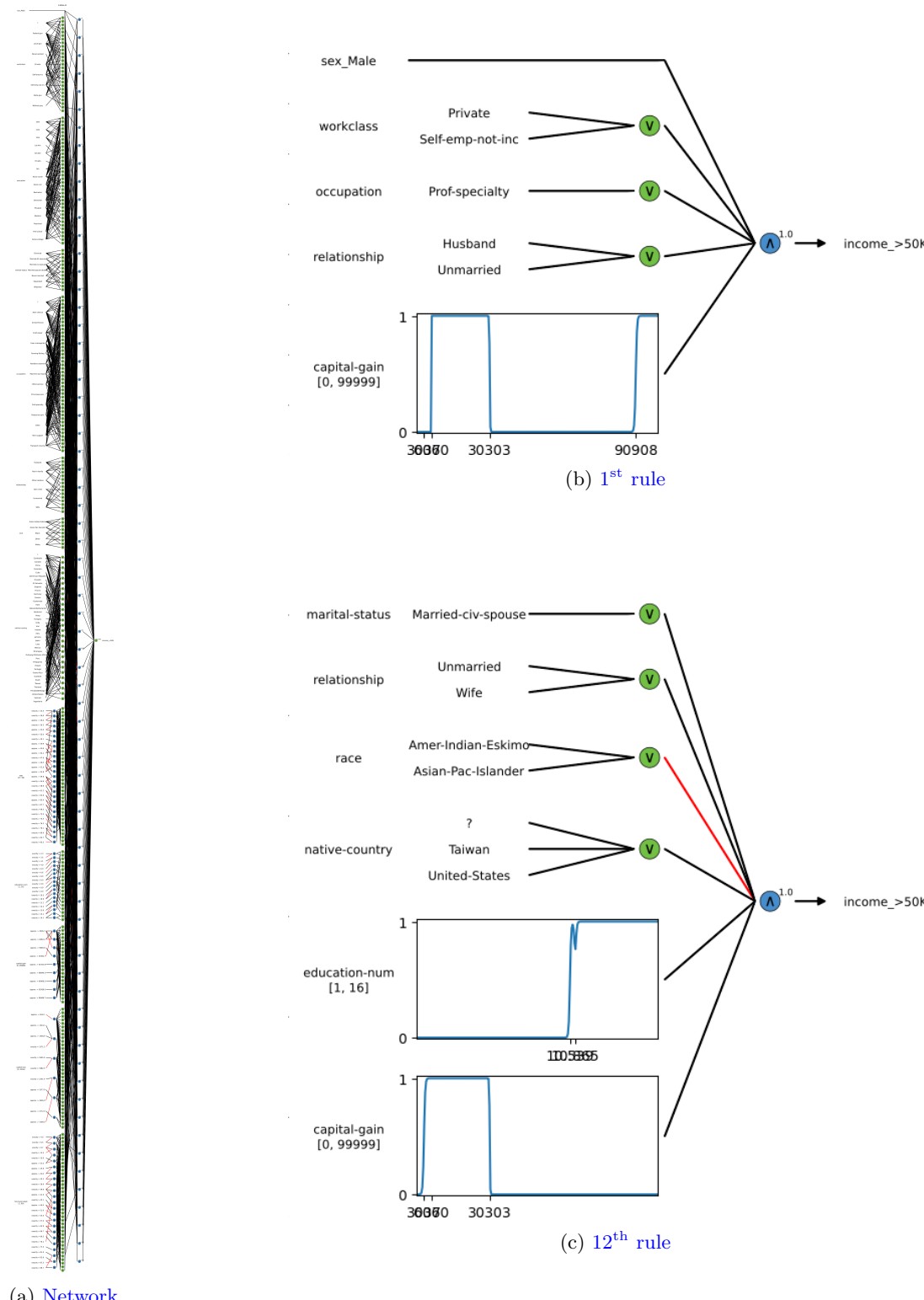

(b) 1st rule

(c) 12th rule

(a) Network

Figure D.1: Best NLN found for the adult dataset

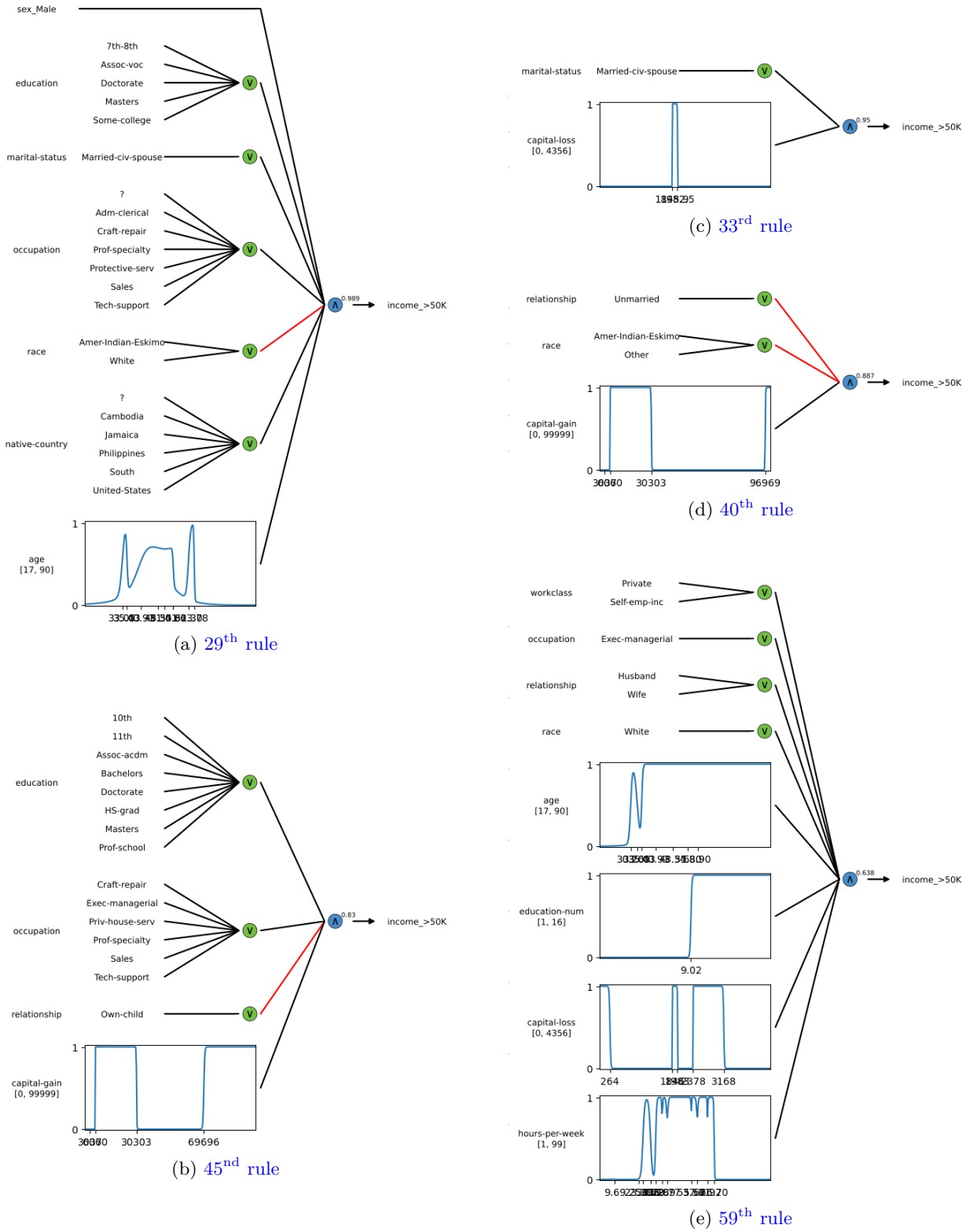

Figure D.2: More rules from the best NLN found for the adult dataset

### D.2.3 Example of rules found for the DARWIN dataset

The DARWIN dataset aims to predict whether an individual has Alzheimer's disease (is a Patient) from 25 handwriting tasks, totaling 450 continuous features. The NLN with the best predictive performance on the whole dataset that was found in the five-fold cross-validation is pictured in Figure D.3(a). It contains 10 rules, some of which are pictured in Figures D.3(b) and D.4. This NLN achieves a f1 score of 96.09 % on the full dataset, with rules that can be interpreted by experts in the handwriting test.

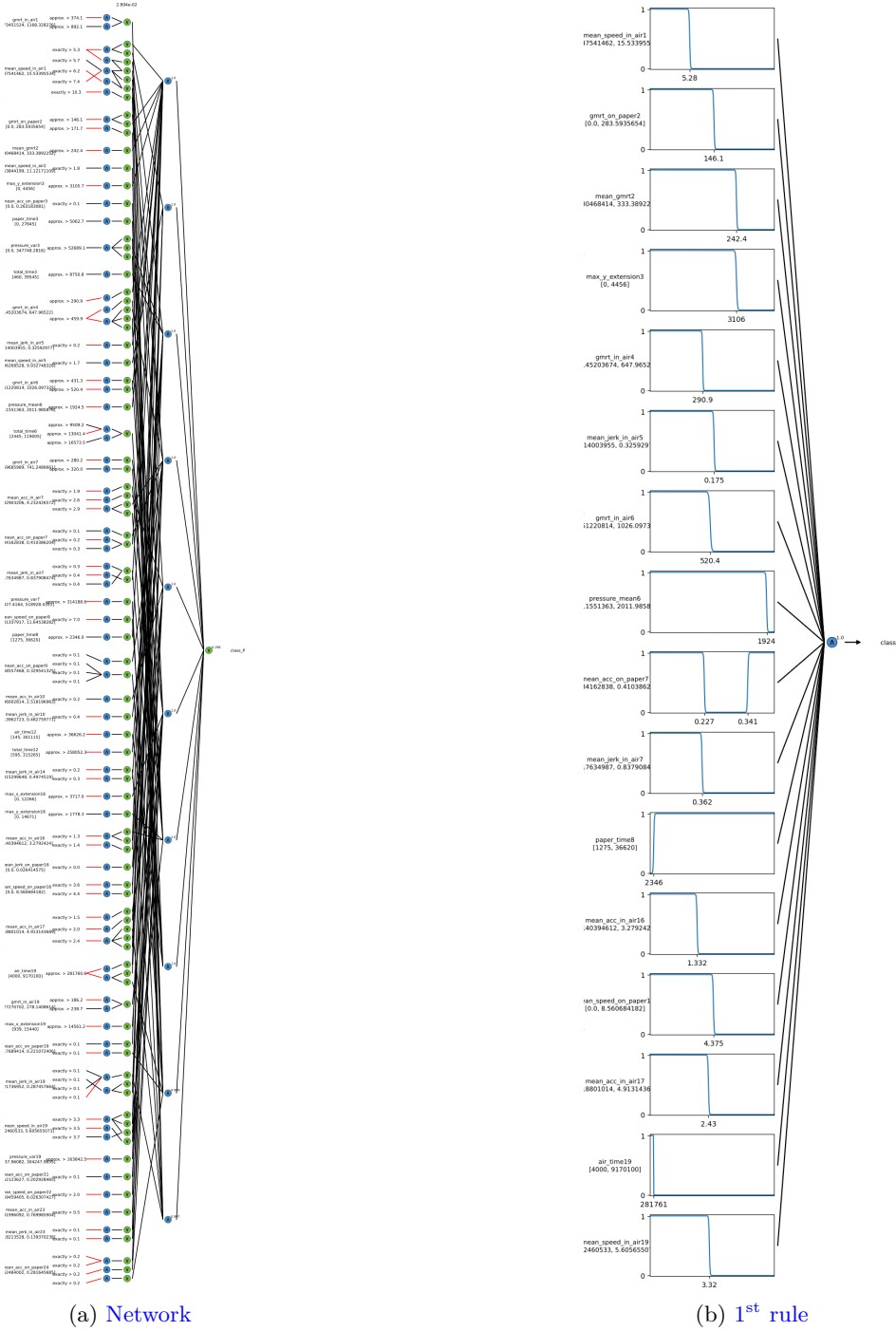

(a) Network  (b) 1$^{\text{st}}$ rule

Figure D.3: Best NLN found for the DARWIN dataset

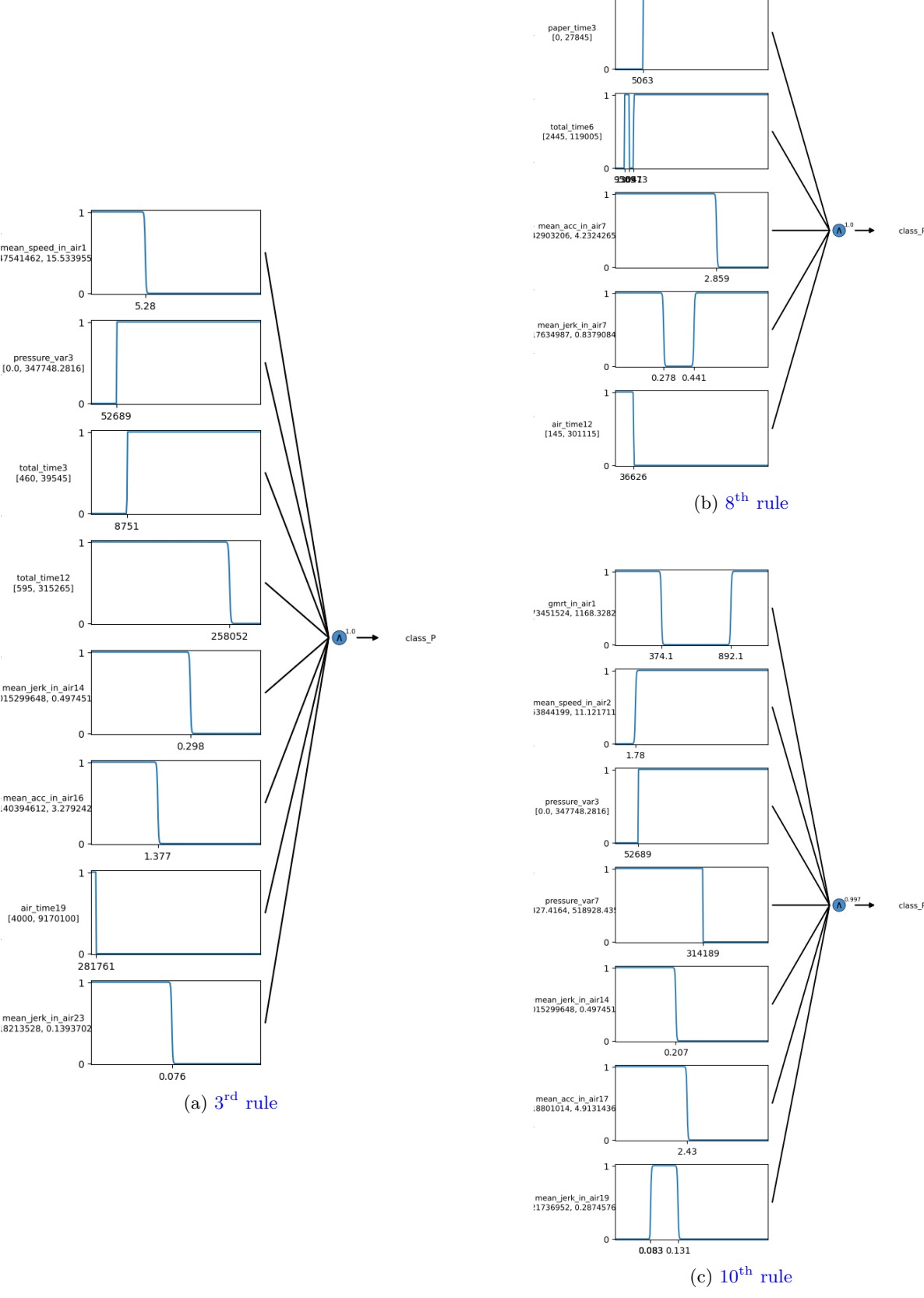

Figure D.4: More rules from the best NLN found for the DARWIN dataset

