# OpenReview forum: "Neural Logic Networks for Interpretable Classification"
_TMLR — Rejected by TMLR_

### Review · Reviewer_cZZ3 · 2025-02-19

**Summary Of Contributions:**

The paper presents a new formulation of AND/OR networks to define a neural logic network that is both lightweight and capable of explaining the rationale behind its computed results.
This formulation has been tested on two different problems: modeling Boolean formulas and performing classification on tabular data.

**Audience:**

Yes

**Broader Impact Concerns:**

There are no ethical implications

**Claims And Evidence:**

No

**Requested Changes:**

On Page 4, when describing the OR concept, the paper introduces concepts C_1 and C_3 with definitions different from those used for C_1 and C_3 in the previous example for AND concepts. I suggest using distinct symbols to avoid confusion.

On the same page, the sentence "random variable x=(...) that is defined by m measures" appears verbatim at the beginning of Section 2.1. I would suggest defining it once.

On the last line of page 4, the symbol n^l is used but never defined.

On page 5, the paper states that the probability of input concepts is known. It is important to specify whether this probability can take only values 0 or 1 or whether it can assume values within the interval [0,1].

In the equations for A_{i,j}^t and O_{i,j}^t, I suggest reordering the concepts consistently, possibly following the layer sequence. Specifically, use (C_i^{(l-1)} \superseteq C_i^{(l)}) in A_{i,j}^t or (C_i^{(l)} \superseteq C_i^{(l-1)}) in O_{i,j}^t.

On page 7, the sentence "As a purely probabilistic model, this issue is catastrophic for NLNs. However, as an ML method, this approximation is necessary to make it practical." is not so clear and convincing. This kind of relaxation should be discussed in more depth.

On page 8, equations D-AND and D-OR are repeated. Instead, I suggest assigning a label to the first definition and  refer to that label later.

On page 9, given the journal's audience, De Morgan's laws and distributivity can be assumed as well-known and should be omitted.

With the space gained by removing repetitions, I would add on section 2.4 examples from appendix B.3 instead of referring to them. This would improve readability and prevent the section from being excessively short.

In the last line before Figure 2 I would add a reference to Section 3.1.2 to enhance readability.
In general, introductory parts of sections that contain subsections should outline the organization of the subsections, guiding the reader to the relevant information.

On page 11, the paper should discuss how fuzzy dichotomies are learned, potentially with an example.

As regards section 3.1.3, it is unclear how discrete and continuous features are managed and how rule modules work. The section narrates how they work, but the paper should also provide rigorous definitions.
Additionally, the title "Rule Modules" suggests a focus solely on rule modules, yet the discussion includes continuous features that do not use rule modules. The section should either be retitled or revised to align with its title.

Section 3.2: lacks an introductory paragraph and directly starts with subsections.

In section 3.2.1, the sentence "we use ADAM gradient descent" is incorrect since ADAM is an optimizer implementing gradient descent. Moreover, the entire first paragraph is unclear to me. What information is it meant to convey? Binary cross-entropy is a loss function typically used for binary classification (typically a single output with sigmoid activation function). Softmax is used with categorical cross-entropy for multiclass classification. ADAM can be used with any differentiable loss function. The choice of the loss depends on the network output, as partly reported by the paragraph. However, it is not clear the meaning of the discussion reported here.

Finally, this section should report the formal definition of the final loss function used in the model.

On page 13, the paper states that the network's weights are probabilities that are hard to interpret. Why are they hard to interpret? An example would be useful.

On page 15, the claim that the proposed approach achieves 100% accuracy on the full dataset is inconsistent with the results for Budding (assuming no errors in the table). This discrepancy should be addressed.

Regarding the 100% Accuracy, could the network learn rules that are not included in or subsumed by the ground truth? Is the ground truth assumed to be complete (i.e., containing all possible rules/descriptions)? A discussion on whether the network could generate novel definitions for a target concept not in the ground truth would be insightful.

On page 16, the paper refers to "half OR and half AND nodes" but does not explain what they are. Consider describing them in the related work section when the corresponding paper is cited.

On the same page, the paper should explain how default values for fuzzy dichotomies and fuzzy intervals were defined. This discussion should be added in the section on dichotomies.

Typos (Partial List)
Page 5: either *used* directly C_j^(l-1) -> used should be using
Page 16: avoir -> avoid

**Strengths And Weaknesses:**

Overall, the paper is interesting but requires substantial revisions. The writing quality should be improved by eliminating typos, verifying notations, clarifying ambiguous sentences, and, in general, restructuring the organization of sections. Many sections are either too short or have titles that do not fully reflect their content. The paper is highly descriptive and narrative, which is acceptable if textual descriptions are supplemented with formal definitions. However, in many cases, concepts are explained only in words, even when they involve mathematical formulations.
Below, I provide detailed comments.

---

> ### Author Response · Authors · 2025-04-21
>
> Dear reviewer, thank you for your comments on the text which have been very helpful in improving its structure, clarifying some ideas including more formal definitions, as well as adding more insights about the method. Each of your specific comments has been addressed below. Moreover, since the initial submission, a new initialization for the method has yielded better results all around, and the text was updated accordingly.
>
> -	On Page 4, when describing the OR concept, the paper introduces concepts $C_1$ and $C_3$ with definitions different from those used for $C_1$ and $C_3$ in the previous example for AND concepts. I suggest using distinct symbols to avoid confusion.
>  -	Apostrophes have been added to distinguish between concepts for the first example $C_i$ and from the second example $C_{i’}$.
> -	On the same page, the sentence "random variable $x=(...)$ that is defined by $m$ measures" appears verbatim at the beginning of Section 2.1. I would suggest defining it once.
>  -	The definition has been kept only for the second instance and the first instance has been simplified to just “$x$”.
> -	On the last line of page 4, the symbol $n^l$ is used but never defined.
>  -	A definition has been added: “We use a network of concepts $C_i^l$ arranged in layers $l \in ${$0, 1, ..., L$} of size $n^l$ with $i \in ${$1, ..., n^l$}”
> -	On page 5, the paper states that the probability of input concepts is known. It is important to specify whether this probability can take only values 0 or 1 or whether it can assume values within the interval [0,1].
>  -	The specification that any value in [0,1] is possible has been added: “These input probabilities $c_i^0(x)$ can take any value in [0, 1], with binary values {0, 1} representing certain knowledge about $x$.”
> -	In the equations for $A_{i,j}^t$ and $O_{i,j}^t$, I suggest reordering the concepts consistently, possibly following the layer sequence. Specifically, use $(C_i^{(l-1)} \superseteq C_i^{(l)})$ in $A_{i,j}^t$ or $(C_i^{(l)} \superseteq C_i^{(l-1)})$ in $O_{i,j}^t$.
>  -	The consistent ordering following the order of layers has been applied everywhere: “$C_j^{l-1} \superseteq C_i^l$” for necessary concepts (AND), and “$C_j^{l-1} \subseteq C_i^l$” for sufficient concepts (OR).
> -	On page 7, the sentence "As a purely probabilistic model, this issue is catastrophic for NLNs. However, as an ML method, this approximation is necessary to make it practical." is not so clear and convincing. This kind of relaxation should be discussed in more depth.
>  -	The approximation has been justified more thoroughly: “As a purely probabilistic model, this issue is catastrophic for NLNs. However, as a ML method, this approximation can be justified. First of all, in the case where all inputs are binary (i.e. with binary or categorical features), where all the weights are given from the set {−1, 0, 1} and where the biases are full with $a_{\cdot}^l = 1$, $o_{\cdot}^l = 0$, then this assumption is not necessary. In fact, none of the assumptions are necessary in that case because the probabilistic formulation (P-AND) and (P-OR) coincide exactly with the logical definition of the AND and OR nodes (see section 2.2). In general, we always need the weights to be integers for maximum interpretability and, as such, we quantize them during post-processing when learning a NLN (see section 3.2.2). Moreover, for any network that contains continuous features, with enough pre-processing nodes (see section 3.1.2), the signal can become arbitrarily close to binary thanks to appropriately scaled sigmoid curves. In practice, we only relax this binary constraint through the possible non-binary biases, as well as by limiting the number of pre-processing nodes for continuous features, so that values away from 0 and 1 can be obtained close to the learned boundaries (see section 3.1.2). Therefore, we keep this approximation leading to (P-AND) and (P-OR) and, in practice, this modeling is still able to obtain promising predictive performance and interpretable rule discovery, even in non-binary cases.”
> -	On page 8, equations D-AND and D-OR are repeated. Instead, I suggest assigning a label to the first definition and refer to that label later.
>  -	Your suggestion of labeling the first instance and referring to it afterwards has been adopted.
> -	On page 9, given the journal's audience, De Morgan's laws and distributivity can be assumed as well-known and should be omitted.
>  -	The definitions have been omitted.
> -	With the space gained by removing repetitions, I would add on section 2.4 examples from appendix B.3 instead of referring to them. This would improve readability and prevent the section from being excessively short.
>  -	The table of possible interpretations as well as the accompanying visual examples have been moved to the main text. Only the counter-intuitive examples of causal structures remain in the appendix.
>
> ...

---

> > ### Author Response · Authors · 2025-04-21
> >
> > -	In the last line before Figure 2 I would add a reference to Section 3.1.2 to enhance readability. In general, introductory parts of sections that contain subsections should outline the organization of the subsections, guiding the reader to the relevant information.
> >  -	References to subsections have been added: “We propose the structure pictured in Figure 3. It contains two fully-connected layers arranged in DNF, i.e. an AND layer with negation followed by an OR layer without negation, in order to learn a logic program for each target (section 3.1.1). The input features that are not binary are pre-processed with appropriate input modules, one for categorical features, and another for continuous features (section 3.1.2).”
> > -	On page 11, the paper should discuss how fuzzy dichotomies are learned, potentially with an example.
> >  -	An example has been added and it has been specified that the fuzzy dichotomies are learned conjointly with all other AND/OR concepts in the network: “For instance, in a task that uses a continuous feature $x_k$ representing weight, one rule might hold only for very light objects or somewhat heavy objects such as $x_k \in [0, 0.1] \cup [10, 15]$ which would be learned as $(x_k < 0.1) \cup ((x_k > 10) \cap (x_k < 15))$. […] The fuzzy dichotomies are learned conjointly with all the AND/OR concepts in the NLN.”
> > -	As regards section 3.1.3, it is unclear how discrete and continuous features are managed and how rule modules work. The section narrates how they work, but the paper should also provide rigorous definitions.
> >  -	Clear definitions have replaced the more narrative description: “Each rule module contains a single AND rule that takes its inputs from (1) the binary features, (2) its own OR equivalency classes, one for each categorical feature, and (3) its own OR collections of fuzzy intervals, one for each continuous feature. In turn, binary features are used directly, but categorical features have a shared one-hot encoding, and continuous features are encoded with shared fuzzy dichotomies and AND fuzzy intervals. This factorization of the structure allows each rule to learn more independently of the others and reduces the number of parameters in the model. However, the fuzzy interval encodings of the continuous features are still learned conjointly for the whole NLN.”
> > -	Additionally, the title "Rule Modules" suggests a focus solely on rule modules, yet the discussion includes continuous features that do not use rule modules. The section should either be retitled or revised to align with its title.
> >  -	The title has been changed to “Input encodings and rule modules” to better reflect its content.
> > -	Section 3.2: lacks an introductory paragraph and directly starts with subsections.
> >  -	Sections 3.2 Training and 3.3 Post-processing have been restructured as subsections 3.2.1 and 3.2.2 of a new section 3.2 Learning. This merges many smaller subsections into one with paragraphs. An introductory paragraph has also been added to the beginning of the new section 3.2 to facilitate reading: “Learning a NLN is done in two stages: (1) training, and (2) post-processing, which includes weight quantizing, continuous parameter retraining, and pruning.”
> > -	In section 3.2.1, the sentence "we use ADAM gradient descent" is incorrect since ADAM is an optimizer implementing gradient descent.
> >  -	This has been corrected to “we use the ADAM optimizer”.
> > -	Moreover, the entire first paragraph is unclear to me. What information is it meant to convey? Binary cross-entropy is a loss function typically used for binary classification (typically a single output with sigmoid activation function). Softmax is used with categorical cross-entropy for multiclass classification. ADAM can be used with any differentiable loss function. The choice of the loss depends on the network output, as partly reported by the paragraph. However, it is not clear the meaning of the discussion reported here.
> >  -	The text has been adjusted to make its point more clearly. It is meant to answer the question why we did not choose cross-entropy over L2 as our loss function : “We use the ADAM optimizer (Kingma & Ba, 2015) to minimize the L2 loss. Its minimizer is $P[ω \in Y_k | x]$ which is precisely what we want our NLN’s outputs $c_k^2(x) = P[ω \in C_k^2 | x]$ to model. Another choice for the loss function could have been cross-entropy which has also the same minimizer, but since our output layer is not made up of sigmoids or softmax, it does not offer the same structural advantage in NLNs as it does in typical neural networks (Bishop, 1995)”

---

> > > ### Comment · Reviewer_cZZ3 · 2025-04-23
> > > **About cross-entropy and ADAM**
> > >
> > > In my opinion, corrections proposed to the paragraph are not convincing. Specifically, if I correctly understood the meaning, the entire paragraph is useless because if we do not perform classification (i.e., no sigmoid or softmax are used), cross entropy is trivially not the smartest choice. On the other hand, ADAM basically minimizes every function. So, of course, it minimizes both L2 and cross-entropy. What am I missing here?

---

> ### Author Response · Authors · 2025-04-21
>
> -	Finally, this section should report the formal definition of the final loss function used in the model.
>  -	The section has been rewritten to clarify the use of the regularizers, and the objective, the full loss function as well as the domain constraints have been clarified with equations: “To help the learning process we regularize NLNs in two different ways. First, to combat the tendency of unadapted concepts to become trivial, we regularize the AND and OR concepts to have non-empty definitions. For instance, we consider the definition of an AND concept $C_i^l$ to be non-empty if
> $$\sum_j |A_{i,j}^l|\geq 1,$$
> in other words, we consider a concept $C_i^l$ to be non-empty if it attributes a probability mass of at least 1 across all of its possible input concepts $C_j^{l-1}$. We force non-empty definitions in all AND and OR concepts by penalizing
> L_{non-empy} = \sum_{AND weights A_{i,.}^l} || [1 - \sum_j |A_{i,j}^l|]_+ ||_2^2 + \sum_{OR weights O_{i,.}^l} || [1 - \sum_j |O_{i,j}^l|]_+ ||_2^2, [EQUATION DOES NOT APPEAR CORRECTLY HERE, SEE REVISED ARTICLE]
> which is only active when a concept $C_i^l$’s definition attributes a probability mass less than 1. In that case, the penalty will increase the weights of all of its input concepts $C_j^{l-1}$ uniformly until a probability mass of at least 1 is attributed. Moreover, in order to encourage sparser, more interpretable solutions, we also penalize the L1 norm of all weights in the network. The full loss function is then given by
> L(y, c^L(x)) = ||y – c^L(x)||_2^2 + λ_{non-empty} · L_{non-empty} + λ_{sparsity} ( \sum_{AND weights A_{i,.}^l} ||A_{i,.}^l||_1 + \sum_{OR weights O_{i,.}^l} ||O_{i,.}^l||_1 ), [EQUATION DOES NOT APPEAR CORRECTLY HERE, SEE REVISED ARTICLE]
> where λ_{non-empty}, λ_{sparsity} > 0 are the regularization coefficients of the non-empty penalty and the sparsity penalty respectively. We minimize the expectation of this loss
> $$E_{(X,Y )∼D}[L(Y, c^L(X))]$$
> over the training dataset D, subject to the domain constraints of the weights $A_{i,j}^l \in [−1, 1]$, $O_{i’,j’}^{l’} \in [0, 1]$, the biases $a_i^l, o_{i’}^{l’} \in [0, 1]$, and the parameters of the fuzzy dichotomies $B_{i’’,k} \in R$, $α_{i’’,k} > 0$ for appropriate indices $(i, j, l)$, $(i', j’, l’)$ and $(i′′, k)$ according to the NLN’s structure.”
> -	On page 13, the paper states that the network's weights are probabilities that are hard to interpret. Why are they hard to interpret? An example would be useful.
>  -	An example has been added to clarify the difficulty in interpreting a concept with multiple weights that are not integers: “At this point, the NLN has been learned but its weights are still probabilities which are hard to interpret, especially in conjunction with one another. For instance, a simple AND rule that would have weights of (0.80, −0.65, 0.15) for respectively ball, green and heavy is difficult to interpret. It represents a concept that is likely a type of ball, but not necessarily; a concept that is probably not a green object; and a concept that might be a heavy object although it is unlikely; all simultaneously. This is not easily interpretable, unlike the same weights after quantizing which might be (1, −1, 0) that would represent the concept of a ball_that_is_not_green. By quantizing the weights to values of either 0, 1 or −1, we obtain instantly understandable concepts that still retain a probabilistic bias in [0, 1], indicating if we are missing other unobserved concepts in its definition and how often they appear.”
> -	On page 15, the claim that the proposed approach achieves 100% accuracy on the full dataset is inconsistent with the results for Budding (assuming no errors in the table). This discrepancy should be addressed.
>  -	The notation in the table has been changed to alleviate confusion between the ratio of the data used in the 5-fold cross-validation (for instance 100%) and the amount of data available for training (80 % in that case). The claim of 100% accuracy was with the full 100% dataset available for training. Moreover, the new initialization achieves 100% accuracy with as low as 40% in the budding dataset removing the need for the additional claim altogether: “In all four datasets, our method achieves more than 98 % accuracy with as little as 16 % of the data. Moreover, it achieves perfect accuracy with only 40 % of the data. In doing so, our method also discovers the ground-truth boolean networks by correctly identifying all of their necessary rules.”

---

> > ### Author Response · Authors · 2025-04-21
> >
> > -	Regarding the 100% Accuracy, could the network learn rules that are not included in or subsumed by the ground truth? Is the ground truth assumed to be complete (i.e., containing all possible rules/descriptions)? A discussion on whether the network could generate novel definitions for a target concept not in the ground truth would be insightful.
> >  -	In this case, the ground truth logic programs are indeed complete. However, the suggested discussion has been added to the case of tic-tac-toe interpretation where a novel definition was indeed discovered by the network: “However, since our search algorithm is stochastic, many different models with equivalent predictive performance can be obtained during learning with differing sets of rules. For instance, in preliminary tests, a model with perfect accuracy on tic-tac-toe was found that discovered the 3 rows and 3 columns of X, but instead of the 2 diagonals of X, it discovered 2 equivalent rules that each say that, if a diagonal has no O anywhere on it, then X wins. Such equivalent sets of rules can provide additional insight as in this case, although this variability can also be detrimental in other cases where different users may obtain different insights from the same data.”
> > -	On page 16, the paper refers to "half OR and half AND nodes" but does not explain what they are. Consider describing them in the related work section when the corresponding paper is cited.
> >  -	A reference to the section 2.2 detailing their special case has been added as a reminder: “The first is RRL (Wang et al., 2024) which uses the same modeling for the AND/OR nodes, with the exception of the missing bias and the need to double the weights to consider negated concepts (see section 2.2 for their special case).”
> > -	On the same page, the paper should explain how default values for fuzzy dichotomies and fuzzy intervals were defined. This discussion should be added in the section on dichotomies.
> >  -	The default values for fuzzy dichotomies and fuzzy intervals are defined previously in the subsection 3.2.2 on initialization.
> > -	Typos (Partial List) Page 5: either used directly $C_j^{(l-1)}$ -> used should be using Page 16: avoir -> avoid
> >  -	The typos have been corrected.

---

> > ### Comment · Reviewer_cZZ3 · 2025-04-23
> > **About probabilities and weights**
> >
> > I am not sure about the correctness of speaking of probabilities with negative weights. The general meaning of the sentence is clear, but it does not seem formally correct to speak of negative probabilities. Usually, the negation is represented by probability 0. For example, if we associate probability 0.0 with the characteristic ‘green’, the ball has no chance of being green. In this case, the value 0 is used to identify the value ‘unknown’. This, however, leads us to speak of a three-value semantics. However, if a three-value semantics is to be used, this semantics must be formally defined in the document, to mathematically explain how the probability distribution is defined.

---

> > > ### Author Response · Authors · 2025-04-28
> > >
> > > -	In my opinion, corrections proposed to the paragraph are not convincing. Specifically, if I correctly understood the meaning, the entire paragraph is useless because if we do not perform classification (i.e., no sigmoid or softmax are used), cross entropy is trivially not the smartest choice. On the other hand, ADAM basically minimizes every function. So, of course, it minimizes both L2 and cross-entropy. What am I missing here?
> > >  -	This small paragraph is admittedly somewhat stating the obvious. The superfluous information has been removed: “We use the ADAM optimizer (Kingma & Ba, 2015) to minimize the L2 loss. Its minimizer is $P[ω \in Y_k | x]$ which is precisely what we want our NLN’s outputs $c_k^2(x) = P[ω \in C_k^2 | x]$ to model.”
> > > -	I am not sure about the correctness of speaking of probabilities with negative weights. The general meaning of the sentence is clear, but it does not seem formally correct to speak of negative probabilities. Usually, the negation is represented by probability 0. For example, if we associate probability 0.0 with the characteristic ‘green’, the ball has no chance of being green. In this case, the value 0 is used to identify the value ‘unknown’. This, however, leads us to speak of a three-value semantics. However, if a three-value semantics is to be used, this semantics must be formally defined in the document, to mathematically explain how the probability distribution is defined.
> > >  -	The weights do not represent probabilities of a concept like “green” being present in the input, but rather probabilities that this concept is part of the definition of a rule for instance, either directly “green” or through its contrary opposite “not green”. A concept like green can have 3 possible roles in a rule, being part of the rule (1), its contrary being a part of the rule (-1), or not being part of the rule at all (0). As such our weights do not represent truth-values with three values (true, false, unknown), but encode the three possible roles of a concept in a AND/OR definition (used, opposite is used, not used). A sentence has been added to further clarify the associated probability distribution: “This modeling allows a single parameter to learn both possibilities simultaneously, since they are contradictory. However, doing so also assumes that at all times at least one of those probabilities e.g. $\mathbb{P}\left[C_j^{l- 1} \subseteq C_i^l\right]$, $\mathbb{P}\left[(C_j^{l- 1})^c \subseteq C_i^l\right]$} is zero, with the rest of the probability mass distributed between the other and $\mathbb{P}\left[ C_j^{l- 1} \not\subseteq C_i^l, (C_j^{l- 1})^c \not\subseteq C_i^l\right]$.”

---

### Review · Reviewer_2sT9 · 2025-02-24

**Summary Of Contributions:**

This paper presents a general framework for Neural Logic Networks (NLNs) within the context of Interpretable Machine Learning. The framework builds upon previous work on AND/OR logic networks by incorporating a NOT operator and addressing issues with partially defined data instances.  In this study, an NLN architecture is defined as a feedforward neural network, where each unit represents a literal (a concept or its negation). The layers alternate between conjunctive and disjunctive operations. The theoretical part of the study aims at providing a clear probabilistic semantics for NLNs. Notably, it relies on several assumptions to derive concept probabilities in an efficient way. In the experimental section, the NLN architecture, which includes two internal layers (representing probabilistic Disjunctive Normal Form concepts), is compared with other interpretable models across various tabular datasets.

**Audience:**

Yes

**Broader Impact Concerns:**

Not Applicable.

**Claims And Evidence:**

No

**Requested Changes:**

1) The framework of Neural Logic Networks (NLNs) should be compared in detail with probabilistic circuits. Notably, NLNs rely on very strong assumptions for their probabilistic semantics. In contrast, probabilistic circuits utilize structural constraints to manage probabilistic queries in a tractable way.
2) In Section 3, the learning task should be formalized as a standard stochastic optimization problem. This involves clearly identifying the objective, including any penalties, as well as the constraints.
3) Experiments should be conducted using high-dimensional datasets. I recommend utilizing datasets with hundreds of features, such as “Australian,” “Colic,” “Fashion,” “Hungarian,” “MNIST,” and others. Additionally, I suggest making comparisons with logistic circuits and employing MurTree (Demirovic et al., 2022) for comparison with optimal decision trees.

**References**

Emir Demirovic, Anna Lukina, Emmanuel Hebrard, Jeffrey Chan, James Bailey, Christopher Leckie, Kotagiri Ramamohanarao, Peter J. Stuckey: MurTree: Optimal Decision Trees via Dynamic Programming and Search. J. Mach. Learn. Res. 23: 26:1-26:47 (2022)

**Strengths And Weaknesses:**

**Strengths**

1) The paper is relatively well-written; the various concepts are introduced in a clear and pedagogical manner.
2) The framework extends previous Neural Logic Network (NLN) architectures by incorporating a NOT operator and addressing partially defined data.
3) The theoretical section provides a probabilistic semantics for NLNs, enabling comparisons with other graphical probabilistic models.

**Weaknesses**

1) The section on related work is incomplete, as it lacks a thorough overview of probabilistic graphical models. Importantly, NLNs have connections with well-studied probabilistic circuits (see, for example, Choi et al., 2020), which relate to AND/OR graphs and sum-product networks. In particular, logistic circuits (Liang & Van den Broeck, 2019) are a subclass of probabilistic circuits designed for interpretable classification tasks, allowing various probabilistic queries to be addressed in polynomial time.
2) The probabilistic semantics of Neural Logic Networks rely on very strong assumptions. Notably, the third assumption presented on Page 7 is unrealistic and often violated in practice. If this assumption is not satisfied, we lose much of the appeal of NLNs, as we cannot formally interpret the concepts involved in predicting the label of an incoming data instance in a probabilistic manner.
3) The learning task outlined in Section 3 is not well-defined. It is unclear what the objective function is (including its various regularizers) and what the different constraints entail.
4) Most of the comparative experiments are conducted on toy problems. In Table 2, only the "Adult" dataset presents a reasonable challenge. It would be beneficial to utilize higher-dimensional datasets to demonstrate that NLNs can scale effectively, especially in comparison with interpretable competitors.

**References**

* Choi, Y., Vergari, A., & Van den Broeck, G: Probabilistic circuits: A unifying framework for tractable probabilistic models. UCLA 2020.

* Yitao Liang, Guy Van den Broeck. Learning Logistic Circuits. In the Proceedings of the 33rd AAAI Conference on Artificial Intelligence (AAAI), 2019.

---

> ### Author Response · Authors · 2025-04-21
>
> Dear reviewer, thank you for your comments on the text and the experiments which have been very helpful in clarifying some ideas, adding more insights about the method, as well as adding a new high-dimensional dataset in the experiments. Each of your specific comments has been addressed below. Moreover, since the initial submission, a new initialization for the method has yielded better results all around, and the text was updated accordingly.
>
> -	The probabilistic semantics of Neural Logic Networks rely on very strong assumptions. Notably, the third assumption presented on Page 7 is unrealistic and often violated in practice. If this assumption is not satisfied, we lose much of the appeal of NLNs, as we cannot formally interpret the concepts involved in predicting the label of an incoming data instance in a probabilistic manner.
>  -	Indeed, this simplifying assumption is very problematic for NLNs. First, we have added a mention in the text that this assumption only holds for networks having a decomposable structure: “this assumption is only true if no two concepts share a common necessary/sufficient concept in the previous layer, a property known as decomposability in the probabilistic circuits literature (Choi et al., 2021)”.
>  -	Although this simplifying assumption is very problematic for NLNs, it is not always necessary. We have justified the probabilistic modeling without this assumption in the case of binary inputs, weights and biases. Moreover, we have explained how this can always be achieved with the exception of the biases: “As a purely probabilistic model, this issue is catastrophic for NLNs. However, as a ML method, this approximation can be justified. First of all, in the case where all inputs are binary (i.e. with binary or categorical features), where all the weights are given from the set {−1, 0, 1} and where the biases are full with $a_{\cdot}^l = 1$, $o_{\cdot}^l = 0$, then this assumption is not necessary. In fact, none of the assumptions are necessary in that case because the probabilistic formulation (P-AND) and (P-OR) coincide exactly with the logical definition of the AND and OR nodes (see section 2.2). In general, we always need the weights to be integers for maximum interpretability and, as such, we quantize them during post-processing when learning a NLN (see section 3.2.2). Moreover, for any network that contains continuous features, with enough pre-processing nodes (see section 3.1.2), the signal can become arbitrarily close to binary thanks to appropriately scaled sigmoid curves. In practice, we only relax this binary constraint through the possible non-binary biases, as well as by limiting the number of pre-processing nodes for continuous features, so that values away from 0 and 1 can be obtained close to the learned boundaries (see section 3.1.2). Therefore, we keep this approximation leading to (P-AND) and (P-OR) and, in practice, this modeling is still able to obtain promising predictive performance and interpretable rule discovery, even in non-binary cases.”
>
> ...

---

> ### Author Response · Authors · 2025-04-21
>
> -	The learning task outlined in Section 3 is not well-defined. It is unclear what the objective function is (including its various regularizers) and what the different constraints entail. […] In Section 3, the learning task should be formalized as a standard stochastic optimization problem. This involves clearly identifying the objective, including any penalties, as well as the constraints.
>  -	The section has been rewritten to clarify the use of the regularizers, and the objective, the full loss function as well as the domain constraints have been clarified with equations: “To help the learning process we regularize NLNs in two different ways. First, to combat the tendency of unadapted concepts to become trivial, we regularize the AND and OR concepts to have non-empty definitions. For instance, we consider the definition of an AND concept $C_i^l$ to be non-empty if
> $$\sum_j |A_{i,j}^l|\geq 1,$$
> in other words, we consider a concept $C_i^l$ to be non-empty if it attributes a probability mass of at least 1 across all of its possible input concepts $C_j^{l-1}$. We force non-empty definitions in all AND and OR concepts by penalizing
> L_{non-empy} = \sum_{AND weights A_{i,.}^l} || [1 - \sum_j |A_{i,j}^l|]_+ ||_2^2 + \sum_{OR weights O_{i,.}^l} || [1 - \sum_j |O_{i,j}^l|]_+ ||_2^2, [EQUATION DOES NOT APPEAR CORRECTLY HERE, SEE REVISED ARTICLE]
> which is only active when a concept $C_i^l$’s definition attributes a probability mass less than 1. In that case, the penalty will increase the weights of all of its input concepts $C_j^{l-1}$ uniformly until a probability mass of at least 1 is attributed. Moreover, in order to encourage sparser, more interpretable solutions, we also penalize the L1 norm of all weights in the network. The full loss function is then given by
> L(y, c^L(x)) = ||y – c^L(x)||_2^2 + λ_{non-empty} · L_{non-empty} + λ_{sparsity} ( \sum_{AND weights A_{i,.}^l} ||A_{i,.}^l||_1 + \sum_{OR weights O_{i,.}^l} ||O_{i,.}^l||_1 ), [EQUATION DOES NOT APPEAR CORRECTLY HERE, SEE REVISED ARTICLE]
> where λ_{non-empty}, λ_{sparsity} > 0 are the regularization coefficients of the non-empty penalty and the sparsity penalty respectively. We minimize the expectation of this loss
> $$E_{(X,Y )∼D}[L(Y, c^L(X))]$$
> over the training dataset D, subject to the domain constraints of the weights $A_{i,j}^l \in [−1, 1]$, $O_{i’,j’}^{l’} \in [0, 1]$, the biases $a_i^l, o_{i’}^{l’} \in [0, 1]$, and the parameters of the fuzzy dichotomies $B_{i’’,k} \in R$, $α_{i’’,k} > 0$ for appropriate indices $(i, j, l)$, $(i', j’, l’)$ and $(i′′, k)$ according to the NLN’s structure.”
> -	The section on related work is incomplete, as it lacks a thorough overview of probabilistic graphical models. Importantly, NLNs have connections with well-studied probabilistic circuits (see, for example, Choi et al., 2020), which relate to AND/OR graphs and sum-product networks. In particular, logistic circuits (Liang & Van den Broeck, 2019) are a subclass of probabilistic circuits designed for interpretable classification tasks, allowing various probabilistic queries to be addressed in polynomial time. […] The framework of Neural Logic Networks (NLNs) should be compared in detail with probabilistic circuits. Notably, NLNs rely on very strong assumptions for their probabilistic semantics. In contrast, probabilistic circuits utilize structural constraints to manage probabilistic queries in a tractable way.
>  -	We have added a paragraph in the related works section on probabilistic graphical models and probabilistic circuits: “Due to their probabilistic formalization, NLNs also serve as probabilistic models of the target classes/labels when conditioning on the input features. In doing so, they implicitly learn Probabilistic Graphical Models (PGM), which encode the conditional structure of random variables in graph form (see appendix B.1.1 for the PGM behind NLN’s probabilistic modeling). However, NLNs require an additional approximation to ensure tractability in practical settings. With additional assumptions regarding the direction of causality, NLNs can also be viewed as a special form of structural causal model (Peters et al., 2017) in which the assignment functions have a clear interpretation as AND/OR combinations of binary random variables, some of which may be unobserved. However these stronger causality assumptions are not required to use NLNs, and are only needed to produce interventional distributions or counterfactuals. NLNs are also related to probabilistic circuits (Choi et al., 2021) which study the tractability of probabilistic queries in sum-product networks via structural constraints. In particular, logistic circuits (Liang & Van den Broeck, 2019) are probabilistic circuits with strong structural constraints that combine structure learning and logistic regression to learn AND/OR networks for interpretable classification. However, unlike NLNs, their final learned AND/OR networks contain non-quantized weights which are not as easily interpretable.”

---

> > ### Author Response · Authors · 2025-04-21
> >
> > -	Most of the comparative experiments are conducted on toy problems. In Table 2, only the "Adult" dataset presents a reasonable challenge. It would be beneficial to utilize higher-dimensional datasets to demonstrate that NLNs can scale effectively, especially in comparison with interpretable competitors. […] Experiments should be conducted using high-dimensional datasets. I recommend utilizing datasets with hundreds of features, such as “Australian,” “Colic,” “Fashion,” “Hungarian,” “MNIST,” and others.
> >  -	The idea of using high-dimensional datasets to test the scaling properties of NLNs is a pertinent addition. The suggested datasets however are not appropriate for us: “Australian”, “Colic” and “Hungarian” only have a hundreds of features if the original features are binarized which is not necessary for NLNs, the original features (< 30) can be used directly and they should be used directly in a framework like NLNs so that the final learned network is interpretable in the simplest way; “Fashion” and “MNIST” are images which would be better served by a convolutional version of NLNs to introduce translation invariance, if they were used as tabular inputs as in the logistic circuit article (Liang & Van den Broeck, 2019), the resulting learned rules would not be very insightful as they would not help to understand the general shapes of each target classes, since they could only use pixel markers in the image. We will instead work with the “DARWIN” dataset which is tabular and has 450 continuous features.
> > -	Additionally, I suggest making comparisons with logistic circuits and employing MurTree (Demirovic et al., 2022) for comparison with optimal decision trees.
> >  -	Logistic circuits, although they are similar in structure as AND/OR networks are not totally comparable with NLN, RRL, decision diagrams and decision trees since their final models contain non-quantized weights which are not as easily interpretable. For the comparison with optimal decision trees, we do not see how MurTree brings any additional value beyond our current comparison with optimal decision trees from (Florio et al., 2023).
> >
> > References:
> > Alexandre M. Florio, Pedro Martins, Maximilian Schiffer, Thiago Serra, and Thibaut Vidal. Optimal decision diagrams for classification. Proceedings of the AAAI Conference on Artificial Intelligence, 37(6):7577–7585, Jun. 2023.

---

> ### Author Response · Authors · 2025-04-23
>
> Regarding the Fashion and MNIST datasets, we would like to add  that the current work is meant to be a proof of concept for tabular classification that can then be leveraged for more complex tasks, such as being adapted to graph NLN (rule mining in knowledge graphs), recurrent NLN (multi-step reasoning) and convolutional NLN (image classification). This is a point that we have made more explicit in the conclusion of the paper: “Interpretable tabular classification is a starting point for NLNs. In future research, we will explore how they can be adapted to more complex tasks by using different neural network structures. For instance, convolutional NLNs could leverage the AND/OR nodes to tackle interpretable image classification. With convolutional AND kernels and pooling OR layers, these networks could produce interpretable representations of higher-level concepts from 2D arrangements of lower-level concepts. Another example is recurrent NLNs for multi-step reasoning. The tabular NLN presented here can only do single-step reasoning, but by chaining this reasoning through multiple steps, a recurrent NLN could produce multi-step reasoning and solve problems like Sudoku. Again, like in tabular NLN, by learning to predict the finished Sudoku puzzle, the NLN could also discover the rules of Sudoku. Finally, graph NLNs, by working on input graphs with entities and relations, would introduce features of first-order logic by generalizing AND/OR nodes to define universal quantification $\forall$ and existential quantification $\exists$. Moreover, in doing so, graph NLNs could learn to not only predict missing edges or attributes, but also discover underlying relational rules, also known as rule mining in knowledge graphs.”.

---

### Review · Reviewer_uMHn · 2025-04-19

**Summary Of Contributions:**

This paper extends Neural Logic Networks by enabling support for logical negation operations and incorporating biases that account for unobserved data. The authors propose a probabilistic rule-learning framework consisting of a pre-processing module for encoding categorical and continuous features, a rule module that learns interpretable logical structures using Adam optimization with regularization, tailored initialization, and a rule-reset mechanism, and a post-processing step that binarizes the learned weights for rule extraction. Experiments on Boolean network discovery and tabular classification tasks demonstrate that the proposed approach outperforms a range of baselines in terms of both accuracy and efficiency.

**Audience:**

No

**Claims And Evidence:**

No

**Requested Changes:**

Please consider adding harder and more realistic tasks to better demonstrate the model’s capabilities—e.g., at least solving Sudoku or similar structured reasoning problems that go beyond small toy datasets.

**Strengths And Weaknesses:**

**Strengths:**
- The paper tackles the important problem of improving neural network interpretability through symbolic methods, which is an interesting research direction.
- The proposed approach demonstrates strong empirical performance, outperforming a range of baseline methods in both accuracy and efficiency.

**Weaknesses:**
- The paper is somewhat difficult to follow.
- Neural Logic Networks are limited to propositional logic, which restricts their applicability to more complex reasoning tasks. There is existing work on learning first-order logic rules, which may offer broader expressiveness and reduce the relative impact of this paper.
- The experiments are fairly toyish and may not convincingly demonstrate real-world applicability. Additionally, some relevant baselines—such as SATNet [1] and Recurrent Relational Networks [2]—appear to be missing from the evaluation.

[1] SATNet: Bridging deep learning and logical reasoning using a differentiable satisfiability solver

[2] Recurrent Relational Networks

---

> ### Author Response · Authors · 2025-04-23
>
> Dear reviewer, thank you for your comments and especially your suggestion of adding Sudoku as an extra task. Although we believe it might not be appropriate for the current paper, it is a very interesting idea that we will pursue in future work. We describe below why we do not believe it is appropriate here and address each of your specific comments. Moreover, since the initial submission, a new initialization for the method has yielded better results all around, and the text was updated accordingly.
>
> -	The paper is somewhat difficult to follow.
>  -	The first revision of the paper in response to the first 2 reviews has improved its readability thanks to many small suggestions of the first 2 reviewers, including reformulation of some section titles, restructuring of some subsections, introductory sentences to guide the reader in the beginning of some sections that did not have them, the addition of formal definitions where previously only narrative descriptions were given, as well as the clarification of some ideas and the addition of more insights about the method.
> -	Neural Logic Networks are limited to propositional logic, which restricts their applicability to more complex reasoning tasks. There is existing work on learning first-order logic rules, which may offer broader expressiveness and reduce the relative impact of this paper.
>  -	In order to incorporate first-order logic features in NLNs such as existential and universal quantification, a different type of input than tabular data would be needed like input knowledge graphs, which have entities to quantify over. For this, NLNs would have to be adapted to a graph neural network structure. However, the scope of the current work is limited to tabular classification with fixed-length vectorized input that does not require quantification and is well adapted to propositional logic.
> -	The experiments are fairly toyish and may not convincingly demonstrate real-world applicability. […] Please consider adding harder and more realistic tasks to better demonstrate the model’s capabilities—e.g., at least solving Sudoku or similar structured reasoning problems that go beyond small toy datasets.
>  -	The second reviewer has also suggested to increase the level of difficulty in the tabular datasets, specifically suggesting that we test our method on higher-dimensional datasets. We have thus added the DARWIN dataset which aims to classify patients with Alzheimer’s from 450 continuous input features that were measured in handwriting tests. In order to use a NLN to solve Sudoku, its structure would have to be adapted to a recurrent neural network (just as Recurrent Relational Networks) to produce the multi-step reasoning that is required in Sudoku. While it is a very interesting idea and one that we will pursue in future work, it cannot be achieved by the tabular NLNs covered in this work which can only produce single-step reasoning. In fact, this is a similar problem to the datasets that the second reviewer suggested for high-dimensional datasets which were image datasets. Those datasets would require a convolutional NLN to discover interpretable rules. The current work is meant to be a proof of concept for tabular classification that can then be leveraged for more complex tasks, such as being adapted to graph NLN (rule mining in knowledge graphs), recurrent NLN (multi-step reasoning) and convolutional NLN (image classification). This is a point that we have made more explicit in the conclusion of the paper: “Interpretable tabular classification is a starting point for NLNs. In future research, we will explore how they can be adapted to more complex tasks by using different neural network structures. For instance, convolutional NLNs could leverage the AND/OR nodes to tackle interpretable image classification. With convolutional AND kernels and pooling OR layers, these networks could produce interpretable representations of higher-level concepts from 2D arrangements of lower-level concepts. Another example is recurrent NLNs for multi-step reasoning. The tabular NLN presented here can only do single-step reasoning, but by chaining this reasoning through multiple steps, a recurrent NLN could produce multi-step reasoning and solve problems like Sudoku. Again, like in tabular NLN, by learning to predict the finished Sudoku puzzle, the NLN could also discover the rules of Sudoku. Finally, graph NLNs, by working on input graphs with entities and relations, would introduce features of first-order logic by generalizing AND/OR nodes to define universal quantification $\forall$ and existential quantification $\exists$. Moreover, in doing so, graph NLNs could learn to not only predict missing edges or attributes, but also discover underlying relational rules, also known as rule mining in knowledge graphs.”.
>
> ...

---

> > ### Author Response · Authors · 2025-04-23
> >
> > -	Additionally, some relevant baselines—such as SATNet [1] and Recurrent Relational Networks [2]—appear to be missing from the evaluation.
> >  -	Although these baselines are appropriate for comparison with methods that incorporate symbolic reasoning capabilities in a differentiable model that is learned end-to-end, they are not easily interpretable, which is the focus of this paper. SATNet has somewhat interpretable weights representing clauses in a MAXSAT problem, but since they are not quantized during learning or in post-processing, they are not as easily interpretable as the purely symbolic rules that are discovered by NLNs, RRL and decision trees/diagrams. Recurrent Relational Networks are simply not interpretable at all due to their MLPs.

---

### Decision · Action_Editor_qDvN · 2025-06-09

**Recommendation:** Reject

**Additional Comments:**

The manuscript is readable and the technical contribution reasonable.  The main issue is that two reviewers found the experimental setup limited and in need for expansion.  I recommend the authors to consider additional and more challenging datasets which ideally should showcase the practical benefits of the proposed framework.

**Audience:**

Yes

**Audience Explanation:**

Yes.  Researchers studying rule learning and neuro-symbolic integration would find this paper of interest.

**Claims And Evidence:**

No

**Claims Explanation:**

The authors make several claims.  The theoretical ones are supported by the extensive manuscript.

The authors claim at the end of the Introduction claim that they "will show that our NLN, with its factorized structure and extended modeling, is able to learn sparser and more interpretable rules than its predecessor, the RRL, in tabular classification". This seems to be supported by Table 2.

However, Table 4 also indicates that NLN does not perform as well as other SOTA methods in tabular classification tasks.  While the authors make no major claim in this regard, in some settings, the performance gap wrt to the best method is 20% F1 or more. This downside is quite obvious and rather central, and cannot be overlooked. One reviewer was especially vocal about this.

Two reviewers also lamented the limited complexity and scale of the learning tasks.

The claim that NLNs are truly interpretable is difficult to assess, but this is -- unfortunately -- often the case for similar claims in the ML literature.

**Resubmission Of Major Revision:**

The authors may consider submitting a major revision at a later time.